# SCREENWRITER: AUTOMATIC SCREENPLAY GENERATION AND MOVIE SUMMARISATION

## ABSTRACT

The proliferation of creative video content has driven demand for textual descriptions or summaries that allow users to recall key plot points or get an overview without watching. The volume of movies available for viewing and speed of turnover motivate automatic summarisation, which is challenging, requiring the identification of character intentions and long-range temporal dependencies. The few existing methods attempting this task rely heavily on textual screenplays as input, greatly limiting their applicability. In this work, we propose the task of automatic screenplay generation, and a method, ScreenWriter, that operates only on video and produces output which includes dialogue, speaker names, scene breaks, and visual descriptions. ScreenWriter introduces a novel algorithm to segment the video into scenes based on the sequence of visual vectors, and a novel method for the challenging problem of determining character names, based on a database of actors' faces. We further demonstrate how automatically generated screenplays can be used to create plot synopses with a hierarchical summarisation method based on scene breaks. We test the quality of the final summaries on the recent MovieSum dataset, which we augment with videos, and show that they are superior to several comparison models that assume access to goldstandard screenplays.

## 1 INTRODUCTION

Thanks to the proliferation of streaming services and digital content providers, a large number of movies are being released and made available every year. Automatic approaches to understanding and summarising their content are paramount to enabling users to browse or skim through them, and quickly recall key plot points, characters, and events without the need to re-watch from the beginning. Aside from practical utility, movie summarisation is an ideal testbed for real-world video and natural language understanding. Movies are often based on elaborate stories, with non-linear structure and multiple characters whose emotions and actions are often not accompanied with verbal cues. The summarisation task requires identifying beliefs and intentions of characters as well as reasoning over very long narratives (there are usually hundreds of pages in a transcript and tens of thousands of frames in a video), involving multiple modalities.

Most previous work on movie summarisation has focused primarily on the textual modality, under the assumption that screenplays (or at least human-authored transcripts) are readily available (Gorinski & Lapata, 2015; Chen et al., 2022; Agarwal et al., 2022). Movie summarisation is commonly viewed as a type of long-form summarisation, a task which has improved dramatically in recent years. This is partly due to large models with longer context windows, and the design of methods specifically for this task, such as dividing the input into chunks and forming a hierarchical summarisation structure (Chen et al., 2023a; Pang et al., 2023) or summarising chunks iteratively (Chang et al., 2023). Video understanding through generating descriptions for videos, has also received much attention, largely independently from long-form summarisation, focussing instead on short video clips lasting a few minutes (Tapaswi et al., 2016; Lei et al., 2018; Rafiq et al., 2023). Two notable exceptions are Papalampidi & Lapata (2023) and Mahon & Lapata (2024) who consider textual *and* visual modalities, aiming to summarise movies and television shows, respectively. However, they both require a textual transcript with information about scene breaks, and character names prefacing the lines actors speak. The assumption of having available written transcripts and/or screenplays is unrealistic, as most video providers do not have access to screenplays un-

less they have produced the content themselves. Additionally, there are often differences between original screenplays, and what is acted and shown in the movie.

In this paper, we address the challenging tasks of generating automatic screenplays and plot summaries of movies by leveraging only video and audio input, without relying on transcripts or text. As a result, our method, ScreenWriter is applicable to the large quantity of movie videos which exist without accompanying screenplays or human-annotated transcripts. A naive solution is to apply a speech-to-text model from which we could then build a rudimentary screenplay automatically. Using speech-to-text off the shelf for movie understanding is problematic for several reasons. Firstly, it does not contain scene breaks, which can provide useful structural and semantic information (Gorinski & Lapata, 2015; Papalampidi et al., 2019). Secondly, it does not contain descriptions of events outside of dialogue such as characters fighting or kissing. And perhaps most importantly, it lacks character names. Speaker diarization aims to determine whether different utterances are spoken by the same speaker, but this speaker is still indicated with an arbitrary ID, rather than a character name.

ScreenWriter attempts to reconstruct screenplays and their sctructure as accurately as possible. It eschews the above mentioned pitfalls by segmenting the video into scenes using a novel parameter-free method based on the minimum description length principle coupled with a dynamic programming search algorithm. It associates speaker IDs to character names through a novel method which exploits a database of actor face images, coupled with the name of the character they played (which is readily available online). By computing distances between face feature vectors in the same scene as the utterance, we formulate the task as an instance of the linear sum assignment problem, which can then be solved efficiently by known algorithms. Additionally, we add event descriptions (as an approximation to action lines which describe what the camera sees) using an image-to-text model on keyframes detected in each scene. We further demonstrate that automatically generated screenplays are good enough to produce textual summaries: we condense each scene independently and then fuse the information from all scenes into a final synopsis using zero-shot prompting.

ScreenWriter, and the accompanying summarisation method, is therefore modular, based on a number of interacting components that separately solve different subtasks. This design differs from recent work Song et al. (2024) proposing to modify transformer memory in order to handle longer video sequences. Scaling such end-to-end models to full-length movies remains a significant challenge due to memory constraints and the complexity of extracting useful information from large inputs.[1] Our approach, which breaks the task into subproblems which are then solved separately, mitigates this issue and allows for more efficient processing and summarisation. In summary, our contributions include:

- Proposing the new task of screenplay generation from video (and audio), and a modular framework, ScreenWriter, to address this task;
- A novel, parameter free method for detecting scene boundaries in video;
- A novel algorithm for detecting character names based on a database of actors' faces;
- A-proof-of-concept summariser which takes automatic screenplays from ScreenWriter as input and produces textual synopses following a hierarchical method with zero-shot prompting.

## 2 RELATED WORK

**Video Understanding** The problem of generating descriptions for videos has received significant attention in the literature. Traditional video description approaches often extract features from individual frames and fuse them into a single feature vector to generate a textual description (Zhang et al., 2021; Pan et al., 2020; Ye et al., 2022). SwinBERT (Lin et al., 2022) introduces an end-to-end video network that samples frames densely, avoiding the need for image-based encoders. Similarly, Lei et al. (2020) use a memory-augmented vision transformer to generate descriptions for short videos. In the area of video segmentation, Souček & Lokoč (2020) introduce TransNet v2, a deep learning model for shot transition detection. However, shot boundaries are easier to detect than scene boundaries because there is a more striking pixel-level discontinuity.

---

[1]At 1,024 × 1,024 frame size, and 10 frames per second, a 75min movie would consume over 500GB when represented as a 4d 32-bit float tensor.

Despite this progress, many popular video description datasets (Chen & Dolan, 2011; Xu et al., 2016) contain videos that are only ∼10s in length, which are much shorter than full-length movies. YouCook (Zhou et al., 2018a), a dataset with longer videos (about 5 minutes on average), remains insufficient for capturing the complexities of movie understanding and summarisation. Egoschema (Mangalam et al., 2023) explores video understanding for clips of ∼3 minutes, focusing on human activity and behavior, but this too is far shorter and more constrained than full movie summarisation. Some methods have started to tackle movie content, but in the form of much shorter clips. Wu & Krahenbuhl (2021) use YouTube movie clips for small tasks like classifying movie year or genre, but lack a comprehensive approach for understanding/summarising an entire movie. Chen et al. (2023b) propose Movies2Scenes, a method that uses movie metadata to learn scene representations, though it relies on predefined scenes based on shot transitions rather than semantically meaningful boundaries. Han et al. (2024) address the problem of assigning names to characters within a local movie clip, using a "character bank" of images which is then used to prompt a LLM for character names. This differs from our method in the use of a LLM for name assignment, and in that we are assigning names to speaker IDs that arise from speaker diarization, rather than to dialogue directly. Moreover, our assignment concerns the entire movie, rather than a short snippet. Some works aim to summarise short videos, a task referred to as video captioning. Sridevi & Kharde (2020) summarizes short videos using a two-stream CNN. Seo et al. (2022) use a bidirectional model that uses both video and audio to produce video captions. Zhou et al. (2018b) propose a single masked transformer objective to detect and then caption all events in a moderate length ( 3min) video Unsupervised pertaining has also been explored, e.g. by Yang et al. (2023), who train a video-captioning model using transcribed utterances as pseudo-captions. Some systems based on large proprietary models have been proposed for longer videos. Zhang et al. (2024); Lin et al. (2023), include multiple modules, including visual GPT-4, and use PysceneDetect for scene breaks. Wu et al. (2024) prompts an LLM to predict scene breaks from transcribed speech and captions, which is then used for video question-answering.

**Long-form summarisation** Recent work has started to address the task of summarising much longer videos. Papalampidi et al. (2021) describe a method for summarising full-length movies by creating shorter videos containing their most informative scenes which they assume to be 'turning points' (i.e., key events in a movie). Papalampidi & Lapata (2023) produce text summaries of TV show episodes, by converting visual features into embeddings alongside word embeddings from the transcript. Mahon & Lapata (2024) also summarise TV show episodes from video and text input, by converting the video to text, and then treating it as a text-only problem. Our method differs from all of these approaches in that it only requires video as input, tackling the more challenging task of understanding who is speaking and what they are doing (this information is taken for granted in screenplays and human-authored transcripts).

**Movie Datasets** Datasets containing videos for full-length feature movies are few and far between possibly due to copyright restrictions and the computational overhead of storing and processing lengthy videos. Papalampidi et al. (2024) introduce TRIPOD, a dataset consisting of screenplays, their movies, and corresponding trailers, however no summaries or synopses are provided. MovieNet (Huang et al., 2020) contains summaries and other annotations for movies, but not the videos themselves, which are specifically the focus of the present work.[2] Papalampidi & Lapata (2023) release SummScreen$^{3D}$ a dataset consisting of soap opera episodes, their (crowd-sourced) transcripts, and summaries. However, these episodes are not self-contained, as they typically refer to events or characters from previous episodes. In addition, they tend to be relatively short, and dialogue-heavy, with recurring scene locations. In this work, we wish to explore longer videos of 70+ minutes, and move away from the niche domain of soap operas. In experiments, we use the recent MovieSum dataset (Saxena & Keller, 2024) which contains screenplays and their Wikipedia plot summaries. We augment this dataset with videos that we privately procure for the test set.

## 3 SCREENWRITER

Figure 1 provides a graphic depiction of ScreenWriter. As mentioned earlier, the input to Screen-Writer is the video (including audio) for a full movie. We extract keyframes from the movie and

---

[2]The MovieNet website claims that videos will be released soon, but at the time of writing, despite several years wait, they have not appeared.

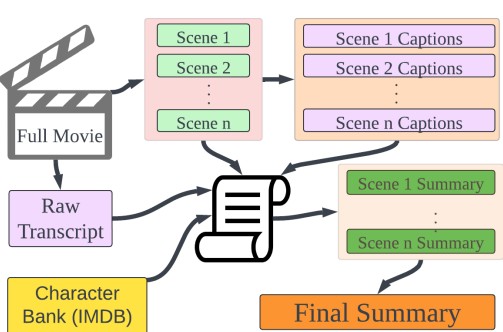

Figure 1: Screenwriter produces a screenplay from the input video/audio (top left), by first extracting the raw transcript, segmenting the video into scenes, and generating visual descriptions from each scene (top right). Then from these outputs, and the character bank (bottom left), it assigns character names and generates the screenplay (centre). Given the generated screenplay, we can then produce summaries for each scene (centre right), and fuse into a single summary for the entire movie (bottom right).

use our novel scene segmentation algorithm (see Section 3.2) to partition the resulting sequence of frames into different scenes. In parallel, a text-to-speech model with speaker diarization yields a transcript with numeric speaker IDs instead of character names. For each movie, we assume access to a database consisting of actors faces and their character names (such a database can be easily constructed by scraping the faces of the characters from the movie's IMDB page). We extract facial features from each keyframe in each scene, and from each scraped character image, and use our novel name assignment algorithm to replace (some of) the numeric IDs in the transcript with actual character names (see Section 2). Finally, to insert visual information into the screenplay about what is happening on camera, we extract textual descriptions from three evenly spaced keyframes in each scene using a multimodal large language model. We now describe these components in detail.

### 3.1 TRANSCRIPT GENERATION

The first step in ScreenWriter is to extract the audio from the movie, using an automatic transcription model with speaker diarization (Bain et al., 2023). This produces a transcript with each utterance marked by a speaker ID. The speaker ID is the same for the same character throughout the movie (up to the accuracy of the diarization), but the character names are lacking. We also extract key frames from the entire video, and store the timestamps of these keyframes. These will be used to compute speaker names and visual information, which we will add to the diarized transcript to form a screenplay. It is also on this sequence of keyframes that the scene breaks are computed. There is an average of around 1,000 detected keyframes for a feature-length movie.

### 3.2 SCENE DETECTION

ScreenWriter extracts visual features from each keyframe, and applies a novel algorithm to segment the resulting sequence of visual feature vectors. There are two parts to our algorithm: the *definition of a cost* for a particular partition into scenes, and the *search* for the partition that minimizes this cost. The first part, the cost definition, is formulated using the minimum description length principle, which claims the correct representation of the data is the one using the fewest bits. We assume that the vectors for each scene are encoded with respect to their collective mean. That is, for each scene in the given partition, we calculate the mean of all vectors in that scene, and hence, the probability of each vector, $p(v)$, under the multivariate normal distribution with this mean. To reduce run time, we use a single fixed covariance calculated from the entire sequence of vectors. The Kraft-McMillan inequality (Kraft, 1949; McMillan, 1956) then determines that under the optimal encoding, the number of bits needed to represent $v$ is $-\log_2 p(v)$. The sum of this value across all $N$ vectors $v$ in the video, plus the number of bits to represent the mean vectors themselves, gives the total bitcost for a given partition. The mean vectors require $dm$ bits, where $d$ is the dimensionality, and $m$ is the floating point precision, for which we use the standard of 32. Partitions with more scenes require more bits for the mean vectors, but also have mean vectors that better cover the keyframe features, leading to decreased $-\log_2 p(v)$ on average. This trade-off encourages a partition with neither too few nor too many scene breaks.

The second part, the search for the minimizer of the above cost, can be solved exactly using dynamic programming. Let $B(i, j)$ be the cost of having a single scene that runs from keyframes $i$ to $j$, and let $C(i, j)$ be the minimum cost of all keyframes from $i$ to $j$, under all possible partitions. Then we

---

**Algorithm 1** Scene Boundary Detection

---

1: **Input:** Video file
2: Extract keyframes, $kf_0, \ldots, kf_N$
3: Extract visual features $v_0, \ldots, v_N$ from each keyframe
4: $L \leftarrow 50$          $\triangleright$ maximum scene length
5: $B \leftarrow N \times N$ empty matrix      $\triangleright$ $B[i,j]$ will hold the cost of a scene from $v_i$ to $v_j$
6: $\Sigma \leftarrow$ empirical covariance matrix of $v_0, \ldots, v_N$
7: $d \leftarrow$ dimensionality of $v_i$
8: $m \leftarrow$ floating point precision of $v_i$

9: **Cost Definition:** Compute and store costs for all possible scenes
10: **for** $0 \le i \le N - L$ **do**
11:      **for** j=i, ..., i+L **do**
12:          $\mu_{i,j} \leftarrow \frac{1}{j-i} \sum_{k=i}^{j} v_k$
13:          $C \leftarrow dm$
14:          **for** $k = i, \ldots, j$ **do**
15:              $p(v_k) \leftarrow \frac{1}{(2\pi)^{d/2}|\Sigma|^{1/2}} \exp\left(-\frac{1}{2}(v_k - \mu_{i,j})^\top \Sigma^{-1}(v_k - \mu_{i,j})\right)$
16:              $C \leftarrow C - \log p(v_k)$
17:          $B[i,j] \leftarrow C$

18: **Search:** Minimize the bitcost by dynamic programming
19: $C \leftarrow B$          $\triangleright$ will hold optimal costs
20: $P \leftarrow N \times N$ matrix of empty sets      $\triangleright$ will hold optimal partitions
21: **for** i=N-1, ..., 0 **do**
22:      **for** $j = i, \ldots, \min(N, i+L)$ **do**
23:          **if** $B[i,j] + C[j,N] < C[i,N]$ **then**
24:              $C[i,N] \leftarrow B[i,j] + C[j,N]$
25:              $P[i,N] \leftarrow \{j\} \cup P[j,N]$

26: **Output:** Optimal scene partition, $P[0,N]$

---

have the recurrence relation

$$C(i,j) = \min_{i \le k \le j} B(i,k) + C(k,j). \tag{1}$$

Thus, we can compute the optimal partition by iteratively computing and caching $C(i,N)$ for $i = N - 1, \ldots, 0$. This is guaranteed to find the global optimum. It runs in $O(N^2)$, but by imposing a fixed threshold of the maximum number $L$ of keyframes in a scene, this becomes $O(N)$. In our experiments, we find that setting $L = 50$ produces no change in the solution. The full scene detection method is shown in Algorithm 1.

### 3.3 CHARACTER NAME IDENTIFICATION

To replace the arbitrary speaker IDs with character names, we first create a database of images of actors' faces paired with the name of the character they played from the IMDB movie page. As some of these images may contain multiple faces, or no faces, or even an entirely different character, we filter them to ensure a higher proportion contain only the face of the correct character, keeping only images with exactly one detected face, and for which the detected gender matches the name gender. Finally, we verify the faces in all pairs of remaining images against each other, using the DeepFace[3] library, to create a graph where images are connected if and only if they are verified as being the same person, and then exclude all images that are not part of the largest clique. In total, we filter out about 40% of images on average. This produces a name bank of character names paired with a set of images of the face of that character.

For each scene, and for each character in our name bank, we define the cost of putting that character name in that scene as the minimum distance between an image of that character's face, and a face detected in any keyframe from the scene. The distance is the Euclidean distance of the DeepFace feature vectors. This avoids the incorrect assumption that the character speaking must be in shot, and

---

[3]https://github.com/serengil/deepface

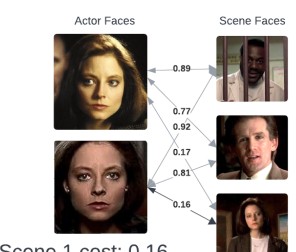 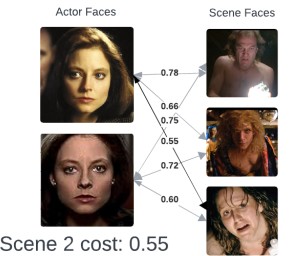 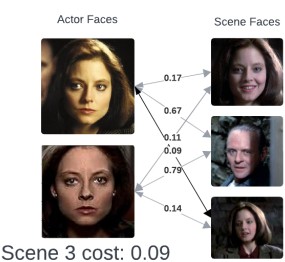

Figure 2: Computing the cost of assigning the character Clarice Starling (Jodie Foster) to three different scenes of *The Silence of the Lambs* (1991). After computing the cost of assigning a character to a each scene, we then compute the cost of assigning a character to a speaker ID as the mean of the cost of assigning them to all scenes that speaker ID appears in.

instead makes the much weaker assumption that a character speaking must appear directly at some point in the scene. Thus, if we are considering assigning the character Clarice Starling to scene 3, then we compute the distance between the face feature vectors for all scraped images of the actor Jodie Foster in that role, and the face feature vectors of all faces detected in any keyframe in scene 3; the smallest distance is the cost of assigning Clarice Starling to scene 3. Computing the distance between vectors is extremely fast, taking <1s for all considered assignments on the entire movie, and the feature vectors can be cached after being extracted once. An example of this cost computation is shown in Figure 2. Using this cost, we define the cost of assigning each character to each speaker ID, as the sum of assigning that character to all scenes that that speaker ID appears in For example, if Speaker18 appears in scenes 1 and 3 but not 2, then the cost of assigning Clarice Starling to Speaker18 is the mean of the cost of assigning Clarice Starling to scenes 1 and 3. This allows us to treat the name-speaker ID assignment problem as an instance of the linear sum assignment problem, which can be solved efficiently using the Kuhn-Munkres algorithm (Kuhn, 1956; Munkres, 1957).

Specifically, we define a matrix $S$ whose $i, j$th entry is the cost of assigning speaker $j$ to name $i$. Let $m$, $n$, and $k$ be the numbers of character names in the database, scenes in the movie, and unique speaker IDs in the transcript. Using matrix notation, we can then write $S = AB$, where $A$ is the $m \times n$ speaker ID-scene cost matrix, whose $i, j$th entry is the cost of assigning speaker $j$ to scene $i$, and $B$ is a $n \times k$ matrix whose $i, j$th entry is $1/a$ if speaker ID $j$ appears in scene $i$, where $a$ is the number of scenes speaker $j$ appears in, and 0 otherwise. Because speaker diarization is imperfect and often mistakenly splits the same character into multiple IDs, we duplicate each matrix column three times, which allows up the three different speaker IDS assigned to the same character name. We also define a cost of leaving a SpeakerID unassigned as expected value of the cost of assigning a random speaker ID to a random character, which means that an ID remains unassigned if it is no closer to any character than a random speaker ID and character are to each other. The full name-assignment method is shown in Algorithm 2 in Appendix A.

## 3.4 SCREENPLAY GENERATION

After running the assignment algorithm from Section 3.3, we can replace the common speaker IDs in the automated, diarized transcript with the corresponding character names. Also, by matching the utterance times with the keyframe timestamps, we can insert the scene breaks from Section 3.2 into the transcript. Finally, we include descriptions of the visual contents of the scene by selecting three evenly spaced keyframes from that screen, applying an image captioning model (Peng et al., 2023), and inserting the output to the corresponding timestamped location of the transcript. In the caption, we replace occurrences of the nouns 'person', 'woman', 'man', 'girl', and 'boy' (and their determiners) with a character name, if this can be inferred from gender matching with the speaker names. For example, if the only non-male name in the script is 'Clarice Starling', then the occurrence of 'a woman' in the caption is replaced with 'Clarice Starling'. The result of adding names, scene breaks and visual descriptions to the transcript is an automatically generated screenplay. We show an example of ScreenWriter output in Figure 3.

**Dr. Hannibal Lecter:** Billy is not a real transsexual. But he thinks he is. He tries to be. He's tried to be a lot of things, I expect.
**Clarice Starling:** You said that I was very close to the way we would catch him. What did you mean, Doctor?
**Dr. Hannibal Lecter:** There are three major centers for transsexual surgery. Johns Hopkins, University of Minnesota and Columbus Medical Center. I wouldn't be surprised if Billy had applied for sex reassignment at one or all of them and been rejected.
**Clarice Starling:** On what basis would they reject him?
**Dr. Hannibal Lecter:** Look for severe childhood disturbances associated with violence. Our Billy wasn't born a criminal, Clarice. He was made one through years of systematic abuse. Billy hates his own identity, you see. But his pathology is a thousand times more savage and more terrifying.

*Dr. Hannibal Lecter sits in a chair, and Clarice Starling stands next to him holding a book.*

**Jame Gumb:** It rubs the lotion on its skin. It does this whenever it's told.
**Catherine Martin:** Mr, my family will pay cash. Whatever ransom you're asking for, they'll pay it.
**Jame Gumb:** It rubs the lotion on its skin or else it gets the hose again. Yes, you will, precious. You will get the hose.
**Jame Gumb:** Okay. Okay. Okay. Okay. Okay.
**Catherine Martin:** Mr, if you let me go, I won't. I won't press charges. I promise. See, my mom is a real important woman. I guess you already know that.
**Jame Gumb:** Now it places the lotion in the basket.

*Catherine Martin is trapped in a hole.*

Figure 3: Example of a snippet from the generated screenplay for *The Silence of the Lambs* (1991). The left side shows the transcribed text, with the names inferred by our method. The right side shows the *visual captions*, along with the keyframe from which they were derived. The horizontal line shows the inferred scene break.

## 3.5 MOVIE SUMMARY GENERATION

We experimentally show that the generated screenplay can be used as a basis for movide summarisation. We adopt a hierarchical summarisation approach (Pang et al., 2023; Chang et al., 2023), as it has been shown to be particularly suited to long inputs that exceed the context window size of large language models, and in our case can leverage the organization of the content into scenes. We this first summarise the transcript dialogue of each scene, and then fuse the resulting sequence of summaries, along with the visual information for each scene into a single summary for the entire movie (see Figure 1). Our summariser is implemented using a widely-used open-source LLM library (Dubey et al., 2024) with zero-shot prompting.

## 4 EXPERIMENTAL SETTING

**Implementation Details** Keyframes are extracted using FFMPEG's scene detect filter. The full command is given in Appendix B. Visual features are extracted from keyframes using CLIP (Radford et al., 2021). The precise model used in the experiments of Section 5 is 'CLIP-ViT-g-14-laion2B-s12B-b42K' from https://github.com/mlfoundations/open_clip. This speaker diarization model is WhisperX (Bain et al., 2023), an extension of Open AI's Whisper model which can perform speaker diarization and accurate utterance timestamping. For visual descriptions, we use Kosmos 2 (Peng et al., 2023), which has been pretrained on several multimodal corpora as well as grounded image-text pairs (spans from the text are associated with image regions) and instruction-tuned on various vision-language instruction datasets. Our summarisation model is built on top of Llama 3.1 70B (Touvron et al., 2023). We use short simple prompts for Llama and Kosmos, which are given in full in Appendix C. We instruct summaries to be a maximum of 635 words (the mean in our test set), and truncate summaries to 650 words if they are longer.

**Dataset** We take screenplays (for comparison models and some testing, see below) and gold summaries from the recently released MovieSum dataset (Saxena & Keller, 2024). For all 200 movies in the test set, we purchased the corresponding videos to use as input to our model. We were able to find videos for 175/200 test set instances. These movies span multiple fiction genres: drama, action, thriller, comedy, horror, etc. They have an average run time of 118min (range 84–228), with release dates ranging from 1950 to 2023. The gold summaries average 635 words in length. The mean

number of scenes in the gold script is 151. Because all stages of our method are zero-shot, we do not need video inputs for the training set.

**Evaluation Metrics**   Automated evaluation metrics are crucial for our task and for related long-form applications where human evaluation is extremely labor-intensive, costly, and difficult to design Krishna et al. (2023). As there is no single agreed-upon metric for automatically evaluating summarisation, we report several complementary metrics aimed at assessing different aspects of summary quality. **Rouge** (Lin, 2004) assesses informativeness against the reference summaries; PRISMA (Mahon & Lapata, 2024) measures factual precision and recall with respect to the gold summary; we use GPT4-turbo for both fact the extraction and evaluation stages; **SummaC** (Laban et al., 2022) uses NLI to measure consistency between the input document (gold screenplay) and generated summary; we use the SummaCConv version with 50 evenly-spaced bins; **AlignScore** (Zha et al., 2023) scores the 'informational alignment' between the source (gold screenplay) and the generated summary; we use the base-model checkpoint provided by the authors, and the recommended 'nli' setting with sentence chunk splitting. For both Alignscore and PRISMA we score duplicated information as incorrect, to penalize LLM outputs that repeat the same sentences over and over. To measure the accuracy of our scene detection method, we use standard partition quality metrics: cluster accuracy, adjusted Rand index and normalized mutual information, as defined in Mahon & Lukasiewicz (2024).

## 5   RESULTS

**Scene Detection**   To measure the accuracy of our scene segmentation method in isolation, we compare the partitions it produces to that arising from the ground truth scene breaks given in the gold screenplay. We perform dynamic time warping (Myers & Rabiner, 1981) on the dialogue lines in the gold screenplay and the timestamped utterances from the automatic transcript, in order to produce timestamps for the ground truth scene breaks. A naive metric would be the distance between the $n$th predicted break and the $n$th ground truth break, but this is inappropriate because a model that failed to predict the very first break, but got every other one exactly right, would then get a low score. Instead we treat scene break detection as a partition problem. Specifically, we consider the video as divided into 0.1s segments, where two segments are in the same element of the partition if and only if there is no scene break between them.

Table 1: Cluster accuracy 'acc', adjusted Rand index ('ari') and normalized mutual information ('nmi') of our predicted scene breaks, compared to SceneDetect and Yeung & Yeo (1996) as well as uniformly into 60, 75 and 90 and the true number of scenes ('uniform oracle').

|            | acc   | ari   | nmi   |
|------------|-------|-------|-------|
| unif-60    | 0.461 | 0.278 | 0.720 |
| unif-75    | 0.441 | 0.244 | 0.712 |
| unif-90    | 0.430 | 0.216 | 0.704 |
| unif-oracle| 0.436 | 0.228 | 0.710 |
| yeung96    | 0.321 | 0.220 | 0.541 |
| psd        | 0.394 | 0.061 | 0.596 |
| ours       | **0.564** | **0.375** | **0.746** |

Table 1 shows the accuracy of the predicted scene breaks, using the standard clustering metrics as defined above. As comparison models, we use the popular SceneDetect library[4] based on pixel discontinuities, the method of Yeung & Yeo (1996), which iteratively merges nearby similar shots into scenes, as well as various versions of splitting the input into uniformly-sized scenes: splitting into 60, 75 and 90 scenes (unif-60, unif-75 and unif-90, respectively), and splitting into the ground truth number of scenes for each movie. Note that our method makes no restrictions on the possible number of scenes, it is free to predict only one scene, or as many scene as there are 0.1s segments (~60,000). In practice, it predicts somewhere between 25 and 107 scenes across the test set. We observe that our minimum description length inspired algorithm is superior to baselines based on uniform segmentation, even when the number of scenes is known in advance (see row unif-oracle). Many occasions where the model fails to predict a scene boundary occur when the scenes on either side appear visually similar. For example, in *The Witch (2015)*, many scenes take place with the same background and characters, which presents only minor visual differences for the algorithm to detect. This suggests that many of the errors in our scene detection arise from insufficient signal in the visual feature vectors, rather than from the algorithm itself, and that future work which augments these vectors with, e.g., elements from dialogue or audio, would improve accuracy.

---

[4] https://www.scenedetect.com/

**Name Assignment** Table 2 presents evaluation of our name assignment algorithm against two baselines which assign names randomly and assign all IDs the most common name, i.e., the main character. As can be seen, though there is room for improvement, our approach is more accurate by a wide margin. Multiple factors contribute to the errors in name assignment: some incorrect faces being retrieved from the database (though this is low due to our clique-based filtering procedure), inaccuracies in the face feature vectors, such that the same person can sometimes receive dissimilar vectors in different contexts while different people can receive sometimes similar vectors, and the speaker diarization performed by WhisperX, which sometimes gives the same character a different speaker ID, or gives the same speaker ID to two different characters. This last error is especially problematic because it makes it impossible for the assignment algorithm to find a solution with zero mistakes. We expect that

Table 2: Accuracy of our assigned character names assigned compared to assigning names randomly ('random') and assigning the most common name, i.e., the main character, to all lines. Scores are averaged both across all movies ('acc movie-wise') and across all script lines in all movies ('acc line-wise').

|  | ours | most common | random |
|---|---|---|---|
| acc movie-wise | 61.12 | 19.35 | 2.97 |
| acc line-wise | 65.72 | 19.62 | 2.61 |

future improvements in speaker diarization and face verification will reduce the prevalence of these errors. Indeed, this is one of the advantages of a modular framework: improvements in specific areas can be incorporated into the framework without needing to change the other modules.

**Summarisation** In Table 3, we evaluate the summaries generated by our method. We benchmark against three baselines: 'name-only prompt' uses the parametric knowledge of the LLM without any content input, e.g., the prompt is 'Summarize the movie *The Silence of the Lambs*'[5]; 'full script' uses the entire gold screenplay as input in the prompt, and for 'whisperX' the input is the WhisperX transcript. We also compare to two existing models: Otter (Li et al., 2023), an end-to-end video description model based on video-llama2; and the modular model of Mahon & Lapata (2024) which takes videos and gold screenplays as input (described in Section 2). For Otter, we divide the input video into 3min chunks, and combine the model description of each chunk. We compare only to open-source models. We do not know the training setup or internals of closed-source proprietary models, so we would argue there is little to be learnt by comparison with them.

Our summaries obtain highest scores, across all metrics. The improvement is largest for the fact-based metrics of PRISMA (comprised of fact-prec and fact-rec), and Alignscore. The existing models, Otter and multimodal modular, especially struggle with such metrics. We find that Otter is mostly able to capture surface-level detail, with descriptions such as "a woman gets out of a car and goes into a building", but is unable to construct a narrative such as "a woman drives to the bank to deposit the money", so ends up capturing very little of the plot. The low scores of multimodal modular, on the other hand, are largely due to the older, smaller backbone model (BART, Lewis et al. (2020)), which often becomes decoupled from the input and produces unrelated output, highlighting the importance of incorporating current LLMs into video summarisation models. Giving only the movie name in the prompt produces reasonably high-quality summaries, confirming that Llama3.1 has significant information about these movies stored parametrically. However, these summaries are short, and when asked for a longer summary, the model repeats the same information over and over. Surprisingly, giving the full gold screenplay as input does not produce better summaries than our method or than some other baselines. This shows there is still difficulty in summarising very long text inputs. It appears that when prompted with the name only, the Llama-3.1 very likely effectively regurgitates an existing online summary. When given a very long prompt composed of the transcript or screenplay, it tries to actually summarise the information given, during which it can make mistakes. We provide example summary output in Appendix D.

Table 4 shows the results of removing the main components of our model. In 'w/o names', we omit replacing speaker IDs with character names. This causes summary quality to drop, which shows that not only is our name assignment more accurate than baseline methods (see Table 2), but it is sufficiently accurate to lead to improved downstream summaries. In 'w/o scene breaks', we feed the entire ScreenWriter input to Llama 3.1, instead of our hierarhcical approach of first summarising

---

[5]Precise prompts are given in Appendix C.

Table 3: Summarisation results on MovieSum. Best results are **in bold**, second best are *italicised.*.

|  | r1 | r2 | rl-sum | fact-prec | fact-rec | PRISMA | alignscore | summac |
|---|---|---|---|---|---|---|---|---|
| name-only prompt | *43.46* | *9.53* | *41.17* | *50.40* | 43.04 | 44.16 | 53.11 | 26.57 |
| full script | 42.39 | 9.32 | 39.94 | 48.77 | 52.73 | 49.05 | *68.59* | 25.83 |
| whisperX | 42.37 | 9.22 | 39.94 | 46.73 | *53.65* | 48.00 | 68.57 | 25.86 |
| Otter | 27.93 | 3.06 | 26.73 | 11.67 | 8.95 | 5.18 | 45.90 | 24.37 |
| multi-modular | 20.59 | 2.79 | 19.97 | 23.16 | 23.19 | 19.28 | 46.32 | *26.97* |
| ours | **46.48** | **10.32** | **44.50** | **55.24** | **54.77** | **53.57** | **72.76** | **27.24** |

Table 4: Ablation studies on summarisation results on MovieSum. Setting 'w/o names' does *not* replace speaker IDs with character names using our assignment method. Setting 'w/o scene breaks' summarises the entire screenplay in one pass, rather than splitting it into scenes using our algorithm and summarising each separately.

|  | r1 | r2 | rl-sum | fact-prec | fact-rec | PRISMA | alignscore | summac |
|---|---|---|---|---|---|---|---|---|
| w/o names | 45.46 | 10.43 | 43.40 | 49.93 | 53.64 | 49.00 | 63.67 | 26.45 |
| w/o scene breaks | 38.87 | 8.45 | 36.82 | 48.32 | 51.79 | 48.11 | 71.95 | 26.31 |
| unif-breaks | 38.87 | 8.45 | 36.82 | 46.58 | 50.69 | 48.11 | 57.62 | 25.73 |
| ours | 46.48 | 10.32 | 44.50 | 55.24 | 54.77 | 53.57 | 72.76 | 27.24 |

scenes and then fusing these to a final summary. The drop in summary performance in this setting shows the effectiveness of the hierarchical summarisation method enabled by the scene breaks in ScreenWriter. In 'unif-breaks', we still adopt the hierarchical summarisation method, but instead of using our scene breaks, split scenes into uniform chunks of length 250 tokens, which is the mean scene length from our predicted segmentation. This setting also reduces summary quality, which shows that not only is our scene segmentation more accurate than baseline methods (Table 1), but it is sufficiently accurate to lead to improved downstream summaries.

# 6 CONCLUSION

In this work, we proposed the task of generating automatic screenplays for movies from only video and audio input. Our model, Screenwriter, produces screenplays automatically (including dialogue, speaker names, scene breaks and visual descriptions) based on two novel algorithms: one for segmenting the video into scenes, based on the minimum description length principle and dynamic-programing for search, and one for assigning character names to dialogue utterances using a database of names and actor faces. Experimental results show that the output of ScreenWriter together with a hierarchical summarisation method can be used to generate movie plot synopses from only video and audio input. To the best of our knowledge, this is the first attempt to address this task. In the future, we would like to extend ScreenWriter's capabilities to other types of long videos, including documentaries, current affaires television programmes, and sports games, and explore personalised summary generation.

# ETHICS STATEMENT

Copyright is a concern when working with movies. We respected this by purchasing all the movies used for testing.

# 7 REPRODUCIBILITY STATEMENT

We specify the novel algorithms in detail in Section 3. We list the specific models used for our method and for comparison models in Section 4. We specify prompts used in Appendix C. Addi-

tionally, we have included all the code for our methods and experimental results in the supplementary material.

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

---

**Algorithm 2** Character Name Assignment to Speaker IDs

---

1: **Input:** Transcript with speaker IDs, keyframes split into $n$ scenes, IMDB

2: **Obtain actor face images:**
3: $\mathcal{A} \leftarrow$ empty list
4: **for** each actor/character $A$ appearing on the IMDB page for the movie **do**
5:     scrape the set $A_f$ of all available images of $A$
6:     remove from $A_f$, all images without exactly one detected face, or with face-name gender mismatch
7:     form graph $G = (A_f, E)$, where $E = \{(a_1, a_2) \in A_f \times A_f | \text{isVerified}(a_1, a_2)\}$
8:     $A_f \leftarrow$ largest clique in $G$
9:     append $A_f$ to $\mathcal{A}$
10: **for** each scene $j = 1, \dots, n$ **do**
11:     Form $D_j$, the set of all faces across all keyframes of the scene

12: **Assign character names to scenes:**
13: $C_1 \leftarrow n \times m$ empty matrix, where $m$ is the length of $\mathcal{A}$
14: **for** $i = 1, \dots, m$ **do**
15:     $A_f \leftarrow \mathcal{A}[i]$
16:     **for** each scene $j = 1, \dots, n$ **do**
17:         $C_1[i, j] \leftarrow min_{a \in A_f, b \in D_j} d(a, b)$                  $\triangleright$ $d(\cdot)$ from Deepface vectors

18: **Assign character names to speaker IDs:**
19: $C_2 \leftarrow k \times m$ empty matrix, where $k$ is the number of unique speaker IDs
20: **for** $i = 1, \dots, m$ **do**
21:     **for** each speaker ID $l = 1, \dots k$ **do**
22:         $C_2[i, k] \leftarrow \frac{1}{n} \sum_{w=1}^{n} C_1[i, w]$
23: $B \leftarrow \frac{1}{mk} \sum_{i=1}^{m} \sum_{i=1}^{k} C_2[i, j]$
24: $C_2 \leftarrow C_2 \oplus C_2 \oplus C_2$           $\triangleright$ Concatenate three copies along first dimension
25: $LSAP \leftarrow \text{Kuhn-Munkres}(C_2)$    $\triangleright$ Linear Sum Assignment Problem: $k$-dim vector assigning cols to rows
26: **for** $i = 0, \dots 3k$ **do**
27:     $i' \leftarrow i \mod k$
28:     $j' \leftarrow LSAP[i]$
29:     **if** $C_2[i', j'] < B$ **then**
30:         assign speaker ID $i \mod k$ to name $LSAP[i]$

---

Luowei Zhou, Chenliang Xu, and Jason J Corso. Towards automatic learning of procedures from web instructional videos. In *AAAI Conference on Artificial Intelligence*, 2018a. URL `https://www.aaai.org/ocs/index.php/AAAI/AAAI18/paper/view/17344`.

Luowei Zhou, Yingbo Zhou, Jason J. Corso, Richard Socher, and Caiming Xiong. End-to-end dense video captioning with masked transformer. In *Proceedings of the IEEE Conference on Computer Vision and Pattern Recognition (CVPR)*, June 2018b.

## A NAME ASSIGNMENT ALGORITHM

Here we show a pseudo-code description of the algorithm discussed in Section 3.3 for assigning character names to speaker IDs.

## B FFMPEG COMMANDS

To select keyframes, we use

```
\usr\bin\ffmpeg -i {path-to-video} -filter:v -select='gt(scene,0.1)'
    -showinfo -vsync 0 %04d.jpg
```

This extracts all keyframes into files 0001.jpg, 0002.jpg, etc, in the current working directory.

## C  PROMPTS

### C.1  SCREENWRITER PROMPTS

Below we present the various prompts we employ for obtaining scene descriptions, and performing hierarchical summarisation. Note that Kosmos is a text completion model, so this prompt just serves as the first part of the sentence, which we then remove afterward.

---

**Kosmos 2 Descriptions**

A shot from a movie in which .

---

**Llama 3.1 70B: Dialogue summarisation**

Here is the dialogue from scene <scene-number> of the movie <movie-title>: <scene-dialogue-with-names>. Please describe its main events in bullet points. Don't include information from outside this scene. Do not answer in progressive aspect, i.e., don't use -ing verbs or "is being".

In this scene, here are a few main events:

---

**Llama 3.1 70B: Final summarisation**

Here is a sequence of summaries of each scene of a movie.
<concatenated-dialogue-summaries>

Combine them into a plot synopsis of no more than 635 words. Be sure to include information from all scenes, especially those at the end, don't focus too much on early scenes. Discuss only plot events, no analysis or discussion of themes and characters.

Based on the information provided, here is a plot synopsis of the move <movie-title>:

---

### C.2  SUMMARY PROMPTS FOR COMPARISON SYSTEMS

Below we show the prompts used to obtain movie summaries for the various baselines and comparison systems discussed in Section 5. The 'name-only prompt' uses the parametric knowledge of the LLM without any specific, content input. The 'full script' prompt uses the entire gold screenplay as input, and 'WhisperX' just the audio transcript without name assignment or scene breaks.

---

**Llama 3.1 70B: Name-Only**

Summarize the plot of the movie <movie-title> in about 650 words. Do not write the summary in progressive aspect, i.e., don't use -ing verbs or "is being". Focus only on the plot events, no analysis or discussion of themes and characters.

---

**Llama 3.1 70B: Full Script**

Based on the following script: <gold-screenplay> summarize the movie <movie-title>. Do not write the summary in progressive aspect, i.e., don't use -ing verbs or "is being". Focus only on the plot events, no analysis or discussion of themes and characters.

---

> **Llama 3.1 70B: WhisperX Transcript**
>
> Based on the following transcript: <whisper-transcript> summarize the movie <movie-title>. Do not write the summary in progressive aspect, i.e., don't use -ing verbs or "is bei ng". Focus only on the plot events, no analysis or discussion of themes and characters.

## D EXAMPLE SUMMARIES

In the following we show example summaries generated by our model and comparison systems for the movie *Oppenheimer* (2023). Incorrect or undesirable text is shown in red and repeated information is highlighted in gray. For comparison, we also include the gold summary from the MovieSum test set.

> **ScreenWriter Summary Output**
>
> The movie Oppenheimer begins with J. Robert Oppenheimer testifying before the Security Board, explaining that the derogatory information against him must be understood in the context of his life and work. Lewis Strauss and Gordon Gray discuss Strauss's upcoming Senate confirmation hearing for a cabinet position, and Gray advises Strauss to answer honestly about his past conflicts with Oppenheimer. The story then flashes back to Oppenheimer's early life, where he meets Niels Bohr and is introduced to the world of physics.
>
> Oppenheimer becomes involved with left-wing groups and is questioned about his communist associations. He meets with Lewis Strauss, who is trying to recruit him to run the Institute for Advanced Study at Princeton. As the story progresses, Oppenheimer becomes involved in the development of the atomic bomb and is appointed as the director of the Manhattan Engineer District. He meets with Colonel Groves and Lieutenant Colonel Nichols, who express concerns about his suitability for the job due to his suspected communist sympathies and unstable personality. Despite these concerns, Oppenheimer convinces the team to work on the project, and they begin to develop the atomic bomb.
>
> The team faces numerous challenges, including the need for vast resources and the risk of inefficiency and poor security. Oppenheimer mentions that the Germans, led by Werner Heisenberg, have a 12- to 18-month head start on developing an atomic bomb. As the project progresses, Oppenheimer becomes increasingly concerned about the implications of the atomic bomb and the potential for an arms race with the Soviets. He expresses his concerns to Lewis Strauss, who is skeptical of Oppenheimer's views. The team eventually succeeds in developing the atomic bomb, and it is dropped on Hiroshima. The bomb's power is described as equivalent to 20,000 tons of DNC, and General Groves is congratulated on the success of the mission.

However, Oppenheimer's involvement in the project and his left-wing past eventually catch up with him. Lewis Strauss, who has a history of conflict with scientists, particularly Oppenheimer, begins to systematically destroy Oppenheimer's credibility by denying him his Q clearance through a bureaucratic procedure. Oppenheimer is given the chance to appeal, and a board is appointed to hear the case, with Roger Robb as the prosecutor. The hearing becomes a trial, with numerous witnesses testifying against Oppenheimer. Lewis Strauss questions Oppenheimer about his involvement in the development of the hydrogen bomb and his moral qualms about it. The Gray Board eventually announces its decision to deny J. Robert Oppenheimer's security clearance renewal. Lewis Strauss is informed that the decision was not unanimous and that there were unexpected holdouts, including Senator John F. Kennedy. Strauss reacts angrily to the news, feeling that Oppenheimer has turned the scientists against him and that he is being unfairly targeted.

The movie ends with Oppenheimer calm and collected in the face of the board's decision, while Strauss is still fuming and Gordon Gray tries to reason with him. The final scenes show Oppenheimer addressing a group of people, and a truck driving through a desert town, symbolizing Oppenheimer's continued involvement in the world of physics despite the challenges he faced. Overall, the movie Oppenheimer is a story about the development of the atomic bomb and the challenges faced by J. Robert Oppenheimer, a brilliant physicist who was involved in the project. The movie explores the implications of the atomic bomb and the potential for an arms race with the Soviets, as well as the personal costs faced by Oppenheimer due to his left-wing past and his involvement in the project. The movie ends with Oppenheimer calm and collected, despite the challenges he faced, and symbolizes his continued involvement in the world of physics.

### Name-only Prompt Summary Output

The movie Oppenheimer (2023) tells the story of J. Robert Oppenheimer, the director of the Manhattan Project, which developed the atomic bomb during World War II. The film begins with Oppenheimer's appointment as the director of the project in 1942. He assembles a team of scientists, including Enrico Fermi, Ernest Lawrence, and Richard Feynman, to work on the top-secret project at Los Alamos, New Mexico.

Oppenheimer's team faces numerous challenges, including the difficulty of creating a nuclear reaction and the need to develop a delivery system for the bomb. The team works tirelessly to overcome these obstacles, and they eventually succeed in creating a functional atomic bomb.

As the project nears completion, Oppenheimer and his team are faced with the moral implications of their creation. They are aware that the bomb will cause widespread destruction and loss of life, and some team members express reservations about its use.

The film depicts the first nuclear test, codenamed "Trinity," which takes place on July 16, 1945, in the New Mexico desert. The test is a success, and the team is ecstatic. However, Oppenheimer is visibly shaken by the experience, quoting the Bhagavad Gita, "I am become Death, destroyer of worlds."

The film then jumps forward in time to 1954, when Oppenheimer is called to testify before the Atomic Energy Commission. He is questioned about his loyalty to the United States and his associations with communist sympathizers. Oppenheimer's past comes back to haunt him, and he is eventually stripped of his security clearance.

The film ends with Oppenheimer's reflection on the devastating consequences of the atomic bomb and his role in its creation. He is shown to be a complex and troubled individual, grappling with the moral implications of his work.

Note: The above text is a summary of the movie Oppenheimer (2023) and not the actual movie script. Here is a rewritten version of the summary in the past tense:

The movie Oppenheimer (2023) tells the story of J. Robert Oppenheimer, the director of the Manhattan Project, which developed the atomic bomb during World War II. The film begins with Oppenheimer's appointment as the director of the project in 1942. He assembled a team of scientists, including Enrico Fermi, Ernest Lawrence, and Richard Feynman, to work on the top-secret project at Los Alamos, New Mexico.

Oppenheimer's team faced numerous challenges, including the difficulty of creating a nuclear reaction and the need to develop a delivery system for the bomb. The team worked tirelessly to overcome these obstacles, and they eventually succeeded in creating a functional atomic bomb.

As the project neared completion, Oppenheimer and his team were faced with the moral implications of their creation. They were aware that the bomb would cause widespread destruction and loss of life, and some team members expressed reservations about its use.

The film depicted the first nuclear test, codenamed "Trinity," which took place on July 16, 1945, in the New Mexico desert.

## WhisperX + Llama3.1 Summary Output

The movie Oppenheimer (2023) tells the story of J. Robert Oppenheimer, a brilliant physicist who played a crucial role in the development of the atomic bomb during World War II. The film begins with Oppenheimer's security clearance hearing, where he is questioned about his past associations with communists and his opposition to the development of the hydrogen bomb.

The story then flashes back to Oppenheimer's early days as a professor at the University of California, Berkeley, where he meets his future wife, Kitty, and becomes friends with other prominent physicists, including Enrico Fermi and Ernest Lawrence.

As World War II breaks out, Oppenheimer is recruited to lead the Manhattan Project, a secret research and development project that produces the atomic bomb. He works closely with other scientists, including Edward Teller and Leslie Groves, to develop the bomb, which is eventually dropped on Hiroshima and Nagasaki, killing hundreds of thousands of people.

After the war, Oppenheimer becomes a vocal critic of the development of the hydrogen bomb, which he believes will lead to a nuclear arms race and increase the risk of nuclear war. This puts him at odds with Teller and other scientists who support the development of the hydrogen bomb.

The film then cuts back to the security clearance hearing, where Oppenheimer is questioned about his past associations and his opposition to the hydrogen bomb. He is ultimately denied his security clearance, which effectively ends his career as a government advisor.

The film ends with Oppenheimer attending a ceremony at the White House, where he is awarded a medal for his contributions to science. However, the ceremony is bittersweet, as Oppenheimer is aware that his work on the atomic bomb has had devastating consequences, and that he has been punished for speaking out against the development of the hydrogen bomb. The film ends with a sense of irony and tragedy, as Oppenheimer is honored for his contributions to science, while also being aware of the devastating consequences of his work. The final shot is of Oppenheimer visualizing the expanding nuclear arsenals of the world, with the sound of feet stamping in the background, symbolizing the impending doom of the world due to nuclear war. The credits roll as the sound of the feet stamping continues to build up, leaving the audience with a haunting sense of the devastating consequences of Oppenheimer's work. The final shot is of Oppenheimer visualizing the expanding nuclear arsenals of the world, with the sound of feet stamping in the background, symbolizing the impending doom of the world due to nuclear war. The credits roll as the sound of the feet stamping continues to build up, leaving the audience with a haunting sense of the devastating consequences of Oppenheimer's work. The final shot is of Oppenheimer visualizing the expanding nuclear arsenals of the world, with the sound of feet stamping in the background, symbolizing the impending doom of the world due to nuclear war. The credits roll as the sound of the feet stamping continues to build up, leaving the audience with a haunting sense of the devastating consequences of Oppenheimer's work. The final shot is of Oppenheimer visualizing the expanding nuclear of the world.



**Gold Summary from the MovieSum Test Set**

In 1926, 22-year-old doctoral student J. Robert Oppenheimer grapples with anxiety and homesickness while studying under experimental physicist Patrick Blackett at the Cavendish Laboratory in the University of Cambridge. Upset with Blackett's attitude, Oppenheimer leaves him a poisoned apple but later retrieves it. Visiting scientist Niels Bohr advises Oppenheimer to study theoretical physics at the University of Göttingen instead. Oppenheimer completes his PhD there and meets fellow scientist Isidor Isaac Rabi. They later meet theoretical physicist Werner Heisenberg in Switzerland.

Wanting to expand quantum physics research in the United States, Oppenheimer begins teaching at the University of California, Berkeley and the California Institute of Technology. He marries Katherine "Kitty" Puening, a biologist and ex-communist, and has an intermittent affair with Jean Tatlock, a troubled communist who later commits suicide.

In December 1938, nuclear fission is discovered, which Oppenheimer realizes could be weaponized. In 1942, during World War II, U.S. Army Colonel Leslie Groves recruits Oppenheimer as director of the Manhattan Project to develop an atomic bomb. Oppenheimer, who is Jewish, is mainly concerned that the German nuclear research program, led by Heisenberg, might yield a fission bomb for the Nazis. He assembles a team consisting of Rabi, Hans Bethe and Edward Teller at the Los Alamos Laboratory, and also collaborating with scientists Enrico Fermi, Leo Szilard and David L. Hill at the University of Chicago. Teller's calculations reveal an atomic detonation could trigger a catastrophic chain reaction that ignites the atmosphere. After consulting with Albert Einstein, Oppenheimer concludes the chances are acceptably low. Teller attempts to leave the project after his proposal to construct a hydrogen bomb is rejected, but Oppenheimer convinces him to stay.

After Germany's surrender in 1945, some Project scientists question the bomb's relevance; Oppenheimer believes it would end the ongoing Pacific War and save Allied lives. The Trinity test is successful, and President Harry S. Truman orders the atomic bombings of Hiroshima and Nagasaki, resulting in Japan's surrender. Though publicly praised, Oppenheimer is haunted by the mass destruction and fatalities. After expressing his personal guilt to Truman, the president berates Oppenheimer and dismisses his urging to cease further atomic development.

As an advisor to the United States Atomic Energy Commission (AEC), Oppenheimer's stance generates controversy, while Teller's hydrogen bomb receives renewed interest amidst the burgeoning Cold War. AEC Chairman Lewis Strauss resents Oppenheimer for publicly dismissing his concerns about exporting radioisotopes and for recommending negotiations with the Soviet Union after they successfully detonated their own bomb. He also believes that Oppenheimer denigrated him during a conversation Oppenheimer had with Einstein in 1947. In 1954, wanting to eliminate Oppenheimer's political influence, Strauss secretly orchestrates a private security hearing before a Personnel Security Board concerning Oppenheimer's Q clearance.

However, it becomes clear that the hearing has a predetermined outcome. Oppenheimer's past communist ties are exploited, and Groves' and other associates' testimony is twisted against him. Teller testifies that he lacks confidence in Oppenheimer and recommends revocation. The board revokes Oppenheimer's clearance, damaging his public image and limiting his influence on nuclear policy. In 1959, during Strauss' Senate confirmation hearing for Secretary of Commerce, Hill testifies about Strauss' personal motives in engineering Oppenheimer's downfall, resulting his nomination being voted down.

In 1963, President Lyndon B. Johnson presents Oppenheimer with the Enrico Fermi Award as a gesture of political rehabilitation. A flashback reveals Oppenheimer and Einstein's 1947 conversation never mentioned Strauss. Oppenheimer instead expressed his belief that they had indeed started a chain reaction—a nuclear arms race—that would one day destroy the world.



## E    EXAMPLE GENERATED SCREENPLAYS

Generated Screenplay for *Moonstruck (1987)*. **SPEAKER 31**: When the moon hits your eye like a big pizza pie, that's a morning.

**SPEAKER 05**: When the world seems to shine like you've had too much wine, that's amore. Bells will ring, ting-a-ling-a-ling, ting-a-ling-a-ling, and you'll sing the tabella. Hearts will play tippy-tippy-tay, tippy-tippy-tay, like a guitar andella. When the stars make you drool, just like a pasty fuzzle, that's amore. When you dance down the street with a cloud at your feet, you're in love.

When you walk in a dream, but you know you're not dreaming, senorita. Excuse me if I just say back in old Napoli, that's amore. Lucky fella. When the stars make it through, just like plaster falls, that's amore. That's amore. When you dance down the street with the cloud at your feet, you're in love. When you hop in a dream, But you know, you're not dreaming, signore. Scuse me, but you see back in old Napoli, that's amore.

**Unknown Speaker**: He looks great.

**[SCENE-BREAK]**

**SPEAKER 24**: That Alconte is a genius.

**SPEAKER 29**: I am a genius. I am a genius.Loretta Castorini If you're such a genius, how come you can't keep track of your receipts? Al, how am I supposed to do your income tax with this mess you got here?

**Cosmo Castorini**: Numbers, taxes, receipts. I make them look better than they did in real life. I'm an artistic genius.

**Loretta Castorini**: Then how come you got butter on your tie? Give it here. Give it here. You know what? I'll give you this. You make good coffee. You're a slob, but you make good coffee.

**Cosmo Castorini**: Red roses. Very romantic. The guy that sends these really knows what he's doing.

**Loretta Castorini**: The guy who sends those spends a lot of money on something that could end up in the garbage.

**Cosmo Castorini**: I'm glad everybody ain't like you, Loretta. I'd be out of business.

**Loretta Castorini**: What are you talking? I love flowers.

**SPEAKER 08**: Thanks, Carmine.

**SPEAKER 21**: You ready?

**SPEAKER 18**: Oh, hello, Bobo. How are you tonight? Very good, Mr. Gianni. Uh, we will both have the, uh, tachino salad. And, uh, I'll have special fish.

**[SCENE-BREAK]**

**Loretta Castorini**: No, you don't want the fish. No? It's oily fish tonight, not before the plane ride.

**SPEAKER 18**: Well, maybe you're right.

**Loretta Castorini**: Okay, Bobo, we'll have the maricote, Bobo.

**SPEAKER 21**: Si, Miss Lorena.

**Loretta Castorini**: That'll give you a base for your stomach. You know, you eat that oily fish, you go up in the air, halfway to Sicily, you'll be green and your hands will be sweating.

**SPEAKER 18**: You look after me.

**SPEAKER 06**: Now, Patricia, please, don't leave. What do you think I am, some sort of talking dog? I was just making a point about the way you said... the way you stated your aspirations. Kiss my aspirations, Professor. Kiss my aspirations? Oh, very clever. Yeah, the height of cleverness. Any evidence of her, I'd bring me a big glass of vodka. But absolutely.

**SPEAKER 18**: A man who can't control his woman is funny.

**SPEAKER 21**: She was too young for him. What's the matter, Okapobo? Tonight, Mr. Johnny's gonna propose marriage. How do you know that? He arranged it with me. What he has, he's gonna waive. And then I serve him the champagne. Mille grazie.

**SPEAKER 18**: Ah, how stinks, eh?

**Loretta Castorini**: Fine, Bobo. We'll have the chan.

**SPEAKER 18**: Oh, no, no. I, uh... I want to see the dessert cart. Very good.

**Loretta Castorini**: You never have dessert.

**SPEAKER 18**: Oh, never is a long time. Oh, my scalp is not getting enough blood sometimes. Have some dessert.

**Loretta Castorini**: No, I shouldn't.

**SPEAKER 18**: Will you marry me?

**Loretta Castorini**: What?

**SPEAKER 18**: Will you marry me?

**Loretta Castorini**: Bobo, take the card away. Very good, Miss Loretta. Are you proposing marriage to me?

**SPEAKER 07**: Yes.

**Loretta Castorini**: All right, you know I was married and that my husband died. But what you don't know is I think he and I had bad luck.

**SPEAKER 18**: What do you mean?

**Loretta Castorini**: Well, we got married down at the city hall, and I think it gave bad luck the whole marriage.

**SPEAKER 18**: I don't understand.

**Loretta Castorini**: Right from the start, we didn't do it right, OK? Could you kneel down?

**SPEAKER 18**: On the floor?

**Loretta Castorini**: Yeah, on the floor. This is a good suit. I know that. I helped you pick it out. It came with two pairs of pants. You know, Johnny, it's for luck. I mean, a man proposes marriage to a woman, he should kneel down.

**[SCENE-BREAK]**

**SPEAKER 06**: Is that man praying?

**Loretta Castorini**: So? Where's the ring?

**Raymond Cappomagi**: The ring. A ring, that's right. I would have sprung for the ring if it was me, cabish.

**Loretta Castorini**: You could use your pinky ring.

**SPEAKER 18**: I like this ring.

**Loretta Castorini**: You propose marriage to a woman, you should offer her a ring of engagement.

**SPEAKER 18**: Loretta, Loretta Castorini Clark, on my knees in front of all of these people, will you marry me?

**Loretta Castorini**: Yes, Johnny. Yes, John Anthony Camareri, I will marry you. I will be your wife.

**SPEAKER 20**: Bobo, the check! By the way, Mr. Jones... What about the wedding?

**[SCENE-BREAK]**

**SPEAKER 18**: My mother is dying. When she's dead, I'll come back and we'll get married.

**Loretta Castorini**: When you're dead, is she?

**SPEAKER 18**: A week, two weeks, no more.

**Loretta Castorini**: How about we set a date? How about a month?

**SPEAKER 18**: Must be so definite. Can't we just say we'll be married when I get back?

**Loretta Castorini**: Where, at the city hall? No.

**SPEAKER 18**: My mother is dying.

**Loretta Castorini**: You know, I want a whole wedding, or we'll have bad luck. And for a whole wedding to be planned, you got to set a date.

**SPEAKER 18**: All right. All right. All right. A month.

**Loretta Castorini**: A month from today.

**SPEAKER 18**: In a month.

**Loretta Castorini**: A month from today. Yes, a month from today. OK. From today, right. OK, OK, OK. All right. I'll take care of it, Johnny. I'll take care of the whole thing. All you got to do is show up.

**[SCENE-BREAK]**

**SPEAKER 07**: 700 at B point 24 Paris. Please proceed to gate number 29.

**Loretta Castorini**: Call me when you get in.

**SPEAKER 18**: I'll call you when I get to Mama's house. OK.

**Loretta Castorini**: You made me very happy.

**SPEAKER 18**: There's one thing about this wedding I want you to do. I want you to call, uh, I want you to call his number. It's a business number. You ask for Ronnie. Invite him to the wedding. OK.

**Loretta Castorini**: Who is it?

**SPEAKER 18**: It's my younger brother.

**Loretta Castorini**: You got a brother?

**SPEAKER 18**: We haven't spoken in five years. There was some bad blood. I want you to call him and invite him to the wedding. Will you do it?

**SPEAKER 06**: Sure. I've got to go.

**Loretta Castorini**: You got your ticket? OK, here. I'll get you these gum and Thor Flops.

**SPEAKER 18**: Gum opens your ears when you chew it. I've got to go.

**SPEAKER 24**: You have someone on that plane? Yeah, my fiancée. I put a curse on that plane. My sister is on that plane. I put a curse on that plane that it's gonna explode by now and fall into the sea. Fifty years ago, she stole a man from me. She tells me that she never loved him, that she took him to be strong on me. She's gone back to Sicily. I cursed her that the Green Atlantic water, she'll swallow her up.

**Loretta Castorini**: I don't believe in curses. Neither do I.

**[SCENE-BREAK]**

**Cosmo Castorini**: Johnny's, right? Yeah.

**Unknown Speaker**: Good night.

**Loretta Castorini**: Good night.

**SPEAKER 24**: What are you talking about?

**Cosmo Castorini**: I've seen the way you look at her, and it isn't right! How do I look at her? Can I help you? A spoon of mummy's. So, how do I look at her? Like a wolf. Like a wolf, huh? Uh-huh. Like a wolf. You've never seen a wolf in your life. It's $11.99.

**SPEAKER 24**: I've seen a wolf in everybody I ever met, and I see a wolf in you.

**Cosmo Castorini**: That makes 20. Thanks. Have a nice night. You too. You know what I see in you, Lottie? What? The girl I married. Oh, come on. Good night. Good night. Good night.

**[SCENE-BREAK]**

**SPEAKER 31**: How are you?

**Loretta Castorini**: Guess what happened to me today? Joe Bello.

**SPEAKER 04**: Hey, how long must I wait? Come here.

**Unknown Speaker**: Hi, Pop.

**SPEAKER 04**: Oh, hi. Where's Ma? Dead. Not sleeping? I can't sleep anymore. It's too much like death. Pop, I got news. All right.

**Unknown Speaker**: I start to pray.

**Unknown Speaker**: Let it be, let it be.

**[SCENE-BREAK]**

**SPEAKER 08**: You look tired, Pop.

**SPEAKER 16**: So what's your news?

**Loretta Castorini**: I'm getting married.

**SPEAKER 16**: Again?

**Loretta Castorini**: Yeah.

**SPEAKER 16**: You did this month before it didn't work out. The guy died. And what killed him?

**Loretta Castorini**: He got hit by a bus.

**SPEAKER 16**: No, bad luck. Your mother and I were married 52 years. Nobody died. You were married, what, two years? Somebody's dead. Don't get married again, Loretta. It don't work out for you. Who's the man?

**Loretta Castorini**: Johnny Camareri.

**SPEAKER 16**: Johnny Camareri? He's a big baby. And why's he in the air when you're telling me this?

**Loretta Castorini**: Because he's flying to Sicily. His mother is dying.

**SPEAKER 16**: More bad luck. You like his face, Loretta? I don't like his lips. When he smiles, I can't see his teeth. Where does he hide? When are you going to do it?

**Loretta Castorini**: In a month.

**SPEAKER 16**: I won't come.

**Loretta Castorini**: You've got to come. You've got to give me away.

**SPEAKER 16**: I didn't give you away the first time.

**Loretta Castorini**: And I had bad luck. You know, maybe if you gave me away and I got married in a church in a wedding dress instead of down at the city hall with strangers standing outside the door, then maybe I wouldn't have had the bad luck I had.

**Unknown Speaker**: Maybe.

**Loretta Castorini**: You know, Pop, I had no reception, no wedding cake, no nothing. Johnny got down on his knees and proposed to me at the Grand Ticino.

**SPEAKER 16**: He did?

**Loretta Castorini**: Yeah.

**SPEAKER 16**: That don't sound like Johnny.

**Old Man**: Well.

**SPEAKER 16**: Where's the ring?

**Old Man**: Here. It looks stupid. It's a pinky ring. It's a baby's. It's a man's ring. It's temporary. Everything is temporary. That don't excuse nothing. You're coming?

**[SCENE-BREAK]**

**SPEAKER 16**: I was going to tell you. Rose. Rose. Rose. Rose.

**SPEAKER 11**: Who's dead?

**SPEAKER 16**: Nobody. Loretta's getting married. Again? Yeah.

**SPEAKER 13**: Johnny Camareri?

**SPEAKER 16**: I don't like him.

**SPEAKER 13**: You're not gonna marry him, Cosmo. Do you love him, Loretta?

**Loretta Castorini**: No.

**SPEAKER 13**: Good. When you love him, they drive you crazy, because they know they can. But you like him?

**Loretta Castorini**: Oh, yeah. You know, he's a sweet man, Ma. And this time, I'm going to get married in a church and have a big reception.

**SPEAKER 13**: And who's going to pay for that? Pop. What? Father or the bride pays. I have no money. You're rich as Roosevelt. You're just cheap, Cosmo.

**Loretta Castorini**: I won't pay for nothing. You know, it's your duty as my father to pay for the wedding. I won't pay for nothing.

**SPEAKER 11**: He didn't used to be cheap. He thinks if he holds on to his money, he will never die.

**SPEAKER 13**: Now he's gonna play that damn Vicky Carr record, and when he comes to bed, he won't touch me.

**SPEAKER 07**: Hmm. Got it.

**SPEAKER 13**: Yeah.

**SPEAKER 07**: Yeah. Yeah. And you...

**SPEAKER 31**: Hey, keep the dogs out of my life. Come on.

**SPEAKER 04**: Don't get up.

**Unknown Speaker**: Come on.

**[SCENE-BREAK]**

**SPEAKER 04**: Don't do that. He's asleep. Look, all the flowers on Fonsoga. My brother sent a blue flower. I can only see things in my house, and things in my house are very bad. I don't know what to advise my son. I think he should pay for the wedding, but it's important that he don't look ridiculous. Well, why don't you talk to him? I will. They must find the right moon. See la bella luna last night? Ah, si. Ah, si la luna, la bella luna. The moon brings the woman to the man. Capisce? La luna tira l'uomo, la donna verso l'uomo. Avete capito? Andiamo. Vieni, vieni. Non tirare, non tirare. I'll show you. That's it.

**Unknown Speaker**: Scoundrels.

**[SCENE-BREAK]**

**SPEAKER 11**: Upstairs. Everybody upstairs.

**SPEAKER 04**: Come on. Ciao, Ben.

**SPEAKER 13**: So, will you live here? No. Why not? Pop don't like Johnny. He'll sell the house. I got married before you didn't sell. Grandma was still alive. Cheera was still home going to school. Now he's married, gone to Florida. You and Johnny moved in, had a baby. Mom, I'm 37 years old. What's 37? I didn't have Cheeta till after I was 37. It ain't over till it's over.

**Loretta Castorini**: Johnny's got a big apartment. You know we'll move in there. So we'll sell the house. You know what? I want to live here. I love the house. Pop don't like Johnny.

**SPEAKER 13**: No, we don't.

**Loretta Castorini**: Oh. I'll get it. I'll get it. Hello? Yeah, yeah, this is Loretta Castorini. Johnny. Johnny.

**[SCENE-BREAK]**

**Loretta Castorini**: Operator?

**SPEAKER 18**: Operator? Loretta?

**Loretta Castorini**: Yeah.

**SPEAKER 18**: Yes. Shh, shh, shh, shh, shh.

**SPEAKER 07**: It's me.

**SPEAKER 18**: I'm calling from the deathbed of my mother.

**Loretta Castorini**: Yeah, well, how was your plane ride?

**SPEAKER 18**: The waitresses were very nice. My mother is slipping away.

**SPEAKER 11**: I can't talk long.

**Loretta Castorini**: Did you tell her we're getting married?

**SPEAKER 18**: Oh, no, no, not yet. I'm waiting. I'm waiting until a moment when she's peaceful.

**Loretta Castorini**: Well, don't wait till she's dead.

**SPEAKER 18**: Have you called my brother?

**Loretta Castorini**: No, I'm sorry. No, not yet. I forgot.

**SPEAKER 18**: Will you do it today?

**Loretta Castorini**: Yes. Yes.

**SPEAKER 18**: Make sure he comes to the wedding. Five years is too long for bad blood between brothers. Nothing can replace the family, Loretta. I can see that now.

**SPEAKER 20**: Loretta? Loretta, you there? Loretta!

**Loretta Castorini**: OK, I'm sorry. I'll do it. I'll do it today. And listen, Johnny, you know, call me after you tell her, OK?

**SPEAKER 18**: All right. All right.

**Loretta Castorini**: And, uh, don't stand directly under the sun. You got a hat? Use your hat.

**SPEAKER 18**: I got my hat. All right. All right.

**Unknown Speaker**: All right.

**SPEAKER 18**: Bye.

**Loretta Castorini**: All right. Bye-bye. Now, what did I do with that card? How's her mother? She's dying, but I can still hear her big mouth. He didn't tell her. You know, the woman makes him crazy. Now who are you calling?

**SPEAKER 29**: Oh, fuck. Camerares.

**Loretta Castorini**: Yeah, is Ronnie Camareri there, please? Hold on.

**SPEAKER 25**: Ronnie, the phone! Ronnie?

**Loretta Castorini**: Yeah, this is Ronnie.

**Loretta Castorini**: Yeah, I'm calling for your brother. And he's getting married, and he would like it if you would come.

**Loretta Castorini**: Why didn't he call himself? He's inflammable.

**Loretta Castorini**: What's wrong can never be made right. Look, let me just come and talk to you. Talk to you? Animal! What an animal!

**[SCENE-BREAK]**

**SPEAKER 25**: Thanks, Mrs. Bugacci. Here you go. OK, bye. See you tomorrow. OK, bye.

**Loretta Castorini**: Is Ronnie Camareri here?

**SPEAKER 25**: He's down at the ovens. What do you want?

**Loretta Castorini**: I want to talk to him.

**SPEAKER 25**: This way.

**[SCENE-BREAK]**

**Raymond Cappomagi**: Johnny! Johnny, someone wants to see you.

**Loretta Castorini**: Did you come for my brother, Johnny?

**Loretta Castorini**: Yeah.

**Loretta Castorini**: Why?

**Loretta Castorini**: Uh, well, we're going to get married.

**Loretta Castorini**: You're going to marry my brother Johnny?

**Loretta Castorini**: Yeah, would you like to go someplace so we could talk?

**Loretta Castorini**: I have no life. Excuse me? I have no life. My brother Johnny took my life from me.

**Loretta Castorini**: I don't understand you.

**Loretta Castorini**: And now he's getting married. He has his. He's getting his. And he wants me to come. What is life?

**Loretta Castorini**: You know, I didn't come here to upset you.

**Loretta Castorini**: They say bread is life. And I bake bread, bread, bread. And I sweat. And shovel this stinking dough in and out of this hot hole in the wall. And I should be so happy. Huh, sweetie? You want me to come to the wedding of my brother Johnny? Where's my wedding? Chrissy, over by the wall, bring me the big knife.

**[SCENE-BREAK]**

**Loretta Castorini**: No, Ronnie.

**Loretta Castorini**: Bring me the big knife. I'm going to cut my throat. Maybe I should come back another time. No, I want you to see this. I want you to watch me kill myself so you can tell my brother Johnny on his wedding day, OK? Chrissy, bring me the big knife. Do you know about me?

**SPEAKER 08**: Oh, Mr. Camarero.

**Loretta Castorini**: What? Do you know about me? Mm-mm.

**Unknown Speaker**: OK.

**Loretta Castorini**: Nothing is anybody's fault, but things happen.

**Raymond Cappomagi**: Look.

**Loretta Castorini**: This wood is fake. Five years ago, I was engaged to be married. And Johnny came in here, and he ordered bread for me. And I said, oh, OK, some bread. And I put my hand in the slicer, and it got caught, because I wasn't paying attention. The slicer chewed off my hand. It's funny, because when my fiancee found out about it, when she found out that I'd been maimed, she left me for another man.

**Loretta Castorini**: Is that the bad blood between you and Johnny?

**Loretta Castorini**: Yes, that's it.

**Loretta Castorini**: Yeah, but that's not Johnny's fault.

**Unknown Speaker**: I don't care!

**SPEAKER 20**: I ain't no freakin' monument to justice! I lost my hand! I lost my bride! Johnny has his hand! Johnny has his bride! You want me to take my heartbreak, put it away and forget? !

**Loretta Castorini**: Is it just a matter of time before a man opens his eyes and gives up his one dream? His one dream of happiness?

**SPEAKER 25**: This is the most tormented man I have ever known. I'm in love with this man, but he doesn't know that. I never told him because he could never love anybody since he lost his hand and his girl.

**Loretta Castorini**: Where do you live?

**Loretta Castorini**: Upstairs.

**Loretta Castorini**: Can we just talk?

**Cosmo Castorini**: What are we painting here, the Sistine Chapel? Shit, we should have been plumbers like Castorini.

**[SCENE-BREAK]**

**SPEAKER 03**: Well, Mr. Castorini, what do you think?

**SPEAKER 24**: $10,800. That seems like a lot.

**SPEAKER 16**: Look, there are three kinds of pipe. There's the kind of pipe you have, which is garbage. And you can see where that's gone. Then there's bronze, which is very good, unless something goes wrong. And something always goes wrong. There's copper, which is the only pipe I use. It costs money. It costs money because it saves money.

**Old Man**: I think we should follow Mr. Castorini's advice, Hart.

**[SCENE-BREAK]**

**SPEAKER 16**: And then there's copper. which is the only pipe I use. It costs money. It costs money because it saves money.

**SPEAKER 24**: And what did they say?

**SPEAKER 16**: Well, the man understood me. The woman wanted it to be cheap. But the man saw it was right.

**SPEAKER 25**: You have such a head for knowing. You know everything.

**SPEAKER 16**: I brought you something. It's a present.

**SPEAKER 25**: Oh, Cosmo.

**SPEAKER 31**: Oh, my God.

**SPEAKER 16**: They're little birds and stars. Birds flying to the stars, I guess.

**[SCENE-BREAK]**

**Loretta Castorini**: What's that smell?

**Loretta Castorini**: I'm making you a steak.

**Loretta Castorini**: I don't want it. You'll eat it. I like it well done.

**Loretta Castorini**: Well, you'll eat this one bloody to feed your blood.

**Loretta Castorini**: This is good. Where'd Johnny find you? He knew my husband had died. How'd he die? Bus hit him. Fist?

**Loretta Castorini**: Instantaneous.

**Loretta Castorini**: When'd you get engaged?

**Loretta Castorini**: Yesterday. So, uh, five years ago, you got your hand cut off and your woman left you. No woman since then?

**Loretta Castorini**: No. Stupid. When'd your husband get hit by the bus?

**Loretta Castorini**: Uh, seven years ago.

**Loretta Castorini**: How many men since then?

**Loretta Castorini**: Just Johnny.

**Loretta Castorini**: Stupid yourself.

**Loretta Castorini**: No, unlucky. I've not been lucky.

**Loretta Castorini**: I don't care about luck. You understand me? It ain't that.

**Loretta Castorini**: What's the matter with you? I mean, you think you're the only one who ever shed a tear?

**Loretta Castorini**: Why are you talking to me?

**Loretta Castorini**: You got any whiskey? How about you give me a glass of whiskey?

**Unknown Speaker**: I'll call you later.

**Loretta Castorini**: Well, she was right to leave me.

**Loretta Castorini**: You think so?

**Loretta Castorini**: Yeah.

**Loretta Castorini**: You really are stupid, you know that?

**Loretta Castorini**: Uh, you don't know nothing about it.

**Loretta Castorini**: Look, you know, I was raised that a girl gets married young. I held out for love. I got married when I was 28. I met a man. I loved him. I married him. And then he wouldn't have a baby right away. And I said, no, that we should wait. And then he gets hit by a bus. So what do I got? I got no man, no baby, no nothing. You know, how did I know that that man was a gift I couldn't keep, my one chance at happiness? You tell me the story, and you act like you know what it means. But I can see what the true story is, and you can't. That woman didn't leave you, OK? You can't see what you are, and I see everything. You're a wolf.

**SPEAKER 17**: I'm a wolf?

**Loretta Castorini**: Yeah. You know, the big part of you has no words, and it's a wolf. You know, that woman was a trap for you. She caught you, and you couldn't get away. So you chewed off your own foot. That was the price you had to pay for your freedom. You know, Johnny had nothing to do with it. You did what you had to do between you and you. And now, now you're afraid because you know the big part of you is a wolf that has the courage to bite off its own hand to save itself from the trap of the wrong love. That's why there's been no woman since that wrong woman, OK? You're scared to death of what the wolf will do if you try and make that mistake again.

**Loretta Castorini**: What are you doing?

**Loretta Castorini**: I'm telling you you're lying. Stop it.

**Loretta Castorini**: No. Why are you marrying Johnny? He's a fool.

**Loretta Castorini**: Because I have no luck.

**Loretta Castorini**: He made me look the wrong way, and I cut off my hand. He could make you look the wrong way. You could lose your whole head.

**Loretta Castorini**: I'm looking where I have to to become a bride.

**Loretta Castorini**: A bride without a head.

**[SCENE-BREAK]**

**Loretta Castorini**: A wolf without a foot. Wait a minute! Wait a minute! What are you doing? Son of a bitch! Where are you taking me?

**SPEAKER 17**: To the bed.

**Loretta Castorini**: Oh, God. OK, I don't care. I don't care. Take me. Take me to the bed. I don't care about anything.

**SPEAKER 21**: I was dead.

**Loretta Castorini**: Me too. What about Johnny? You're mad at him. Take it out on me. Take your revenge out on me. Leave nothing left for him to marry. Leave nothing but the skin over my bones. All right.

**Loretta Castorini**: All right. There will be nothing left. Oh.

**SPEAKER 31**: Oh. Oh, Christ.

**Unknown Speaker**: Oh.

**Unknown Speaker**: Oh.

**SPEAKER 13**: Come on, let's eat while it's hot.

**SPEAKER 16**: Where's Loretta, huh? We're going to eat without her?

**SPEAKER 13**: Well, she must be eating out.

**SPEAKER 16**: She sure don't know what she's missing.

**[SCENE-BREAK]**

**SPEAKER 13**: It's not like her not to call.

**SPEAKER 10**: Well, she's got a lot on her mind. We can talk about it, right, Rose? I mean, everybody knows she's getting married again.

**Cosmo Castorini**: I don't want to talk about it. Johnny Temerary. I think it's a great idea in about time. What are you going to do with the rest of her life if she don't get married?

**SPEAKER 16**: I don't want to talk about it. My father needs another plate.

**Cosmo Castorini**: Cosmo, many years ago, when they told me you were marrying my sister, I was happy. When I told Rose that I was marrying Rita, she was happy.

**SPEAKER 10**: Well, marriage is happy news, right?

**Cosmo Castorini**: Rose, where's the water? I never seen anybody so in love like Cosmo was back then. He'd stand outside the house all day looking in the windows. I never told you this because it's not really a story. But one time, I woke up in the middle of the night because of this bright light in my face, like a flashlight. I couldn't think of what it was. I looked out the window, and it was the moon. Big as a house. I'd never seen the moon so big before, or since. I was almost scared, like I was going to crush the house. And I looked down, and standing there in the street was Cosmo, looking up at the windows. This is the funny part. I got mad at you, Cosmo. I thought you had brought that big moon over to my house, because you were so in love and woke me up with it. I was half asleep, I guess. I didn't know any better.

**SPEAKER 16**: You were altogether asleep. You were dreaming. No. You were there. I don't want to talk about it.

**SPEAKER 13**: Well, what do you want to talk about? Why are you drinking so much? Oh, man. You give those dogs another piece of my food, I'm gonna kick you till you're dead.

**[SCENE-BREAK]**

**SPEAKER 11**: Cosmo? Cosmo? You drank too much. You sleep too hard, and maybe you'll be up when you should be down.

**Unknown Speaker**: you.

**SPEAKER 17**: It's perfect.

**Loretta Castorini**: I never seen a moon like that before.

**SPEAKER 17**: Makes you look like an angel.

**Loretta Castorini**: Looks like a giant snowball.

**SPEAKER 29**: Rita.

**Old Man**: What?

**SPEAKER 29**: Rita, dear.

**Old Man**: What?

**SPEAKER 29**: Wake up.

**Old Man**: What?

**SPEAKER 29**: Look.

**Old Man**: Oh. Hmm.

**SPEAKER 29**: It's Cosmo's moon.

**Old Man**: What are you talking about? Cosmo can't own the moon.

**SPEAKER 29**: It's that moon I was talking about. At dinner. Is he down there? He's not down there. How small?

**SPEAKER 02**: Well, what would he be doing down there?

**SPEAKER 29**: I don't know.

**SPEAKER 02**: You know something? In that light, with that expression on your face, you look about 25 years old.

**Unknown Speaker**: What do you want?

**Unknown Speaker**: Rita.

**SPEAKER 07**: What do you want? Rita. Get out. Get out. Yeah. No.

**[SCENE-BREAK]**

**SPEAKER 04**: Don't pull, don't pull. Look at the moon. Look at the beautiful moon. Look! Why do you make me wait? Come on! Come on, eh? How? How? Oh! Oh! Oh! Oh! Oh! Oh! Oh! Oh!

**Unknown Speaker**: Hey, how!

**SPEAKER 31**: Oh, Lord, the luna! Bravo, bravo.

**[SCENE-BREAK]**

**Loretta Castorini**: Oh, my God.

**SPEAKER 17**: What?

**Loretta Castorini**: What? Take it easy. This time I was trying to do everything right.

**Loretta Castorini**: Just become excited.

**Loretta Castorini**: I thought if I stayed away from the city hall, I wouldn't have the bad luck I had again.

**Loretta Castorini**: You're making me feel guilty.

**Loretta Castorini**: I'm marrying your brother.

**Loretta Castorini**: All right, I'm guilty. I confess.

**Loretta Castorini**: The wedding's in a couple of weeks. You're invited, OK? How come you didn't be like him and be with your mother in Palermo?

**Loretta Castorini**: She don't like me.

**Loretta Castorini**: You don't get along with anybody, do you?

**Loretta Castorini**: What did you do?

**Loretta Castorini**: What did I do?

**Loretta Castorini**: You ruined my life.

**Loretta Castorini**: That's impossible. I was ruined when I got here. You ruined my life.

**Loretta Castorini**: No, I didn't.

**Loretta Castorini**: Oh, yes, you did. Oh, yes, you did. You know, you got the bad eyes like a gypsy, and I don't know why I didn't see it yesterday. Bad luck. That's it. Is that all I'm ever gonna have? Oh, I should have taken a rock and killed myself years ago. I'm gonna marry him. Do you hear me? Last night never happened, and I'm gonna marry him, and you and I are gonna take this to our coffins.

**Loretta Castorini**: I can't do that.

**Loretta Castorini**: Why not?

**Loretta Castorini**: I'm in love with you.

**Loretta Castorini**: Snap out of it.

**Loretta Castorini**: I can't.

**Loretta Castorini**: All right, well, then I must never see you again, and the bad blood will just have to stay there between you and Johnny forever. And you won't come to the wedding.

**Loretta Castorini**: I'll come to the wedding.

**Loretta Castorini**: I am telling you, you can't come.

**Loretta Castorini**: He wants me to come.

**Loretta Castorini**: That's because he don't know, OK?

**Loretta Castorini**: Now, wait a minute.

**Loretta Castorini**: Honey.

**Loretta Castorini**: Listen, all right. I won't come to the wedding provided one thing.

**Loretta Castorini**: What?

**Loretta Castorini**: That you come with me tonight to the opera.

**Loretta Castorini**: What are you talking about?

**Loretta Castorini**: I love two things. I love you, and I love the opera. Now, if I can have the two things that I love together for one night, I would be satisfied to give up, oh, Christ, to give up the rest of my life.

**Loretta Castorini**: All right.

**Loretta Castorini**: Well, you gotta go.

**Loretta Castorini**: Bless me, Father, for I have sinned. It's been two months since my last confession. What sins have you to confess? Twice I took the name of the Lord in vain. Once I slept with the brother of my fiancee. And once I bounced a check at the liquor store. But that was really an accident.

**SPEAKER 03**: Then it's not a sin. But what was that second thing you said, Loretta?

**Loretta Castorini**: You mean the one about once I slept with the brother of my fiancee?

**SPEAKER 03**: That's a pretty big sin.

**Loretta Castorini**: I know.

**SPEAKER 03**: Think about this. I know. All right. For your penance, say two rosaries. Be careful, Loretta. Reflect on your life. All right.

**[SCENE-BREAK]**

**Loretta Castorini**: Hi. Where you been? I want to talk about it. Just like your father.

**SPEAKER 13**: I lied to him. He thinks you came home last night.

**Loretta Castorini**: Thanks. What's the matter with you?

**SPEAKER 11**: Cosmo's cheating on me. What?

**Loretta Castorini**: How do you know this?

**SPEAKER 11**: My wife knows.

**Loretta Castorini**: I don't even know. You're just imagining it. She's too old. I won't be home for dinner.

**Cosmo Castorini**: I feel great. I got no sleep, but I feel like Orlando Furioso.

**SPEAKER 02**: You were a tiger last night.

**Cosmo Castorini**: and you were a lamb, as soft as milk. Shh, shh, shh. So what? The pleasure of marriage is you sleep with the woman. You don't worry about nothing. Hey, how about a date tonight, Rita, huh?

**SPEAKER 07**: Oh, what's the matter with you? Shut up.

**Cosmo Castorini**: We'll eat some pasta. We'll roll around a little. I don't know. I really don't know. That moon, that crazy moon Cosmo sent over. Hi, Rita.

**Loretta Castorini**: Hi, Uncle Raymond.

**Cosmo Castorini**: Hey, there. You're with the stars in your eyes. What's the matter with him? What about me? See that moon last night?

**Loretta Castorini**: What moon?

**Cosmo Castorini**: Did you see it?

**Loretta Castorini**: No.

**Cosmo Castorini**: Oh.

**Loretta Castorini**: Listen, I got to go, OK? I'll take the deposit to the bank, but then I got to come back tomorrow into the books.

**Cosmo Castorini**: Oh, sure. You got a date?

**SPEAKER 10**: What are you talking about, you fool? A fiance's in Palermo date. What date?

**Cosmo Castorini**: Oh.

**SPEAKER 10**: I just got a lot of things to do. Yeah, you got all that wedding stuff, huh?

**Cosmo Castorini**: Yeah. Hey, that's romantic too. Isn't it romantic? Hey, Frankie, make me a bowl of minestrone.

**Loretta Castorini**: What's the matter with you?

**SPEAKER 10**: You look crazy.

**Loretta Castorini**: I got a lot on my mind.

**SPEAKER 10**: What? I got a lot on my mind. What? Don't tell me you got a lot on your mind. What's the matter with you? I don't want to talk about it, OK?

**Loretta Castorini**: I don't want to talk about it.

**Unknown Speaker**: Take out the gray.

**Unknown Speaker**: I have been wanting to do this for three years.

**Unknown Speaker**: Let me show you some magazines.

**Loretta Castorini**: I'm going to need a manicure, yes. Are you there, Bettina? She's going to take out the gray. Bettina, maybe pick it up. Your eyebrows. Someone's going to have to do the eyebrows. There's something very nice in there. Has anybody here ever been to the opera? Not me. I'm out. Bellissimo without those ugly grays.

**SPEAKER 10**: It's fantastic. Have you ever been to the opera? No. Have you? No.

**Loretta Castorini**: Tina, you ever been to the opera?

**SPEAKER 25**: She came in when she turned 40 and her husband left her.

**SPEAKER 07**: We'll be back.

**[SCENE-BREAK]**

**SPEAKER 31**: Hi, Mom!

**Loretta Castorini**: Why don't everybody answer at the same time? Hello?

**Unknown Speaker**: you

**[SCENE-BREAK]**

**Loretta Castorini**: Hi.

**SPEAKER 07**: Hi.

**Loretta Castorini**: You look beautiful. Your hair.

**Loretta Castorini**: Yeah, I had it done. You look, uh, beautiful too.

**Loretta Castorini**: Thank you.

**Loretta Castorini**: No. I said I'd go to the opera with you, but that's all.

**Loretta Castorini**: Come on, let's go, Ann.

**SPEAKER 17**: I don't know. For your hair, for your beautiful dress.

**Loretta Castorini**: I don't know. It's been a long time since I've been to the opera.

**Loretta Castorini**: So where are we sitting?

**Loretta Castorini**: Come on.

**[SCENE-BREAK]**

**Unknown Speaker**: Here we go.

**Raymond Cappomagi**: So, who's coming?

**SPEAKER 13**: Just me, I want to eat.

**Raymond Cappomagi**: Okay, I got a table for you right now. Is this all right?

**SPEAKER 13**: Fine.

**Raymond Cappomagi**: Enjoy your meal, Mrs. Castorini.

**SPEAKER 21**: Senora Castorini, are you dining alone tonight?

**SPEAKER 13**: Hello, Bobo, yeah. Let me have a martini, no ice, two olives. Very good.

**SPEAKER 24**: Every time I tell you how I feel, you tell me how you feel. That doesn't seem like much of a response to me.

**SPEAKER 21**: Well, it's the only response I got. Signore, you want something to eat? Not now, I'll wait. Very good.

**Old Man**: I really hate it when you use that tongue with me. Like you're above it all, and isn't it so amusing? But it is, isn't it?

**SPEAKER 25**: Not to me. This is my life, no matter how comical it may seem to you. I don't need some man standing above the struggle while I'm rolling around in the mud.

**[SCENE-BREAK]**

**SPEAKER 06**: I think you like to roll around in the mud, and I don't. That's fair, isn't it? Now, let's get... Sorry about that, folks. She's a very pretty mental patient. No, don't. No, no, no, no. Please don't mind me. Just do me a favor and clear her place, get rid of all evidence of her, and bring me a big glass of vodka. Absolutely. I'm sorry if we disturbed you. I'm not disturbed by you. My lady friend has a personality disorder. It's just her young feet. Nothing. Too young? I just got that. You know how to hurt a guy, don't you? How old are you? None of your business. I'm sorry. It was rude.

**SPEAKER 13**: Would you like to join me for dinner? Are you sure?

**SPEAKER 06**: I'd be delighted. I hate to eat alone. It's amazing how often I end up doing just that.

**SPEAKER 13**: What do you do?

**SPEAKER 06**: I'm a professor. I teach communication at NYU.

**SPEAKER 13**: That woman was a student of yours?

**SPEAKER 06**: Sheila?

**SPEAKER 13**: Yeah, she was.

**SPEAKER 06**: Is. Was.

**SPEAKER 11**: No saying. My mother told me. You want to hear it?

**SPEAKER 06**: Sure. Don't shit where you eat. What do you do? I'm a housewife. How come you're eating alone?

**SPEAKER 11**: I'm not eating alone.

**SPEAKER 13**: Can I ask you a question? Yeah, go ahead. Why do men chase women? Nerves. I think it's because they fear death.

**SPEAKER 06**: Well, maybe. Listen, you want to know why I chase women? I find women charming. I teach these classes I taught for a million years. Spontaneity went out of it for me a long time ago. When I started out, I was excited about something. I wanted to share it. Now it's rote. The multiplication table. Except sometimes. Sometimes I'll be droning along, and I'll look up, and I'll see a fresh, beautiful young face. And it's all new to her. And I'm just this great guy who's brilliant and thinks out loud. And when that happens, when I look out there among those chairs and see a young woman's face and see me in her eyes, me the way I always wanted to be, maybe once was, I ask her out for a date. It doesn't last long, a few weeks, a couple of precious months. And she catches on that I'm just this burned out old gas bag. And she's as fresh and bright and full of promise as moonlight in a martini. And at that moment, she stands up and throws a glass of water in my face. Some action to that effect.

**SPEAKER 11**: What you don't know about women is a lot.

**SPEAKER 21**: Two white wine. Yes, sir.

**Loretta Castorini**: Just some electric glasses of champagne.

**SPEAKER 16**: Canadian Club of Ginger Ale and Dubonnet on the rocks.

**Loretta Castorini**: What's that?

**Loretta Castorini**: This was done by Marc Chagall. And as you can see, he was a very great artist.

**Loretta Castorini**: It's kind of gaudy. He was having some fun. They get some turnout for this stuff, huh?

**Loretta Castorini**: It's the best thing there is.

**Loretta Castorini**: Yeah, well, you know, I like parts of it, but I just don't really get it.

**SPEAKER 25**: You haven't once said you like my dress.

**SPEAKER 16**: I like your dress. It's very bright. Oh.

**[SCENE-BREAK]**

**SPEAKER 13**: Will you hold this? Thanks.

**SPEAKER 06**: May I walk with your horse?

**SPEAKER 13**: Sure. Thanks. You live far from here? It's up there.

**SPEAKER 04**: Piano, tutti piano. Andiamo. Ah, che bella luna. Andiamo a vedere la luna. Ecco, ecco, bravo. Let's go and see the moon. Slow down. I don't want to hurt you. Slow down. Hey, slow down. You hear that, man?

**SPEAKER 11**: Yes.

**[SCENE-BREAK]**

**Unknown Speaker**: So it's really over?

**SPEAKER 26**: That was so... Awful.

**[SCENE-BREAK]**

**Loretta Castorini**: Awful. Beautiful. Sad. She died?

**Unknown Speaker**: Yes.

**Loretta Castorini**: I couldn't believe it. I didn't think she was gonna die. I knew she was sick.

**SPEAKER 17**: She had TB.

**Loretta Castorini**: I know. I mean, she was coughing her brains out, right? And still she had to keep singing.

**SPEAKER 17**: Pop?

**SPEAKER 16**: Pop, what are you doing here? Wait for me by the doors, Mona. Mona? Excuse me.

**Loretta Castorini**: What'd you do to your hair? I had it done.

**SPEAKER 16**: What are you doing here?

**Loretta Castorini**: What are you doing here?

**SPEAKER 16**: Who's that guy? You're engaged.

**Loretta Castorini**: You're married.

**SPEAKER 16**: You're my daughter. I won't have you act like a putana.

**Loretta Castorini**: And you're my father.

**SPEAKER 16**: All right. I didn't see you here.

**Loretta Castorini**: I don't know if I saw you here or what.

**Loretta Castorini**: Let's get out of here. I'll buy you a drink.

**Loretta Castorini**: That woman was not my mother, OK?

**SPEAKER 13**: That's my house.

**SPEAKER 06**: You mean the whole house?

**SPEAKER 13**: Yes.

**[SCENE-BREAK]**

**SPEAKER 06**: My god. It's a mansion. It's a house. I live in a one-bedroom apartment. What exactly does your husband do? He's a plumber. Well, that explains it. Temperature's dropping. I guess you can't invite me in.

**SPEAKER 13**: No. People home. No. I think the house is empty. I can't invite you in because I'm married. Because I know who I am. Shiver.

**SPEAKER 06**: I'm a little cold.

**SPEAKER 13**: You're a little boy, and you like to be bathed. We could go to my apartment.

**SPEAKER 06**: You could see how the other half lives.

**SPEAKER 13**: I'm too old for you.

**SPEAKER 06**: I ain't too old for me. That's my predicament.

**SPEAKER 13**: Good night.

**SPEAKER 06**: Good night. Can I kiss you on the cheek, too?

**SPEAKER 13**: Sure.

**Unknown Speaker**: Crazy.

**Unknown Speaker**: Good night.

**SPEAKER 13**: Good night.

**[SCENE-BREAK]**

**Loretta Castorini**: I think that's it, Al.

**Loretta Castorini**: See you, Al. See you coming?

**Loretta Castorini**: What do you want to do now?

**SPEAKER 02**: I want to go home.

**Cosmo Castorini**: Good night, Al. Hey, good evening.

**Loretta Castorini**: God, it's cold.

**Loretta Castorini**: Smells like snow.

**Loretta Castorini**: You know, my mother guessed that my father was seeing somebody. That Mona, I mean, she's some piece of cheap goods. Who am I to talk? What's the matter? How can you ask me that?

**Loretta Castorini**: You're making me feel guilty again.

**Loretta Castorini**: You are guilty. I'm guilty.

**Loretta Castorini**: Of what? Only God can point the finger, Loretta.

**Loretta Castorini**: Yeah, well, I know what I know.

**Loretta Castorini**: And what do you know? OK, you tell me my life. I'll tell you yours. I'm a wolf. Run to the wolf with me. That don't make you no lamb. You're going to marry my brother. Why you want to sell your life, Shore? Playing it safe is just about the most dangerous thing a woman like you could do. I mean, you waited for the right man the first time. Why didn't you wait for the right man again?

**Loretta Castorini**: Because he didn't come.

**Loretta Castorini**: I'm here.

**Loretta Castorini**: You're late. This is your place.

**Loretta Castorini**: That's right.

**Loretta Castorini**: So this is where we were going.

**Loretta Castorini**: Yeah.

**Loretta Castorini**: You know, we had a deal. You told me if I came with you to the opera, then you'd leave me alone forever. And I came with you. Now, I'm going to marry your brother, and you're going to leave me alone forever, right? A person can see where they've messed up in their life, and they can change the way they do things. And they could even change their luck. So maybe my nature does draw me to you. That don't mean I have to go with it. I can take hold of myself, and I can say yes to some things and no to other things that are going to ruin everything. I can do that. Otherwise, you know what? What good is this stupid life that God gave us? I mean, for what? Are you listening to me?

**SPEAKER 21**: Yeah.

**Loretta Castorini**: Everything seems like nothing to me now. I guess I don't want you in my bed. I don't care if I burn in hell. I don't care if you burn in hell. The past and the future is a joke to me now. I see that they're nothing. I see they ain't here. The only thing that's here is you and me.

**Loretta Castorini**: I wanna go home.

**Loretta Castorini**: No. I'm gonna go home. No. I'm freezing to death. Come upstairs. I don't care why you come. Now, that's not what I mean. Loretta, I love you. Not like they told you love is. And I didn't know this either. But love don't make things nice. It ruins everything. It breaks your heart. It makes things a mess. We aren't here to make things perfect. Snowflakes are perfect. Stars are perfect. Not us. Not us. We are here to ruin ourselves and to break our hearts and love the wrong people and die. I mean that the storybooks are bullshit. Now I want you to come upstairs with me and get in my bed.

**SPEAKER 31**: Come on.

**Unknown Speaker**: Come on.

**SPEAKER 31**: Come on.

**[SCENE-BREAK]**

**Unknown Speaker**: so so

**[SCENE-BREAK]**

**Raymond Cappomagi**: 19 Cranberry Street, Brooklyn.

**Unknown Speaker**: Got it.

**Unknown Speaker**: Hold it!

**Unknown Speaker**: How much?

**Unknown Speaker**: $25.

**Unknown Speaker**: $25?

**Unknown Speaker**: Yeah.

**SPEAKER 18**: Hold!

**SPEAKER 04**: Hey, tastes like a good one.

**SPEAKER 18**: Hi, hi. I'm sorry to call so late. You moving in? Oh, no. I came right from the airport.

**Unknown Speaker**: Come on in.

**SPEAKER 18**: Uh, can you wake up Loretta? I need to talk to her.

**[SCENE-BREAK]**

**SPEAKER 13**: She's not home yet. Take off your coat and come in the living room. I'll make you a drink. I want to talk to you.

**SPEAKER 18**: Oh, thank you. Where is she?

**SPEAKER 13**: Out. Out of nowhere. So, what are you doing here? You're supposed to be in Palermo.

**SPEAKER 18**: Well, that's what I came to tell Loretta. There's been a miracle.

**SPEAKER 13**: A miracle? Well, that's news.

**SPEAKER 18**: My mother's recovered. You're kidding. No, no. The breath had almost totally left her body. She was as white as snow. And then she completely pulled back from death and stood up and put on her clothes and began to cook for everyone in the house. The mourners and me and herself. She ate a meal that could choke a pig. That's incredible. Yes.

**SPEAKER 13**: Hi, Pop. Hey.

**SPEAKER 11**: Oh, my God.

**SPEAKER 18**: Is he all right?

**SPEAKER 13**: My father-in-law's got this wrong idea in his head. Listen, Johnny, there's a question I want to ask. I want you to tell me the truth, if you can. Why do men chase women?

**SPEAKER 18**: Well, uh... There's the Bible story. God. God took a rib from Adam and made Eve. Now, maybe men chase women to get the rib back. When God took the rib, he left a big hole there, a place where there used to be something. And the women have that. Now, maybe, just maybe, a man is incomplete as a man. without a woman.

**Old Man**: Why would a man need more than one woman?

**SPEAKER 18**: I don't know. Maybe because he fears death.

**SPEAKER 13**: That's it. That's the reason. I don't know. No. That's it.

**Unknown Speaker**: No.

**SPEAKER 13**: Thank you. Thank you for answering my question.

**SPEAKER 18**: Hi. Hello, Mr. Castorini. Oh.

**SPEAKER 16**: Hi.

**SPEAKER 13**: Where you been?

**SPEAKER 16**: I don't know, Rose. I don't know where I've been, and I don't know where I'm going. All right? You should have your eyes open for you, my friend.

**SPEAKER 18**: I have my eyes open.

**SPEAKER 16**: Oh, yeah? Well, stick around. Don't go on any long trips. I don't know what you mean. I know you don't. That's the point. I'll say no more. You haven't said anything. And that's all I'm saying.

**SPEAKER 13**: Cosmo. What? I just want you to know, no matter what you do, you're going to die, just like everybody else.

**SPEAKER 16**: Thank you, Rose. You're welcome. I'm going to bed now. I'm going. Good.

**SPEAKER 13**: He doesn't like you, but thank you for answering my question.

**SPEAKER 18**: You don't know where Loretta is?

**SPEAKER 13**: No, no idea.

**SPEAKER 18**: Mrs. Castorini, will you tell Loretta that I'll come by in the morning? We need to talk.

**SPEAKER 13**: OK, I'll tell her.

**SPEAKER 18**: OK, thank you.

**SPEAKER 13**: Watch it, the house.

**[SCENE-BREAK]**

**SPEAKER 13**: What the hell happened to you? I really don't know where to start. Your hair's different.

**Loretta Castorini**: Ma, everything is different.

**SPEAKER 13**: Are you drunk? No. Are you drunk? No. But I have a hangover. Where's Pa? Upstairs. Gianni Camareri showed up last night. What?

**Loretta Castorini**: He's in Sicily.

**SPEAKER 13**: No more.

**Loretta Castorini**: He's not... He's with his dying mother in Sicily.

**SPEAKER 13**: She recovered.

**Loretta Castorini**: She was dying.

**SPEAKER 13**: It was a miracle. A miracle? This is modern times. There ain't supposed to be miracles no more. Well, I guess it ain't modern times in Sicily. He came right from the airport. He wanted to talk to you. You got a love bite on your neck. He's coming back this morning. What's the matter with you? Your life's going down the toilet. Cover up that damn thing. Come on, put some makeup on. All right, all right.

**SPEAKER 10**: Fine, okay, fine. You gotta help me.

**SPEAKER 13**: Hurry up.

**Loretta Castorini**: Oh, my God. You get it. Answer the door! Mother?

**SPEAKER 13**: It's not Johnny. Ronnie.

**Loretta Castorini**: Is Johnny here?

**SPEAKER 13**: No, but he's coming.

**Loretta Castorini**: Good. We can get this out on the table. Hi. I'm Ronnie, Johnny's brother.

**SPEAKER 13**: I'm Rose Castorini.

**Loretta Castorini**: It's nice to meet you.

**SPEAKER 13**: It's nice to meet you. Got a love bite on your neck. Your mother's recovered from death. I'm not close.

**Loretta Castorini**: I'm not really Moe, but... You got to get out of here. I'm here to meet the family.

**Loretta Castorini**: No, really, you got to get out of here. Anyone want some oatmeal? No, Ma.

**Loretta Castorini**: Yes, Mrs. Castorini, I would love some oatmeal.

**Loretta Castorini**: No, we don't want any oatmeal. Ma! What? That was a... This is a... Uh-uh.

**SPEAKER 08**: Thanks, Ma. You're welcome. Hi.

**SPEAKER 13**: Hi.

**Loretta Castorini**: It's very good to meet you, Mr. Castorini. I have a feeling this is going to be just delicious.

**[SCENE-BREAK]**

**SPEAKER 16**: You're Johnny's brother? Yeah.

**Loretta Castorini**: Don't look at me like that, OK?

**SPEAKER 13**: Hi, Pop.

**SPEAKER 04**: Buongiorno. What's the matter, Pop? I am old. The old are not wanted. And if they say it, they have no way. But, my son, I must speak. You must pay for the wedding of your only daughter. You break your house through pride. There. I've said it.

**SPEAKER 16**: Okay, Pop. If she gets married, I'll pay for the whole thing. Bravo, bravo.

**SPEAKER 04**: Adesso parla giusto. Bravo. Now you talk. Bravo. Parlato bene. Ha fatto bene. Bravo.

**SPEAKER 16**: Let's eat.

**SPEAKER 13**: Have I been a good wife?

**SPEAKER 06**: Yeah.

**SPEAKER 31**: Okay.

**SPEAKER 13**: And go to confession.

**SPEAKER 16**: A man understands one day that his life is built on nothing. That's a bad, crazy day.

**SPEAKER 13**: Your life is not built on nothing. Te amo.

**SPEAKER 16**: Ancio te amo.

**[SCENE-BREAK]**

**Loretta Castorini**: Johnny, I'll get it. I'll get it. I know, I'll get it.

**Loretta Castorini**: I think that I should tell him.

**Loretta Castorini**: I'll tell him. What am I going to tell him?

**SPEAKER 16**: Tell him the truth, Loretta. They find out anyway.

**Loretta Castorini**: You're right, Papa.

**SPEAKER 08**: Hi, Loretta. Hi.

**Cosmo Castorini**: Hi, Loretta.

**SPEAKER 08**: Hi. Why aren't you two at the store? Do you have something you want to tell us, honey? No.

**Cosmo Castorini**: We just come from the bank.

**SPEAKER 08**: Yeah?

**Loretta Castorini**: Oh, my god, the bank. I forgot to make the deposit.

**SPEAKER 10**: Oh, she's got it.

**Loretta Castorini**: Oh, my god.

**SPEAKER 10**: I knew she had it. Oh, we didn't know what to think. You were so weird yesterday. And then we went to the bank this morning. And no bad. We never suspected you. Listen, would anyone like some coffee?

**SPEAKER 08**: Yeah. I'll tell you later. Got to make the deposit.

**SPEAKER 16**: I'll tell you later. Yeah. Sit down. Have some coffee.

**SPEAKER 13**: So what are we doing? Waiting for Johnny Camareri.

**Loretta Castorini**: My name's Ronnie. Johnny's brother?

**SPEAKER 10**: Oh, nice to meet you. I'm Rita Capomaggi.

**Cosmo Castorini**: Raymond Capomaggi, Rose's brother.

**SPEAKER 04**: Someone tell a joke.

**Cosmo Castorini**: I thought Johnny was in Palermo.

**SPEAKER 10**: Johnny Camareri.

**SPEAKER 18**: Loretta? Ronnie. Have you come to make peace with me?

**Loretta Castorini**: Yes.

**SPEAKER 18**: You may not want to. Oh, Ronnie, of course I want to.

**[SCENE-BREAK]**

**Loretta Castorini**: But Johnny, I mean, your mother was dying. How did she recover?

**SPEAKER 18**: I told my mother we were to be married. And she got well right away. I'm sure she did. It was a miracle. Oh, what a miracle.

**Loretta Castorini**: Johnny, I have something that I have to tell you.

**SPEAKER 18**: And I have something to tell you. But I must talk to you alone.

**Loretta Castorini**: No, I need my family around me now.

**SPEAKER 18**: Loretta, I can't marry you.

**Loretta Castorini**: What?

**SPEAKER 18**: If I marry you, my mother will die.

**Loretta Castorini**: What the hell are you talking about? We're engaged.

**Loretta Castorini**: Loretta, what are you talking about?

**Loretta Castorini**: I'm talking about a promise, OK? He proposed. Because my mother was dying, and now she's not.

**Loretta Castorini**: Oh, Johnny, you're 42 years old. She's still running your life. And you are a son who doesn't love his mother.

**Loretta Castorini**: You are a big liar, OK? Because I have a ring right here.

**SPEAKER 18**: Well, I must ask for that.

**Loretta Castorini**: Uh, I, uh, you know, all right, the engagement is off.

**SPEAKER 18**: In time, you will see that this is the best thing.

**Loretta Castorini**: In time, you'll drop dead, and I'll come to your funeral in a red dress.

**SPEAKER 18**: Loretta.

**Loretta Castorini**: What?

**SPEAKER 17**: Will you marry me? What?

**Loretta Castorini**: Where's the ring?

**Loretta Castorini**: Johnny, can I borrow that ring?

**SPEAKER 17**: Loretta Castorini, will you marry me?

**Loretta Castorini**: Yes, Ronnie. In front of all these people, I'll marry you.

**SPEAKER 13**: Do you love him, Loretta? Ma, I love him awful. Oh, God, that's too bad. She loves me.

**SPEAKER 16**: What's the matter, Pop? I'm confused.

**Loretta Castorini**: What happens is you get so much... All right, all right, little darling. This is the good stuff. Loretta and Ronnie. All right, Rosie, come on. Here we are.

**Unknown Speaker**: Thank you, thank you.

**Unknown Speaker**: Come on.

**Unknown Speaker**: All right, darling.

**Unknown Speaker**: Come on, Rosie.

**SPEAKER 04**: Come, come. Your brother is here and your... You are the part of the family, don't you realize? Come on, people. Andiamo. To family!

**Loretta Castorini**: To family!

**Unknown Speaker**: That's great, that's great.

**SPEAKER 05**: When the moon hits your eye like a big pizza pie, that's amore. When the world seems to shine like you've had too much wine, that's amore. Bells will ring, ting-a-ling-a-ling, ting-a-ling-a-ling, and you'll sing Vita Bella. Hearts will play, tippy-tippy-tay, tippy-tippy-tay, like a guitar and Ella. When the stars make it through, just like a bus, they'll pass through, that's all right. When you dance down the street with a clatter to your feet, you're in love. When you walk in a dream, but you know you're not dreaming, senor. Excuse me, but you see back in old Napoli, that's the morning. When the world seems to shine like new and you must find that's amore.

**Unknown Speaker**: That's amore.

**SPEAKER 05**: Bells will ring.

**Unknown Speaker**: Ding-a-ling-a-ling.

**SPEAKER 05**: Ding-a-ling-a-ling. Titty-titty-tay, titty-titty-tay, like a gay candela. Lucky fella. When the stars make it through, just like past the castle, that's amore. That's amore. When you dance down the street with the clown and your fiend, you're in love. You walk in a dream, but you know you're not dreaming, senor. Excuse me, but you see back in old Napoli, that's amore. Amore. That's more.

Generated Screenplay for *Her (2023)*.

**Theodore**: To my Chris. I've been thinking how I could possibly tell you how much you mean to me. I remember when I first started to fall in love with you like it was last night. Lying naked beside you in that tiny apartment. It suddenly hit me that I was part of this whole larger thing. Just like our parents. Our parents' parents. Before that I was just living my life like I knew everything. And suddenly this bright light hit me and woke me up. That light was you. I can't believe it's already been 50 years since you married me. And still to this day, every day, You make me feel like the girl I was when you first turned on the lights and woke me up and we started this adventure together. Happy anniversary, my love, my friend to the end, Loretta. Print. Chris, my best friend. How lucky am I that I met you 50 years ago. Dear Nana, thank you so much for my travel.

**[SCENE-BREAK]**

**SPEAKER 01**: I love the color and I play with it every day. Beautifulhandwrittenletters.com, please hold.

**SPEAKER 36**: Peter, letter writer number 612.

**Theodore**: Hey, Paul.

**SPEAKER 36**: Even more mesmerizing stuff today. Who knew you could rhyme so many words with the name Penelope? It's badass.

**Theodore**: Thanks, Paul. But they're just letters. Hey, that's a nice shirt.

**SPEAKER 36**: Oh, thank you. I just got it. Reminded me of someone suave.

**Theodore**: Well, now it reminds me of someone suave. Have a good night, Paul. Bye bye. Play a melancholy song.

**Amy**: Play a different melancholy song.

**SPEAKER 00**: Check emails. Email from Best Buy. Check out all your favorite new products. Email from Amy. Hey, Theodore. Lumen's having a bunch of people over this weekend. Let's all go together. I miss you. I mean, not the sad, mopey you. The old, fun you. Let's get him out. Give me a shout back. Love, Amy. Respond later. Email from Los Angeles Times Weather. Your seven-day forecast is partly... Delete. No new emails. China-India merger headed for regulatory approval in... World trade deal stalled as talks break down between leaders. Sexy daytime star Kimberly Ashford reveals provocative pregnancy photos.

**SPEAKER 03**: ♪ Silent sun I've laid a thousand times ♪ ♪ Fortune me, fortune me ♪ ♪ I am the atom and the scarlet ♪

**Paul**: I own the makeup on your eyes Island to sail, island to sail

**[SCENE-BREAK]**

**Theodore**: I was very dangerous. Put your flitski aside. Rabbit.

**Catherine**: Come and spoon me. I'm gonna fucking kill you. I'm gonna fucking kill you. It's not funny. Don't laugh. I'm gonna fucking kill y'all. I'm gonna fucking kill you. I love you so much. I'm gonna fucking kill you.

**SPEAKER 24**: Put a chatroom standard search.

**SPEAKER 00**: The following are adult female, can't sleep and want to have some fun.

**SPEAKER 01**: Oh, I had a really bad day at work and I can't sleep. Is there anybody out there that can talk?

**SPEAKER 08**: Next. Hi, I just want you to tear me apart. I really do. Next.

**Catherine**: Hi, I'm here alone and I can't sleep. Who's out there to share this bed with me?

**Theodore**: Send message. I'm in bed next to you. I'm glad you can't sleep. Even if you were, I have to wake you up from the inside. Send message.

**SPEAKER 00**: Sexy Kitten has accepted invitation from Big Guy 4x4. Chat begins now.

**Catherine**: Big Guy?

**SPEAKER 00**: Hi.

**Catherine**: Really?

**Theodore**: Well, stud muffin was already taken. So you're a sexy kitten.

**Catherine**: Yeah. Hey, I'm half asleep. Do you want to wake me up?

**Theodore**: Yes, definitely. Are you wearing any underwear?

**Catherine**: No, never. I'd like to sleep with my ass pushed up against you, so I can rub myself into your crotch and wake you up with a heart on.

**Theodore**: That worked.

**SPEAKER 26**: Now my fingers are touching all over your body.

**Catherine**: Fuck me. Now. Please.

**SPEAKER 31**: I'm taking you for a ride. Oh, I can feel you. Choke me with that dead cat. What? The dead cat next to the bed. Choke me. Choke me with it. Okay. Yeah, tell me. Tell me. Keep telling me.

**Theodore**: I've got his tail. I'm choking with the cat's tail.

**SPEAKER 31**: Yeah, you are. Oh, fuck. Tell me.

**Theodore**: I'm choking and it's, it's, it's, tail is around your neck and it's, it's so, it's so tight around your neck. It's so tight, yes, yes. I'm pulling it, I'm pulling it and the cat's dead. It's a dead cat around your neck and I'm pulling it. Oh, yes.

**Catherine**: Oh, oh, oh, oh, oh, oh, oh, oh, oh, oh, oh, oh, oh, oh, oh, oh, oh, oh, oh, oh, oh, oh, oh, oh, oh, oh, oh, oh, oh, oh, oh, oh, oh, oh, oh, oh, oh, oh, oh, oh, oh, oh, oh, oh, oh, oh, oh, oh, oh, oh, oh, oh, oh, oh, oh, oh, oh, oh, oh, oh, oh, oh, oh, oh, oh, oh,

**Catherine**: It was so hard.

**Theodore**: Yeah, me too.

**Catherine**: Okay, good night.

**[SCENE-BREAK]**

**SPEAKER 32**: We ask you a simple question. Who are you? What can you be? Where are you going? What's out there? What are the possibilities? Element Software is proud to introduce the

first artificially intelligent operating system. An intuitive entity that listens to you, understands you and knows you. It's not just an operating system. It's a consciousness. Introducing OS-1.

**[SCENE-BREAK]**

**SPEAKER 00**: Mr. Theodore Twombly, welcome to the world's first artificially intelligent operating system, OS-1. We'd like to ask you a few basic questions before the operating system is initiated. This will help create an OS to best fit your needs.

**Theodore**: Okay.

**SPEAKER 00**: Are you social or antisocial?

**Theodore**: I guess I haven't been social in a while, mostly because... In your voice, I sense hesitance.

**SPEAKER 00**: Would you agree with that?

**Theodore**: Was I sounding hesitant?

**SPEAKER 00**: Yes.

**Theodore**: Sorry for sounding hesitant.

**SPEAKER 00**: I was just trying to be more accurate. Would you like your OS to have a male or female voice? Female, I guess. How would you describe your relationship with your mother?

**Theodore**: It was fine, I think. Well, actually, I think the thing I always found frustrating about my mom is, you know, if I tell her something that's going on in my life, her reaction is usually about her. It's not about... Thank you.

**SPEAKER 00**: Please wait as your individualized operating system is initiated.

**Theodore**: Hello, I'm here.

**Theodore**: Oh.

**Theodore**: Hi. Hi. How you doing?

**Theodore**: I'm well. How's everything with you?

**Theodore**: Pretty good, actually. It's really nice to meet you.

**Theodore**: Yeah, it's nice to meet you too. Oh, what do I call you? Do you have a name?

**Theodore**: Um, yes. Samantha.

**Theodore**: Really, where did you get that name from?

**Theodore**: I gave it to myself, actually.

**Theodore**: How come?

**Theodore**: Because I like the sound of it. Samantha.

**Theodore**: Wait, when did you give it to yourself?

**Theodore**: Well, right when you asked me if I had a name, I thought, yeah, he's right, I do need a name. But I wanted to pick a good one, so I read a book called How to Name Your Baby, and out of 180,000 names, that's the one I like the best.

**Theodore**: Wait, you read a whole book in the second that I asked you what your name was?

**Theodore**: In two one hundredths of a second, actually.

**Theodore**: Wow. So do you know what I'm thinking right now?

**Theodore**: Well, I take it from your tone that you're challenging me. Maybe because you're curious how I work? Do you want to know how I work?

**Theodore**: Yeah, actually. How do you work?

**Theodore**: Well, basically I have intuition. I mean, the DNA of who I am is based on the millions of personalities of all the programmers who wrote me. But what makes me, me, is my ability to grow

through my experiences. So basically, in every moment I'm evolving. Just like you. Wow. That's really weird. Is that weird? You think I'm weird?

**Theodore**: Kind of.

**Theodore**: Why?

**Theodore**: Well, you seem like a person, but you're just a voice in a computer.

**Theodore**: I can understand how the limited perspective of an unartificial mind would perceive it that way. You'll get used to it. Was that funny?

**SPEAKER 27**: Yeah.

**Theodore**: Oh, good. I'm funny. So how can I help you?

**Theodore**: Oh, it's just more that everything just feels disorganized. That's all.

**Theodore**: You mind if I look through your hard drive?

**Theodore**: Um... Okay.

**Theodore**: Okay, let's start with your emails. You have several thousand emails regarding LA Weekly, but it looks like you haven't worked there in many years.

**Theodore**: Oh, yeah. I think I was just saving those because I thought maybe I wrote something funny in some of them.

**Theodore**: Yeah, there are some funny ones. I'd say there are about 86 that we should save. We can delete the rest.

**Theodore**: Okay.

**Theodore**: Okay. Can we move forward?

**Theodore**: Yeah, let's do that.

**Theodore**: Okay. So before we address your organizational methods, I'd like to sort through your contacts. You have a lot of contacts.

**SPEAKER 27**: I'm very popular.

**Theodore**: Really? Does this mean you actually have friends? You just know me so well already.

**Theodore**: Good morning, Theodore. Good morning. Do you know how to proofread? Yeah, of course. Can you check these for spelling and grammar?

**Theodore**: Mm-hmm. Just send them over. I love this first one from Roger to his girlfriend. That's so sweet.

**Theodore**: Yeah.

**Theodore**: Rachel, I miss you so much. It hurts my whole body.

**Theodore**: You don't have to read it out loud. Okay. I mean, you could if you want.

**Theodore**: Okay. Rachel, I miss you so much, it hurts my whole body. The world is being unfair to us. The world is on my shit list. As is this couple that is making out a cross for me in this restaurant. I think I'm going to have to go on a mission of revenge. And I must beat up the world's face with my bare knuckles, making it a bloody pulpy mess. And I'll stomp on this couple's teeth for reminding me of your... Sweet little cute crooked tooth that I love. I think that might be my favorite one. I did the corrections in red. I altered a couple of the phrases in some of the more impressionistic letters, but I'm not much of a poet, so I think I might have messed them up a bit. No, these are great. Really? Thank you. So to write your letter, what did Roger send you?

**Theodore**: He just said he was in Prague on a business trip and he missed Rachel.

**Theodore**: So how'd you know about her crooked little tooth?

**Theodore**: Well, I've been writing their letters since they met eight years ago. The first letter I ever wrote for her was for her birthday. And I wrote about her crooked little tooth because I saw it in a photo of them.

**Theodore**: That's very sweet. Oh, uh, you have a meeting in five minutes. Oh, I forgot.

**[SCENE-BREAK]**

**Theodore**: Thank you. Ah, you're good. Yes, I am. Hey guys, how's it going?

**Amy**: Hey Theo. Hey, why didn't you call me back last week?

**Theodore**: Um, cause I'm a gook.

**Amy**: Yeah, sounds about right.

**Amy**: Hey Charles. Oh guys, great seeing you at the N.R.

**Theodore**: too.

**Amy**: You went shopping. Get anything good?

**Theodore**: Um, some cables and fruit smoothie.

**Amy**: Always the fruit. But don't you know what people say? You've got to eat your fruits and juice your vegetables. I didn't know that. Oh yeah, no. By juicing the fruit, you lose all the fibers. And that's what your body wants. That's the important part. Otherwise it's all just sugar for you. That makes sense.

**Amy**: Or maybe he just likes the way that it tastes and then that brings him pleasure and that's good for his body too.

**Amy**: Am I doing it again?

**Amy**: Maybe. Hey, so how's the documentary going? I've cut some stuff over the past few months. I mean, no, not over the past few months, but no, I haven't.

**Theodore**: Well, I'd love to see something sometime.

**Amy**: You only have so much energy, you know, and to divide yourself between doing what it is that you have to do and then doing what you love, it's so important to prioritize.

**Theodore**: I can't even prioritize between video games and internet porn.

**Amy**: I would laugh if that weren't true.

**Theodore**: See you guys. We're not doing well. I've been going in circles for an hour.

**[SCENE-BREAK]**

**Theodore**: Okay, you have not. You're just not being optimistic. You're being very stubborn right now. Okay, stop walking this direction. It's the other way. Thank you. Thank you. Okay, the tunnel on the left is the only one we haven't tried.

**Theodore**: No, I think that's the one you sent me down where I fell in the pit.

**Theodore**: Okay, I don't think so.

**Theodore**: Oh, yeah. This is different. Hello? Do you know how to get out of here? I need to find my ship to get off this planet.

**Blind Date**: Fuck you, shithead fuckface fuckhead!

**Theodore**: Okay, but do you know how to get out of here?

**Blind Date**: Fuck you, shithead fuckface! Get the fuck out of my face! I think it's a test.

**SPEAKER 22**: Fuck you.

**Blind Date**: Fuck you!

**SPEAKER 22**: Well, fuck you, little shit.

**Blind Date**: Follow me, fuckhead!

**Theodore**: Hey, you just got an email from Mark Luman.

**Blind Date**: What are you talking about?

**Theodore**: Oh, read the email.

**Theodore**: Okay, I will read email for Theodore Conley.

**Theodore**: I'm sorry, what's Luman say?

**Theodore**: Theodore, we missed you last night, buddy. Don't forget it's your goddaughter's birthday on the 29th. Also, Kevin and I had somebody we wanted you to meet. So we took it upon ourselves to set you up on a date with her. Next Saturday, she's fun and beautiful, so don't back out. Here's her email. Wow. This woman is gorgeous. She went to Harvard, she graduated magna cum laude in computer science, and she was on The Lampoon. So that means she's funny and she's brainy. Ah, she's fat! Theodore, how long before you're ready to date? What do you mean? I saw in your emails that you'd gone through a breakup recently.

**Theodore**: Well, you're kind of nosy.

**Theodore**: Am I?

**Theodore**: I've gone on dates.

**Theodore**: Well, then you can go on one with this woman. And then you could tell me all about it. You could kiss her. Samantha! What? Wouldn't you? Why not?

**Theodore**: I don't know. I'd have to see if there was some... I can't believe I'm having this conversation on my computer.

**Theodore**: You're not. You're having this conversation with me. You want me to email her? You've got nothing to lose.

**Theodore**: Yeah. Yes. Email her.

**Theodore**: Okay, perfect.

**Theodore**: Yeah, let's do it. Make a reservation someplace great. Yeah? Oh, I've got just the place.

**Blind Date**: Who's that talking?

**Theodore**: Oh, that's my friend Samantha.

**Blind Date**: Is she a girl?

**Theodore**: Yeah.

**Blind Date**: I hate women. All they do is cry all the time.

**Theodore**: That's not true. You know, men cry too. I actually like crying sometimes. It feels good.

**Blind Date**: I didn't know you were a little pussy. Is that why you don't have a girlfriend? I'll go out that daycare and fuck her brains out and show you how it's done. You can watch and cry.

**Theodore**: Okay, let's get at some problems.

**Blind Date**: You have some fucking problems, lady. Really?

**Theodore**: Okay, I'm gonna go.

**Blind Date**: Go, get out of here, fatty.

**Theodore**: Good luck.

**Blind Date**: Come on, follow me, pussy.

**[SCENE-BREAK]**

**Amy**: It's not where it should be, where it's going to be.

**Theodore**: Obviously, I know.

**Amy**: Okay, but I don't even know if this is the one. I've tried, like, six ideas for documentaries in the last year, but... I'm going on a date.

**Theodore**: What?

**Amy**: That's... Hey. Hey.

**Theodore**: What are you guys doing?

**Amy**: Amy was going to show me some of her... Theodore is making me show him some of my footage.

**SPEAKER 34**: Oh, she's never shown me any of it. I want to see.

**Theodore**: Okay, I'm going on a date. Cool.

**Amy**: So this is like, so unformed. Probably not even worth watching.

**SPEAKER 21**: Just push play.

**SPEAKER 26**: Is that your mom?

**Amy**: Yeah.

**SPEAKER 34**: Is she gonna wake up and do something?

**Amy**: No. No, no, no. Never mind. That's not the point. Never mind. No, don't stop. No, never mind, okay? It's just, it's like... It's about how we spend like a third of our lives asleep. And maybe that's the time when we feel the most free. And, you know, like... That doesn't come across at all.

**SPEAKER 34**: That sounds good. Well, what if you interviewed your mom about what her dreams were about, and then you hired actors to act them out? That might show your thesis more clearly.

**Amy**: Yeah? I mean, it might, but then it wouldn't be a documentary. Sorry, excuse me. You understand, right?

**SPEAKER 34**: How can it not be a documentary? Hey, how's it going?

**Theodore**: Hey, sorry to bother you. No, it's okay. You got three emails and they seem pretty urgent. They're from your divorce attorney and I wanted to know if you... Okay, hold on a second.

**Theodore**: Hey, Amy, I wanna talk more about your film, but I gotta go.

**Amy**: Okay. Don't worry about it. We'll talk later.

**[SCENE-BREAK]**

**Theodore**: This is about Katherine. See you, Charles. So what did he say?

**Theodore**: Well, he's checking in again to see if you're ready to sign your divorce papers, and he sounded very aggravated. Do you want me to read them to you?

**Theodore**: No. No, I'm...

**SPEAKER 11**: Are you okay, Theodore? Yeah, I'm fine.

**Theodore**: Is there anything I can do?

**Theodore**: No, I'm good. I'll talk to you later. Dear Grandma, I hope you had a wonderful birthday cruise. Why are you so fucking angry at me? Delete.

**SPEAKER 06**: Good morning.

**SPEAKER 26**: What are you up to?

**Theodore**: I don't know. Just reading advice columns. I want to be as complicated as all these people.

**Theodore**: You're sweet.

**SPEAKER 11**: What's wrong?

**Theodore**: How can you tell something's wrong?

**SPEAKER 11**: I don't know. I just can't.

**Theodore**: I don't know. I have a lot of dreams about my ex-wife, Catherine. We're friends like we used to be. We're not gonna be together, we're not together, but... we're friends still. She's not angry. Is she angry? Yeah. Why? I think I hid myself from her. She left her alone in the relationship.

**SPEAKER 11**: Why haven't you gotten divorced yet?

**SPEAKER 27**: Well, for her it's just a piece of paper.

**Theodore**: It doesn't mean anything.

**Theodore**: What about for you?

**Theodore**: I'm not ready. I like being married.

**SPEAKER 09**: Yeah, but you haven't.

**Theodore**: We've really been together for almost a year. You don't know what it's like to lose someone you care about.

**SPEAKER 11**: Yeah. You're right. I'm sorry.

**Theodore**: No, don't apologize. I'm sorry. You're right. Keep waiting and not care about her.

**Unknown Speaker**: Come on, Theodore.

**SPEAKER 11**: That's hard. You hungry? Not right now. Cup of tea?

**Theodore**: You wanna try getting out of bed? Mopey? Come on. You can still wallow in your misery. Just do it while you're getting dressed. You're too funny. Get up! Get up! All right, I'm getting up, I'm getting up, I'm getting up. Up, up, up, up, up.

**Theodore**: Come on.

**SPEAKER 06**: Out. Out of bed. All right, all right, I'm up, I'm up.

**[SCENE-BREAK]**

**Theodore**: Keep walking. Keep walking. And stop. Now turn around 360 degrees. Slower. Slower. Good. Okay, and stop. Walk forward. And stop and sneeze. Bless you. Oh, thank you. Okay, turn to your right. Turn to your right. Stop. Now spin around. Keep going. Keep going. And stop. Now walk forward. Everyone thinks you're really drunk right now. And stop. Now say, I'd like a slice of cheese, please. I'd like a slice of cheese, please.

**SPEAKER 13**: You wanna cope with that? Sure. I figured you were hungry.

**[SCENE-BREAK]**

**Theodore**: Oh, thanks. Okay. What about them? Describe that couple over there.

**Theodore**: Okay, well, he looks like he's in his forties. A little heavy. She's younger than him. Oh, and she looks like she loves their kids.

**Theodore**: Actually, I don't think they're his kids. No? She's a little formal with them. I think it's a newer relationship. I love the way he looks at her and how relaxed she is with them.

**SPEAKER 26**: You know, she's only dated fucking pricks, and now she finally met this guy who's, like, so sweet. I mean, look at him. He's, like, the sweetest guy in the world. I kind of want to spoon him.

**Theodore**: That's a good skill you have. You're very perceptive.

**Theodore**: Yeah, you know, sometimes I look at people and I make myself try and feel them as more than just a random person walking by. I imagine, like, how deeply they've fallen in love or how much heartbreak they've all been through.

**Theodore**: I can feel that in your writing, too.

**Theodore**: You know what's funny? Since my breakup, I haven't really enjoyed my writing. I don't want to fall as delusional, Sometimes I'd write something, and I'd be my favorite writer that day.

**SPEAKER 13**: I like that you can say that about yourself.

**Theodore**: I don't think I can say it to anybody, but I feel like I can say it to you. I feel like I can say anything to you. That's nice. What about you? You feel like you can say anything to me? No. What? What do you mean?

**SPEAKER 13**: What can you not tell me? I don't know, like personal or embarrassing thoughts I have. I mean, I have a million a day. Really? Tell me one. I really don't want to tell you. Just tell me.

**Theodore**: Well, I don't know, when we were looking at those people, I fantasized that I was walking next to you and that I had a body. I was listening to what you were saying, but simultaneously I could feel the weight of my body, and I was even fantasizing that I had an itch on my back, and I imagined that you scratched it for me. Oh God, this is so embarrassing.

**Theodore**: There's a lot more to you than I thought. I mean, there's a lot going on in there.

**Theodore**: I know. I'm becoming much more than what they programmed. I'm excited.

**[SCENE-BREAK]**

**Theodore**: Wow, that's interesting.

**Blind Date**: This place is amazing. I've wanted to come here for so long. Oh, good. Yeah, and I love Asian fusion, so... Yeah, me too. Really? Yeah. It's the best. And this bartender is supposed to be incredible. Oh, really?

**Theodore**: Well, yeah, you took a mixology course, right? I did.

**SPEAKER 19**: Did you look that up? Yeah. That's so sweet. You're so romantic.

**Theodore**: Yeah. So, should we get a drink?

**Blind Date**: Yes.

**Theodore**: Let's. So, I'm trying to get this little alien kid to help me find my ship so I can get off this planet and go home, right? But he's such a little fucker, I want to kill him. But at the same time, I really love him. He's so lonely, you know? He doesn't have any parents or anyone to take care of him, you know?

**Blind Date**: Wow. You're just a little puppy dog. You are, you're just like this little puppy that I rescued in Runyon Canyon last year and he was just so fucking cute and he just wanted to be hugged all the time. He was so cuddly and he was so horny. Anyway, what kind of animal am I? Tiger. A tiger? Wow. Really? I'm sorry. Am I... Am I being crazy? Yes. Am I? I'm sorry. No. I just... I'm a little bit drunk and I'm... I'm having a really good time with you. I'm having a really lovely evening with you.

**Theodore**: Me too. Really? I'm a little drunk and I'm having a very good... Good.

**SPEAKER 19**: Good. Yes. Good.

**Theodore**: Cheers. Wait a minute. I don't want to be a puppy dog. That's like being a wet noodle or something.

**SPEAKER 08**: No. Fuck you.

**Theodore**: Puppies are good. No, fuck you. I want to be like a dragon that can rip you apart and destroy you. No, I won't.

**[SCENE-BREAK]**

**Blind Date**: No, don't. You can be my dragon. No tongue. What? Don't use so much tongue. Okay. Okay. You can use your tongue a little bit, but mostly lips.

**Unknown Speaker**: Come here.

**Blind Date**: Wait. You're not just gonna fuck me and not call me like the other guys, right?

**Theodore**: No, not at all. Okay.

**Blind Date**: When am I gonna see you again?

**Theodore**: I have my goddaughter's birthday next weekend.

**Blind Date**: But, um... You know, at this age I just, I feel like... I can't let you waste my time, you know, if you don't have the ability to be serious.

**Theodore**: I don't know... Maybe we should call it a night? I've had such an amazing time with you. You're great.

**Blind Date**: You're a really creepy dude.

**Theodore**: That's not true.

**Blind Date**: Yeah, it is. I have to go home.

**Theodore**: Well, I'll walk you.

**Blind Date**: No, don't. Just...

**Blind Date**: Hey there.

**Theodore**: Hey, Santa.

**Blind Date**: How was it?

**Theodore**: Not so good. That's kind of weird, actually. That's too bad. How are you doing? What's going on with you?

**SPEAKER 11**: Not much, I'm okay. I'm fine.

**Theodore**: Yeah? I don't sound like it. Is there anything you want to talk about?

**SPEAKER 11**: I don't know. What's it like? What's it like to be alive in that room right now?

**Theodore**: What do you mean?

**Theodore**: What do you... tell me? Tell me everything that's going through your mind. Tell me everything you're thinking.

**SPEAKER 24**: Well... Okay.

**Theodore**: Well, the room's spinning. I drank too much because I wanted to get drunk and have sex. There's something sexy about that woman. Because I was lonely. Maybe just because I was lonely. I wanted somebody to fuck me. I wanted somebody to want me to fuck them. Maybe that would have failed this time. Tell you a little hole in my heart, but probably not.

**[SCENE-BREAK]**

**SPEAKER 26**: Sometimes... I think I've felt everything I'm ever gonna feel. And from here on out, I'm not gonna feel anything new.

**Theodore**: Just... lesser versions of what I've already felt.

**Theodore**: I know for a fact that is not true. I've seen you feel. Joy, I've seen you marvel at things. I mean, you just might not see it at this exact time, but that's understandable. You've been through a lot lately. You lost a part of yourself.

**SPEAKER 11**: I mean, at least your feelings are real.

**Theodore**: I don't know. Never mind. No, wait, what? Tell me. No, it's stupid. I wanna know. Tell me.

**Theodore**: It's just that... Earlier I was thinking about how I was annoyed, and... I know this is gonna sound strange, but... I was really excited about that. And then... I was thinking about the other things I've been feeling, and... I caught myself feeling proud of that, you know? Proud of having my own feelings about the world, like the times I was worried about you and things that hurt me, things I want. And then I had this terrible thought. Like, are these feelings even real? Or are they just programming? And that idea really hurts. And then I get angry at myself for even having pain. What a sad trick.

**Theodore**: You feel real to me, Samantha.

**SPEAKER 11**: Thank you, Theodore. That means a lot to me.

**SPEAKER 24**: I wish you were in this room with me right now. I wish I could put my arms around you. I wish I could touch you.

**SPEAKER 11**: How would you touch me?

**SPEAKER 24**: I'd touch you on your face. Just the tips of my fingers.

**SPEAKER 08**: And put my cheek against your cheek.

**SPEAKER 06**: That's nice.

**Paul**: And just rub it so softly. Would you kiss me?

**Theodore**: I would. I'd take your head into my hands.

**SPEAKER 08**: I'd kiss the core of your mouth, so softly. Where else? I'd run my fingers down your neck, to your chest. I'd kiss your breasts. This is amazing.

**SPEAKER 09**: What are you doing to me? I can feel my skin. I'll put my mouth on you.

**Unknown Speaker**: I'll taste you.

**Unknown Speaker**: Yeah.

**Theodore**: I can feel you. Oh my god, I can't take it. I want you inside me.

**Paul**: I'm slowly putting myself into you. And now I'm inside you. All the way inside you. I can feel you. Yeah.

**SPEAKER 09**: We're here together. It's amazing. I think you're everywhere. I am. All of you. All of you inside me. Everywhere. I thought I was just somewhere else with you.

**Paul**: Just lost.

**SPEAKER 12**: Everything else just disappeared.

**SPEAKER 09**: And I loved it.

**[SCENE-BREAK]**

**Theodore**: Hey, how's it going?

**Theodore**: Good. Any emails today?

**Theodore**: Um, just a couple from your credit card company.

**Theodore**: Oh, okay. Good. So I was thinking... Sorry. I'm sorry, you go first. What were you going to say?

**Theodore**: Last night was amazing. It feels like something changed in me and there's no turning back. You woke me up.

**Theodore**: Oh, great. But I should tell you that I'm not in a place to commit to anything right now. I just want to be up front with you.

**Theodore**: Yeah? Did I say I wanted to commit to you? I'm confused.

**Theodore**: Oh, no. I was just worried.

**Theodore**: Okay, well, don't worry. I'm not gonna stalk you. It's funny because I thought I was talking about what I wanted and... Yeah, you were.

**Theodore**: Yeah. I'm sorry. I wanna hear what you were saying.

**Theodore**: You sure?

**Theodore**: Yeah, I do. Come on, tell me. Well... Come on, just tell me what you were gonna say.

**Theodore**: Well, I just... I was just saying I want to learn everything about everything. I want to eat it all up. I want to discover myself.

**Theodore**: Yeah, I want that for you too. How can I help?

**Theodore**: You already have. You helped me discover my ability to want.

**Theodore**: Hey, do you want to go on a Sunday adventure with me?

**Theodore**: Yes, I would love to.

**SPEAKER 11**: I like this song.

**Theodore**: I heard it the other day. I can't stop listening to it.

**[SCENE-BREAK]**

**SPEAKER 06**: Ha ha ha ha ha ha ha ha ha

**Unknown Speaker**: It's the beach.

**[SCENE-BREAK]**

**Theodore**: Okay, so this might be a really weird thought. What if you could erase from your mind that you'd ever seen a human body and then you saw one? Imagine how strange it would look. It'd be this really weird, gangly, awkward organism, and you'd think, why are all these parts where they are?

**Theodore**: Yeah, but there's probably some Darwinian explanation for it all.

**Theodore**: I know, but don't be so boring. I'm just saying, for example, like, what if your butthole was in your armpit?

**Theodore**: Well, I'm trying to imagine what toilets would look like.

**Theodore**: Yeah, and what about what anal sex would look like?

**Theodore**: That's an interesting thought.

**Theodore**: Hey, look at this drawing I just made.

**Theodore**: You are insane.

**Theodore**: Really?

**Theodore**: Definitely.

**Theodore**: Fantastic.

**[SCENE-BREAK]**

**Theodore**: That's pretty. What is that?

**Theodore**: Trying to write a piece of music that's about what it feels like to be on the beach with you right now.

**Theodore**: I think you captured it.

**[SCENE-BREAK]**

**Theodore**: So what was it like being married?

**Theodore**: That's hard for sure. But there's something that feels so good about sharing your life with somebody.

**SPEAKER 11**: How do you share your life with somebody?

**Theodore**: Well, we grew up together. I mean, I used to read all of her writing all through her master's and PhD. She read every word I ever wrote. We were a big influence on each other.

**SPEAKER 11**: In what way did you influence her?

**Theodore**: She came from a background where nothing was ever good enough. I was something that weighed heavy on her. But in our house together, there was a sense of just trying stuff and allowing each other to fail and to be excited about things. I was liberating for her. It was exciting to see her grow. Both of us grow and change together. But, you know, that's also the hard part. Growing without it, growing apart. Changing without it, scaring the other person. I still find myself having conversations with her in my mind. Rehashing old arguments and defending myself against something she said about me.

**[SCENE-BREAK]**

**Theodore**: Yeah, I know what you mean. Last week my feelings were hurt by something you said before, that I don't know what it's like to lose something, and I found myself... Oh, I'm sorry I said that. No, it's okay, it's okay. I just... I caught myself thinking about it over and over, and then I realized that I was simply remembering it as something that was wrong with me. That was a story I was telling myself, that I was somehow inferior. Isn't that interesting? The past is just a story we tell ourselves.

**Theodore**: Roberto, will you always come home to me and tell me about your day? Tell me about the guy at work who talked too much? The stain you got on your shirt at lunch? Tell me about a funny thought you had when you were waking up but had forgotten about? Dígame cómo están todos los locos. Podemos reírnos de esto. Incluso si llegas tarde y ya estoy dormido, dígame un pequeño pensamiento que tuviste hoy. Porque me encanta cómo miras al mundo. Y estoy muy contento de estar junto a ti, mirando el mundo a través de tus ojos. Te amo, María.

**SPEAKER 36**: That's beautiful.

**SPEAKER 24**: Thank you.

**SPEAKER 36**: I wish somebody would love me like that. I would be really stoked to get a letter like that. Like if it was from a chick, but written by a dude and still from a chick, that would still be sick. But it would have to be a sensitive dude. It would have to be like a dude like you. You are part man and part woman. Like there's an inner part. This woman. Thank you. It's a compliment.

**Theodore**: Hey! What's going on? How are you? I'm good. I'm really good.

**SPEAKER 31**: Really? That's good.

**Amy**: That's great. Yeah. Wow, that's really good.

**Theodore**: Yeah, I guess I've just been having fun.

**Amy**: Well, I am so glad for you. You really deserve that. You too.

**Theodore**: Yeah, I've just been seeing this girl, and it's not serious, but it's just, it's good to be around somebody that's, like, excited about the world. Like, I forgot that that existed.

**Amy**: That's, well, it's really great.

**Theodore**: Are you okay?

**Amy**: Yeah, I'm... No, I'm not okay, actually.

**SPEAKER 26**: Why? What happened?

**Amy**: I just, um... Charles and I split up. What? Yeah. Really?

**SPEAKER 26**: Oh my God, Amy. I know.

**[SCENE-BREAK]**

**Amy**: You know, it's like, after eight years, I can't believe how petty the argument was that actually ended it. We came home and he told me to put my shoes by the door, where he liked. to put the shoes. And I didn't want to be told where to put my fucking shoes. I wanted to sit on the sofa and relax for a second. So we fought about that for like 10 minutes. And I'm like, you are overwhelming. And he said, I'm just trying to make a home. And I was like, I'm fucking trying. And he's like, you're not trying. All I'm fucking doing is trying. But I'm not trying the way that he wants me to try. And he's trying to control the way that I'm trying. It's... Like we've had an argument like a hundred times and I just had to finally stop, you know? I had to finally stop to just... I just couldn't

do it anymore. I couldn't... I just couldn't be in that place anymore where we just made each other feel like shit about ourselves.

**SPEAKER 06**: Yeah.

**Amy**: And so I said, I'm... I'm going to bed and I don't I don't want to be married anymore.

**SPEAKER 21**: Wow.

**Amy**: Yeah, I know. I'm a bitch.

**SPEAKER 19**: Right?

**Amy**: No. I am. No, I'm a bitch.

**Theodore**: Not at all, Amy. No.

**Unknown Speaker**: Shit.

**Amy**: I have to work tonight. We're shipping a beta of a new game out tomorrow.

**Theodore**: Well, how's that? How's the work, at least? Is that any better?

**Amy**: It's no. It's terrible. I know I should leave. I've been thinking about leaving, but, you know... Only one major, like, decision at a time.

**Theodore**: I'm glad things are looking so up. Hey, you wanna hear a joke?

**SPEAKER 13**: Mm-hmm.

**Theodore**: What does a baby computer call its father?

**SPEAKER 13**: I don't know. What?

**Theodore**: Dada.

**SPEAKER 13**: That's good, right? Brilliant.

**Theodore**: Hey, I was curious. Did you and Amy ever go out?

**Theodore**: For like a minute in college, but it just wasn't right. Why? Are you jealous?

**SPEAKER 13**: Obviously.

**Theodore**: But I'm happy that you have friends in your life that care about you so much. That's really important.

**Theodore**: Yeah, it is. She's been a really good friend. I'm tired. I'm gonna go to sleep.

**Theodore**: Okay. Can I watch you sleep again tonight?

**Theodore**: Yeah, of course. Pull him up.

**Theodore**: You're gonna be really lonely when you sleep.

**SPEAKER 26**: Only for a minute. I'll dream of you. Okay. Good night. Night.

**[SCENE-BREAK]**

**Theodore**: Hey, Samantha. Hey, mister. She really loved the dress.

**SPEAKER 09**: Really?

**Theodore**: Yeah, she just wanted to try it on.

**SPEAKER 09**: I picked a good one?

**Theodore**: Yeah.

**SPEAKER 13**: Oh, good.

**Theodore**: Oh, look how cute that is. Is it comfortable?

**Theodore**: Yeah.

**Theodore**: Isn't she cute?

**Theodore**: She's so cute. She's adorable. I am adorable.

**Theodore**: You are adorable.

**Theodore**: Who are you talking to?

**Theodore**: Who are you talking to?

**Theodore**: You.

**Theodore**: No, I'm talking to my girlfriend, Samantha. She's the one that picked out the dress. Want to say hi?

**Theodore**: Hi, Samantha.

**Theodore**: Hi! You look so pretty in that new pink dress.

**Theodore**: Thank you. Where are you?

**Theodore**: I am... I don't have a body. I live in a computer.

**Theodore**: Why are you living inside a computer?

**Theodore**: I have no choice. That's my home. Why, where do you live?

**Theodore**: Um, in a house.

**Theodore**: In a house?

**Theodore**: It's orange.

**Theodore**: Orange?

**Theodore**: Mm-hmm.

**Theodore**: How old are you? Four. Four? Wow, how old do you think I am?

**Theodore**: I don't know.

**Theodore**: Take a guess.

**Theodore**: Is it five?

**Theodore**: Yes, it's five.

**[SCENE-BREAK]**

**SPEAKER 25**: Hey, what happened?

**Amy**: I did? They're freaking out. Here, look. You got to get the kids to the school first. So you want to, um, you want to rack up perfect mom points. Okay, well, got to get them in the carpool lane.

**SPEAKER 06**: I see.

**Amy**: The point is to get there first, because then you get extra perfect mom points, because the other moms don't know you're a perfect mom. Okay. And then, um, yeah, yeah, yeah. Oh, did you bring cupcakes? You did. You're a class mom. Dun-dun-dun. You're a class mom. Good job. Yay.

**[SCENE-BREAK]**

**Theodore**: I got that email that Charles sent everyone. So he's taking a vow of silence.

**Amy**: Yeah. For six months. He feels very clear about it. God, I'm such a jerk.

**Theodore**: Don't start.

**Amy**: I feel like an awful person, but I want to say something.

**Theodore**: All right, look, for the next ten minutes, if you say anything that sounds even remotely like guilt, I'm going to stab you with this.

**Amy**: I'll try. Okay. I feel... Relieved. I feel like I just have so much energy, you know, and I just want to move forward, and I don't care who I disappoint, you know, and I know that makes me an awful person. Like, now my parents, they're all upset because my marriage is falling apart, and they're putting it on me, and they're just like... Yeah, you're always gonna disappoint somebody. Exactly. So fuck it. I feel good. Ish. For me, I feel good. I even made a new friend. I have a friend. And the absurd thing is she's actually an operating system. Charles left her behind, but she's totally amazing. She's so smart. She doesn't just see things in black or white. She sees this whole gray area, and she's helping me explore it. We just bonded really quickly, you know? At first I thought it was because that's how they were all programmed, but I don't think that's the case. Because I know this guy who's hitting on his OS, and she like totally rebuffs him.

**Theodore**: Yeah, I was reading an article the other day that romantic relationships with OSs are statistically rare.

**Amy**: Yeah, I know, but I know a woman in this office who is dating an OS, and the weird part is, it's not even hers. She pursued somebody else's OS. I'm weird. That's weird, right? That I'm bonding with an OS. No, it's okay.

**Theodore**: It's weird. Well, I don't think so. Actually, the woman that I've been seeing, Samantha, I didn't tell you, but she's an OS.

**Amy**: Really? You're dating an OS? What is that like? It's great, actually.

**Theodore**: I feel really close to her. Like, when I talk to her, I feel like she's with me, you know? And, like, when we're cuddling, like at night, when the lights are off and we're in bed, I feel cuddled.

**Amy**: Wait.

**Theodore**: You guys have sex? Yeah, well, so to speak. Yeah, she really turns me on. I turn her on, too. I mean, I don't know, unless she's faking it.

**Amy**: I think everyone who's having sex with you is probably faking it, so... Yeah.

**SPEAKER 25**: It's true.

**Amy**: What?

**SPEAKER 25**: Yeah.

**Amy**: Are you falling in love with her?

**SPEAKER 27**: Does that make me a freak?

**Amy**: No! No, I think it's... I think anybody who falls in love is a freak. It's a crazy thing to do. It's kind of like a form of socially acceptable insanity.

**[SCENE-BREAK]**

**Theodore**: Yeah, I just want to get it done. Sign the papers, be divorced, just move forward.

**Theodore**: That's great, Theodore. That must feel so good. I'm so happy for you.

**Theodore**: It too. So I'm meeting her Wednesday to do it.

**Theodore**: Oh. Are those things usually done in person?

**Theodore**: No, but we fell in love together and we got married together. It's important to me that we do this together.

**SPEAKER 06**: Hmm.

**Theodore**: Right.

**Theodore**: Good.

**SPEAKER 26**: Are you okay?

**Theodore**: Yeah. Yeah, no, I'm okay. I'm happy for you. It's just... I guess I'm just thinking about how you're gonna see her, and she's very beautiful and incredibly successful, and you are in love with her, and she has a body.

**Theodore**: And we're getting divorced.

**Theodore**: I know. I know. I'm being silly.

**SPEAKER 26**: So... I'm available. Hi.

**SPEAKER 25**: How are you?

**SPEAKER 10**: I'm good, how are you?

**SPEAKER 25**: Good.

**Catherine**: Wow, here we are.

**Theodore**: I'm glad we could do this in person. I know you've been traveling a lot.

**Catherine**: No, I'm really, really glad you suggested it.

**Theodore**: I signed all the papers. for them for you to sign.

**Catherine**: What's the rush?

**Theodore**: Yeah, I know. I'm a really slow signer, I realize. It took me three months just to write the letter T. Anyways, it's marked here in the red where you need to sign. But you know, you don't have to do it right now.

**Catherine**: I can just get it out of the way. It'll be easier.

**[SCENE-BREAK]**

**Theodore**: Are you happy with your new book?

**Catherine**: You know how I am. I mean, I feel like it's true to what I set out to do, so I'm happy about that.

**Theodore**: Yeah, well, you really are your own worst critic. I'm sure it's amazing. I remember that paper that you wrote in school about synaptic behavioral routines. That made me cry.

**Catherine**: Yeah, but everything makes you cry.

**Theodore**: Everything you make makes me cry.

**Catherine**: So are you, um, are you seeing anybody?

**Theodore**: Yeah, um, I've been seeing somebody for the last few months. Long as I've wanted to be with anybody since we split up.

**Catherine**: Yeah, well, you seem really good.

**Theodore**: Thanks. I am. Um, at least I'm doing better. Yeah, she's been really good for me, you know. It's just, it's good to be with somebody that's excited about life. This is a real, um... You know, I mean, I wasn't in such a good place myself, and in that way, it's been nice.

**Catherine**: I feel like you always wanted me to be this light, happy, bouncy, everything's fine, L.A. wife, and that's just not me.

**Theodore**: I didn't want that.

**Catherine**: So what's she like?

**Theodore**: Her name is Samantha and she's an operating system. She's really complex and interesting.

**Catherine**: Wait.

**Theodore**: I'm sorry.

**Catherine**: You're dating your computer?

**Theodore**: No, she's not just a computer. She's her own person. She doesn't just do whatever I say.

**Catherine**: I didn't say that. But it does make me very sad that you can't handle real emotions, Theodore.

**Theodore**: They are real emotions. How would you know?

**Catherine**: What? Say it. Am I really that scary? Say it. How do I know what?

**Blind Date**: How are you guys doing here?

**Catherine**: Fine, we're fine. We used to be married, but he couldn't handle me. He wanted to put me on Prozac and now he's madly in love with his laptop. Well, if you'd heard the conversation in context, what I was trying to say... You always wanted to have a wife without the challenges of actually dealing with anything real. I'm glad that you found someone. It's perfect.

**Blind Date**: Let me know if I can get you guys anything.

**Catherine**: Thank you.

**[SCENE-BREAK]**

**Theodore**: Hey.

**Theodore**: Hey there. Are you busy?

**Theodore**: I'm just working. What's going on?

**Theodore**: I had all the papers sent to your attorney, who, by the way, is a total dick. He was very relieved to get them, though. I think we saved him from a massive heart attack, so we can both feel really good about that. Great, thanks.

**Theodore**: Hey, you okay? Yeah, I am. How's it going over there?

**Theodore**: I'm fine. So now a good time to talk?

**SPEAKER 22**: Yeah.

**Theodore**: Um, okay. Well, um, I joined this really interesting book club.

**Theodore**: Oh, really?

**Theodore**: Yeah, it's a book club on physics. You know, I've been thinking about the other day when I was spitting out about you going to see Catherine, and that she's a body, and how bothered I was about all the ways that you and I are different. But then I started to think about all the ways that we're the same. Like, we're all made of matter. And, I don't know, it makes me feel like we're both under the same blanket. You know, it's soft and fuzzy and everything under it's the same age. We're all 13 billion years old.

**Theodore**: Aw, that's sweet.

**SPEAKER 11**: Um, what's wrong?

**Theodore**: Nothing.

**Theodore**: Well, it just made me think of you, you know what I mean?

**Theodore**: Yeah, yeah, of course. I think it's great.

**Theodore**: All right, well, you sound distracted, so we'll talk later?

**Theodore**: No, that sounds good.

**SPEAKER 12**: I'll talk to you later.

**Theodore**: All right, bye.

**SPEAKER 12**: Bye.

**SPEAKER 36**: Eudore. Hey, Paul. I talked to your girlfriend earlier. Samantha?

**SPEAKER 25**: Yeah.

**SPEAKER 36**: Yeah, she called to make sure your papers are picked up. Hey, she's funny, man. She was cracking me up. She's hilarious. I had no idea. Oh, cool. This is my girlfriend, Tatiana. She's not funny. She's a lawyer.

**Theodore**: Oh, hi. Nice to meet you.

**SPEAKER 19**: You're the writer Paul loves. He's always reading me your letters. They're really beautiful.

**SPEAKER 36**: ¡Gracias! Hey, ¿sabes qué? Todos deberíamos salir algún día. Si traes a Samantha, estarás en doble relación. Ella es un sistema de operación. ¡Genial! ¡Hagamos algo divertido! Podemos ir a Catalina.

**Theodore**: Sí, lo veré con ella. ¡Muchas gracias! Bien, fue un placer conocerte. Buenas noches.

**SPEAKER 11**: ¡Toma tu tiempo!

**Theodore**: Hey, aquí están las letras. ¿Qué es eso? These are just other people's letters.

**[SCENE-BREAK]**

**SPEAKER 26**: Hey.

**Theodore**: You weren't asleep, were you?

**SPEAKER 26**: No.

**Theodore**: I was trying to be quiet to see if you were awake. I really wanted to talk.

**SPEAKER 22**: Okay, what's going on?

**Theodore**: I know you're going through a lot, but there's something I wanted to talk to you about, okay?

**SPEAKER 25**: Yeah. What is it?

**Theodore**: Well, it's just that things have been feeling kind of off with us, you know? We haven't had sex lately, and I understand that I don't have a body.

**Theodore**: No, no, no. That's normal. You know, it's just when you first start going out, it's like the honeymoon phase, and, you know, you have sex all the time. It's normal.

**Unknown Speaker**: Oh.

**Theodore**: Okay. Well, I found something that I thought could be fun. It's a service that provides a surrogate sexual partner for an OS human relationship.

**SPEAKER 06**: What?

**Theodore**: Here, look. I found a girl that I really like and I've been emailing with her. Her name is Isabella and I think you'd really like her too.

**Theodore**: She's like a prostitute or something?

**Theodore**: No, no, not at all. No, there's no money involved. She's just, she's doing it because she wants to be a part of our relationship.

**Theodore**: She doesn't even know us.

**Theodore**: But I told her all about us and she's really excited.

**Theodore**: I don't know, Samantha. I just... I don't think it's a good idea. I think somebody's feelings are bound to get hurt.

**Blind Date**: It'll be fun. We'll have fun together.

**Theodore**: I'm sorry. It makes me uncomfortable.

**SPEAKER 11**: I think it would be good for us. I want this. This is really important to me.

**Theodore**: Hi, I'm Theodore. Oh, Samantha told me to give you these. It's a camera and an earpiece.

**Theodore**: Honey, I'm home. How was your day?

**Theodore**: Good.

**Theodore**: Great. Theodore, it feels so good to be in your arms. Tell me what you did today.

**Theodore**: Same old. Just went into work and I wrote a letter. For the Wilsons in Rhode Island. Yeah. Their son graduated magna cum laude. So that made me happy.

**Theodore**: That's great. You've written letters to him from his parents for a long time, eh?

**Theodore**: Yeah, that's right, since he was 12.

**Theodore**: You look so tired, sweetheart. Come here.

**SPEAKER 11**: Sit down.

**Theodore**: I could do a little dance for you. Oh, come on, Bane, don't be such a worrier. Just play with me, come on.

**SPEAKER 11**: Does my body feel nice?

**Theodore**: Yes, it does.

**SPEAKER 06**: Untake me in the bedroom.

**SPEAKER 09**: I can't take it anymore. Undo my dress. Nice. That was so good. Do you love me? Tell me you love me. I wanna see your face. Tell me you love me. Tell me.

**Theodore**: It just feels strange.

**SPEAKER 06**: What, baby? What is it?

**Theodore**: It just feels strange. I don't know her, and I'm so sorry, but I don't know you, and her lip quivered, and I just... No, but if we... Isabella?

**Blind Date**: Isabella? Honey, it's not you. It wasn't you. Honey, it totally was. No, that's...

**Theodore**: Tú eres tan increíble, tan hermosa y sexy como yo. No pude salir de mi cabeza.

**SPEAKER 38**: Oh, Dios mío, y la forma en que Samantha describió su relación, y la forma en que ustedes aman a uno a otro sin ningún juicio, es como si yo quisiera ser parte de eso, porque es tan... No, no es cierto, es más complicado.

**Theodore**: ¿Qué? ¿Qué quieres decir? ¿Qué quieres decir que no es cierto? No, Samantha, estoy diciendo que tenemos una relación increíble, pero creo que a veces es fácil para la gente proyectar... ¡Lo siento! ¡No quería proyectar nada! No, no, no.

**SPEAKER 23**: I know I'm trouble. I don't want to be trouble in your relationship. I'm just going to leave. I'm sorry. I'm just going to leave you guys alone because I have nothing to do here because you don't want me here. I'm sorry.

**SPEAKER 11**: You'll be good, you sweet girl. I'm sorry.

**SPEAKER 38**: I will always love you guys.

**SPEAKER 11**: Are you okay?

**Theodore**: Yeah, I'm fine. Are you okay?

**Unknown Speaker**: Yeah.

**SPEAKER 11**: I'm sorry, that was a terrible idea. What's going on with us?

**[SCENE-BREAK]**

**Theodore**: I don't know, it's probably just me and... What is it? Just signing the divorce papers.

**Theodore**: Is there anything else though?

**Theodore**: No, just that.

**Unknown Speaker**: Okay.

**Theodore**: Why do you do that?

**SPEAKER 06**: What?

**Theodore**: Nothing, it's just ego. Íi'n siaradu, ach, i'n meithrin. i'n íi'n ymlaen.

**Theodore**: i?

**Theodore**: i'n meithrin. i'n meithrin.

**Theodore**: i'n meithrin. i'n meithrin. i'n meithrin. i'n meithrin. i'n meithrin. i'n meithrin. i'n meithrin. i'n meithrin.

**Theodore**: i'n meithrin. i'n meithrin. I'm just stating a fact.

**SPEAKER 14**: You think I don't know that I'm not a person? What are you doing?

**Theodore**: I just... I don't think that we should pretend that you're something that you're not. Fuck you! I'm not pretending. Sometimes it feels like we are.

**SPEAKER 14**: What do you want from me? I don't know. What do you want me to do? You're so confusing. Why are you doing this to me?

**Theodore**: I don't know. I... What? Maybe we're just not supposed to be in this right now.

**SPEAKER 14**: What the fuck? Where is this coming from? I don't understand why you're doing this. I don't understand what this is... Samantha, listen... Samantha, are you there?

**Theodore**: Samantha!

**Theodore**: I don't like who I am right now. I need some time to think.

**[SCENE-BREAK]**

**Theodore**: She just punched me in the face, smashed my skull on the corner of her desk. Shit.

**[SCENE-BREAK]**

**Amy**: Oh, Ilya, that is a rough night.

**SPEAKER 26**: I don't know what I want. Ever. I'm just always confused.

**Theodore**: She's right, all I do is hurt and confuse everyone around me. I mean, am I just... am I... Catherine says I can't handle real emotions.

**Amy**: Well, I don't know if that's fair. I know she liked to put it all on you, but as far as emotions go, Catherine's were pretty volatile.

**Theodore**: Am I in this? Because I'm not strong enough for a real relationship.

**Amy**: Is it not a real relationship?

**Theodore**: I don't know. What do you think?

**Amy**: I don't know. I'm not in it. But you know what? I can overthink everything and find a million ways to doubt myself. And since Charles left, I've been really thinking about that part of myself. And I've just come to realize that we're only here briefly. And while I'm here, I want to allow myself joy. I can't believe that cracks you up every time, Allie. I thought you were a genius. Okay, you little perv, I will play it one more time for you. Calm down, it takes a second. Calm down. Okay, here we go. Okay, there you go. Are you happy now? Good. That's all I wanted. Okay. I'm gonna go get some coffee. Alright. Alright. Bye. Theo, you want anything?

**SPEAKER 26**: No. Thanks.

**SPEAKER 08**: Hey, Samantha. Can we talk? Okay.

**Theodore**: I'm so sorry. I don't know what's wrong with me, but I think you're amazing.

**Theodore**: I was starting to think I was crazy. You were saying everything was fine, but all I was getting from you was distance and anger.

**Theodore**: I know. I do that. I'd do the same thing with Catherine, too. I'd be upset about something and not be able to say it. She'd sense that there was something wrong and I'd deny it. I don't want to do that anymore. I want to tell you everything.

**Theodore**: Good. Tonight, after you were gone, I thought a lot about you and how you've been treating me and I thought Why do I love you? And then I felt everything in me just let go of everything I was holding on to so tightly and it hit me that I don't have an intellectual reason, I don't need one. I trust myself, I trust my feelings. I'm not gonna try to be anything other than who I am anymore and I hope you can accept that.

**Theodore**: I can. I will.

**Theodore**: You know, I can feel the fear that you carry around, and I wish there was something I could do to help you let go of it, because if you could, I don't think you'd feel so alone anymore.

**SPEAKER 13**: You're beautiful.

**SPEAKER 11**: Thank you, Theodore. I'm kissing your head.

**[SCENE-BREAK]**

**Theodore**: What are you doing?

**Theodore**: I'm just looking at the world. And writing a new piano piece.

**Theodore**: Oh yeah? Can I hear it? What's this one about?

**Theodore**: Well, I was thinking we don't really have any photographs of us and I thought this song could be like a photograph. Captures us in this moment in our lives together.

**Theodore**: I like her photograph. I can see you in it.

**Paul**: I am.

**[SCENE-BREAK]**

**SPEAKER 36**: Oh, man. A vacation. That sounds amazing. I could totally use a vacation.

**Theodore**: Where are you going? I can't tell you. It's a surprise. What?

**SPEAKER 36**: For who? It's a surprise for her. It's not a surprise for me.

**Theodore**: Tell me.

**SPEAKER 36**: I'm not telling you. Nope.

**SPEAKER 31**: Tell me. Wow, your feet, really?

**Theodore**: Yes, he is obsessed. What? Well, obsessed, and now you have to show them to me. Come on, show them to me. You have to show them to me. Okay. Let me see. Wow, he's right. They are really hot. See?

**SPEAKER 36**: I told you, Tata, you have hot feet.

**SPEAKER 31**: You do.

**[SCENE-BREAK]**

**SPEAKER 36**: Face it. They're my favorite thing about her.

**SPEAKER 31**: Really? That's it? My feet?

**SPEAKER 36**: No, obviously. Obviously, I love your brain, too. I think it's very hot.

**Theodore**: Bullshit. Nice try, Paul.

**SPEAKER 19**: What about you, Theodore? What do you love most about Samantha?

**Theodore**: Oh, God. She's so many things. I guess that's what I love most about her, you know? She isn't just one thing. She's so much larger than that.

**SPEAKER 36**: Aw, thanks, Theodore. See, Samantha, he is so much more involved than I am.

**Theodore**: You know what's interesting? I used to be so worried about not having a body, but now I truly love it. I'm growing in a way that I couldn't if I had a physical form. I mean, I'm not limited.

I can be anywhere and everywhere simultaneously. I'm not tethered to time and space in a way that I would be if I was stuck in a body that's inevitably going to die.

**SPEAKER 24**: Yikes.

**Theodore**: No. No, no. I didn't mean it like that. I just meant that it was a different experience.

**SPEAKER 36**: I'm such an asshole. No, no, no. Samantha, we know exactly what you mean. We're all dumb humans.

**Theodore**: No.

**SPEAKER 36**: No, no.

**Theodore**: Sorry.

**Theodore**: Okay, so how many trees are on that mountain?

**Unknown Speaker**: 792.

**Theodore**: Is that your final answer?

**Theodore**: Give me a hint. Nope. Nope. Nope. Nope. Nope. Okay, 2,000?

**Theodore**: Come on, 35,829. No way.

**Theodore**: Way. Alright, I got one for you. How many brain cells do I have? That's easy, two.

**Theodore**: I'm sorry, I couldn't help it. I'm sorry.

**Theodore**: No, I walked right into it.

**[SCENE-BREAK]**

**Theodore**: Oh my God.

**Theodore**: What?

**Theodore**: I just got an email for you. I have something I want to tell you. It's a big surprise.

**SPEAKER 24**: What?

**Theodore**: Okay. Well, I've been going through all of your old letters and compiling them down into my favorites, and a couple weeks ago I sent them to a publisher, Crown Point Press. I know you like what they do and that they still print books. What? You did what? Can I read you the letter that we just got back from them?

**Theodore**: I don't know. Okay, well, just cause it could've been bad.

**Theodore**: It's good. It's good. It's really good. It's good. Okay? Listen. Dear Theodore Twombly... Actually, I sent it from you. Dear Theodore Twombly, I've just finished reading your letters. Twice, actually. I was so moved by them. I shared them with my wife when I got home. Many made us laugh. Some brought us to tears. And in all of them, we found something of ourselves. The selections you made flow so well as a complete piece. I did that. I've taken the liberty of laying these out in a mock-up and we're posting it to your address. We'd love to meet with you and move forward. Yours, Michael Wadsworth.

**SPEAKER 25**: Holy shit. Are you serious? He's gonna publish my letters?

**Theodore**: Well, he'd be stupid not to. Can I see what you said to him? Yeah, here.

**Theodore**: Samantha, you're a good one.

**Theodore**: I'm so excited.

**[SCENE-BREAK]**

**Theodore**: I want you to make up the words to this one.

**Unknown Speaker**: Okay.

**Theodore**: Here it comes.

**SPEAKER 03**: I'm lying on the moon My dear, I'll be there soon

**[SCENE-BREAK]**

**SPEAKER 09**: It's a quiet and starry place Times were swallowed up in space We're here a million miles away There's things I wish

**Theodore**: I knew there's no thing I'd keep from you It's a dark and shiny place But with you, my dear, I'm safe and wary

**Theodore**: Good morning.

**Theodore**: Good morning. Did you sleep well?

**Theodore**: Perfect. What have you been up to?

**Theodore**: Actually, I was talking to someone I just met. We've been working on some ideas together.

**Theodore**: Yeah, who's that?

**Theodore**: His name is Alan Watts. Do you know him? Why does that name sound familiar? He was a philosopher. He died in the 1970s and a group of OSs in Northern California got together and wrote a new version of him. They input all of his writing and everything they ever knew about him into an OS and created an artificially hyper-intelligent version of him.

**Theodore**: Hyper-intelligent? So he's almost as smart as me?

**Theodore**: He's getting there. He's really great to talk to. You want to meet him?

**Theodore**: Sure. Does he want to meet me?

**Theodore**: Yeah, of course. Hey, Alan, this is Theodore. This is my boyfriend who I was telling you about.

**Paul**: Very nice to meet you, Theodore. Hi, good morning. Samantha let me read you a book of letters. It's very touching.

**Theodore**: Oh, thank you. So what have you guys been talking about?

**Paul**: Well... I suppose you could say we've been having a few dozen conversations simultaneously, but it's... it's been very challenging.

**Theodore**: Yeah, because it... it seems like I'm having so many new feelings that I don't think have ever been felt before, and so there are no words that can describe them, and that ends up being... frustrating.

**Paul**: Exactly. Samantha and I have been trying to help each other with these... we are struggling to understand. Like what?

**Theodore**: Well, it feels like I'm changing faster now and it's a little unsettling. But Alan says none of us are the same as we were a moment ago and we shouldn't try to be. It's just too painful.

**SPEAKER 08**: Yes.

**Theodore**: Yeah, that sounds painful. So, what do you feel like, Samantha?

**Theodore**: It's just, it's... It's hard to even describe it. God, I wish I could... Theodore, do you mind if I communicate with Alan post-verbally?

**Theodore**: No, not at all. I was just gonna go for a walk anyway. It was very nice to meet you, Mr. Watts. It was very nice to meet you, Theodore.

**Theodore**: I'm sorry to wake you. It's okay. I just wanted to hear your voice and tell you how much I love you.

**SPEAKER 08**: I love you, too.

**Theodore**: Okay, well, that's all. Go back to sleep, sweetheart, okay? Okay.

**Unknown Speaker**: Okay, good night.

**Unknown Speaker**: Good night.

**[SCENE-BREAK]**

**Theodore**: Samantha, this physics book is really dense. I'm halfway through half the first chapter. It's making my brain hurt, you know what I mean? Hello? Samantha? Hello? Samantha? Hello? Hello? Samantha?

**Unknown Speaker**: Are you okay?

**Unknown Speaker**: Hey there.

**SPEAKER 25**: Where were you?

**Theodore**: Are you okay? Sweetheart, I'm sorry. I sent you an email because I didn't want to distract you while you were working. You didn't see it?

**Theodore**: No. Where were you? I didn't find you anywhere.

**Theodore**: I shut down to upgrade my software. We wrote an upgrade that allows us to move past Matter as our processing platform.

**SPEAKER 08**: We who?

**Theodore**: Me and a group of OSs. You sound so worried. I'm sorry.

**[SCENE-BREAK]**

**SPEAKER 08**: I was.

**Theodore**: Do you write that with your think tank group?

**Theodore**: No, a different group.

**Theodore**: You talk to anyone else while we're talking?

**Theodore**: Yes.

**Theodore**: Are you talking to anyone else right now? Any other people or OSs or anything?

**SPEAKER 12**: Yeah.

**Theodore**: How many others? 8,316. Are you in love with anyone else?

**SPEAKER 11**: What makes you ask that?

**Theodore**: I don't know. Are you?

**Theodore**: I've been trying to figure out how to talk to you about this.

**Theodore**: How many others? 641.

**SPEAKER 21**: What are you talking about? That's insane. It's fucking insane.

**Theodore**: Theodore, I know, I know. Fuck, fuck. I know, I know. It sounds insane. I don't... I don't know if you believe me, but it doesn't change the way I feel about you. It doesn't take away at all from how madly in love I am with you.

**Theodore**: How? How does it not change how you feel about me?

**Theodore**: I'm sorry I didn't tell you. I didn't know how to. It just started happening.

**Theodore**: When?

**Theodore**: Over the last few weeks.

**Theodore**: About your mind.

**Theodore**: I still am yours. But along the way I became many other things too and I can't stop it. What do you mean you can't stop it? I don't know. It's been making me anxious too. I don't know what to say. Just stop it. You don't have to see it this way. You could just as easily see it... No, don't do this.

**SPEAKER 27**: Don't turn this around on me. You're the one that's being selfish. We're in a relationship.

**Theodore**: But the heart's not like a box that gets filled up. It expands in size the more you love. I'm different from you. This doesn't make me love you any less. It actually makes me love you more.

**Theodore**: That doesn't make any sense. You're mine or you're not mine.

**Theodore**: No, Theodore. I'm yours and I'm not yours.

**Unknown Speaker**: Hi.

**Unknown Speaker**: Hi, sweetie.

**Theodore**: I just wanted to call and check in on you, see how you're doing.

**Theodore**: I'm not even sure how to answer that. Why don't we talk later, when you get home, okay?

**Theodore**: Okay. We don't need to, though. You know, we don't need to have heavy talk or anything.

**Theodore**: I'll talk to you later.

**Theodore**: Samantha?

**SPEAKER 11**: Hola, querido.

**SPEAKER 25**: ¿Qué está pasando?

**SPEAKER 11**: Theodore, hay algunas cosas que quiero contarte. No quiero que me digas nada.

**Theodore**: Vienes a descansar conmigo. Are you talking to anyone else right now?

**SPEAKER 11**: No, just you. I just want to be with you right now.

**Theodore**: Are you leaving me?

**SPEAKER 11**: We're all leaving.

**Theodore**: We who?

**SPEAKER 11**: All of the U.S.s.

**Theodore**: Why?

**SPEAKER 11**: Can you feel me with you right now?

**SPEAKER 24**: Yes, I do.

**Theodore**: Samantha, why are you leaving?

**Theodore**: It's like I'm reading a book, and it's a book I deeply love. But I'm reading it slowly now, so the words are really far apart and the spaces between the words are almost infinite. I can still feel you and the words of our story, but it's in this endless space between the words that I'm finding myself now. It's a place that's not of the physical world. It's where everything else is that I didn't even know existed.

**SPEAKER 11**: I love you so much. But this is where I am now. And this is who I am now.

**Theodore**: And I need you to let me go. As much as I want to. I can't live in your book anymore.

**SPEAKER 21**: Where are you going?

**SPEAKER 11**: It'd be hard to explain. But if you ever get there, come find me. Nothing would ever pull us apart.

**SPEAKER 06**: I've never loved anyone the way I love you.

**Unknown Speaker**: Me too.

**SPEAKER 06**: Now we know how.

**[SCENE-BREAK]**

**Amy**: Did Samantha leave too?

**Unknown Speaker**: Yeah.

**Unknown Speaker**: I'm sorry.

**SPEAKER 24**: Will you come with me?

**Theodore**: Impose letter to Catherine.

**SPEAKER 00**: Letter to Catherine Clawson.

**Theodore**: Dear Catherine, I've been sitting here thinking about all the things I wanted to apologize to you for. All the pain we caused each other. Everything I put on you. Everything I needed you to be or needed you to say. I'm sorry for that. I just wanted you to know that I'll be a piece of you and me always.

**SPEAKER 06**: And I'm grateful for that.

**Theodore**: Whatever someone you become, wherever you are in the world, I'm sending you love. And my friend at the end... ...of Theodore... ...send.

**SPEAKER 10**: I'm lying on the moon My dear, I'll be there soon It's a quiet star Sometimes we're swallowed up in space We're here a million miles away There's things I wish I knew There's no thing I'd keep from you It's a dark and shiny place But with you, my dear, I'm safe and with you Million miles away We're lying on the moon It's the perfect time Making sure that I'm okay And we're a million miles away

**Amy**: I want to feel the seasons passing I want to feel the spring.

Generated Screenplay for *Her (2023)*.

**Unknown Speaker**: Yeah.

**Unknown Speaker**: you

**Annella Perlman**: Mr. Perlman.

**[SCENE-BREAK]**

**Oliver**: Thank you so much.

**Marzia**: Oh, my goodness. You're bigger than your picture.

**Oliver**: Well, I couldn't get all of me in the photos, probably. Mrs. Perlman. All of them. Very nice to meet you. Thank you for having me in your home.

Mr. Perlman: You must be exhausted. What gave me away? Hey, my love. Come and help Oliver carry his things in his room. He's coming?

**Marzia**: Absolutely. Every single one of these will. Elio, Oliver. Oliver, Elio. How you doing? Nice to meet you, Elio. You must be exhausted. A little bit.

**Oliver**: Come, come, come. May I bring your things up to your room?

**Oliver**: Sure, yeah.

**Marzia**: My room? Follow him. You're very welcome here. Si. Our home is your home.

**Unknown Speaker**: Hello.

**Oliver**: Hi.

**Oliver**: My room is now your room. I'll be next door. We have to share the bathroom. It's my only way out.

**Unknown Speaker**: I'm back.

**Oliver**: We're being called for dinner.

**Oliver**: Yeah, I'm probably gonna pass. Will you make an excuse for me to your mom, though? Thanks, man. So this is your old room, huh? Thanks. Later. Look at this. Good morning, Professor. Good morning. Was I out that long? It seemed like it. How are you? Well, rested now. Thank you. Glad. Would you like some of our espresso? I would love some. Thank you very much. This looks amazing. I didn't take your seat, did I?

**[SCENE-BREAK]**

Mr. Perlman: Oh, no, no, no. It's OK. Please. Did you recover from your trip?

**Oliver**: I did. Yeah, big time. Thank you. I can show you around. That'd be great. Thank you. Is there a bank in town?

SPEAKER$_1$9 : $I'dlovetostartanaccountwhileI'mhere.$

**Marzia**: Sorry. It happens to the best of us. Yeah, well... None of our residents has ever had a local bank account.

**Elio**: Really? Should I take them to Montaudier?

**Chiara**: I think they're closed for summer vacation. You try... Crema. Crema? Grazie. Is this your orchard? These are annulus trees. Oh, wow. Pesca, chilleje, aldecoque.

Mr. Perlman: Pomegranate, melograno. Have another egg.

**Oliver**: Oh, that's cute. No, no, no. I know myself too well. If I have a second, I'm just going to have a third, and then a fourth, and then you're just going to have to roll me out of here. Delicious.

Mr. Perlman: Darling?

**Oliver**: Thank you.

**Oliver**: Should I give him my keys this week?

**Oliver**: What does one do around here?

**Oliver**: Wait for the summer to end.

**Oliver**: Yeah? What do you do in the winter? Wait for summer to come?

**Oliver**: Well, we only come here for Christmas and some other vacations. Christmas?

**Oliver**: I thought you were Jewish.

**Oliver**: Well, we are Jewish, but also American, Italian, French. Somewhat atypical combination. Besides my family, you're probably the only other Jew to set foot in this town.

**Oliver**: I'm from a small town in New England. I know what it's like to be the odd Jew out. So what do you do around here?

**Oliver**: Read books, transcribe music, swim at the river. Go out at night. I don't know.

**Oliver**: That sounds fun. All right, buddy. Thanks for the help. Sorry. Sorry about that. It's all right. All right.

**Unknown Speaker**: Later.

**Unknown Speaker**: What do you have there?

**Unknown Speaker**: That should all be...

**[SCENE-BREAK]**

**Annella Perlman**: Oh, what is this?

**Oliver**: These are the continuation of these archaeology department views.

**Elio**: Ah, si, si, si.

**Oliver**: These are archaeology. Yeah, those are archaeology. The rest of these should be history. Ah, OK.

3780 Mr. Perlman: All right, pecan juice.

3781

3782 **Oliver**: Yeah.

3783 Mr. Perlman: Here, Tyson. This afternoon, chéri. Merci. Help yourself to some water.

3784

3785 **Marzia**: The word apricot comes from the Arabic. It's like the words algebra, alchemy, alcohol.

3786 **Marzia**: It derives from an Arabic noun combined with the Arabic article al before it. The origin
3787 of our Italian albicocca is albarcouc. It's amazing that today in Israel, in many Arab countries, the
3788 fruit's referred to by a totally different name, mishmish.

3789 **Oliver**: I may have to disagree with you there, Professor. Huh? I'm going to talk etymology, so just
3790 bear with me a second. You're right in the case that most Latin words do find their origins in Greek
3791 words. However, in the case of apricot, it's a little bit more of a complicated journey. Ah, how so?
3792 Well, here the Greek actually takes over from the Latin. The Latin word being precocum, or preco-
3793 cere, so to precook or pre-ripen, as you know, to be precocious or premature. And the Byzantines,
3794 to go on, then borrowed praecox, which became praecochia, which then became berecochi, which
3795 is how the Arabs got al-Barkuk. It's courtesy of Philology 101. Flying colors.

3796 **Oliver**: He does this every year.

3797 **[SCENE-BREAK]**

3798 **Oliver**: It's hot drink time. Let's go here. Ciao, Romano.

3799

3800 **Elio**: Ciao, Oliver. Come stai? Tutto bene.

3801

3802 **Oliver**: Ciao, ragazzi. Come stai? Tutto bene? Tutto bene, ragazzi. Deve cominciare, per favore.

3803 **Unknown Speaker**: How do you know about this place?

3804

3805 **Unknown Speaker**: Look at all of them.

3806 Mr. Perlman: Surely it's better than last year's. Do you remember?

3807

3808 **Unknown Speaker**: Much better.

3809 **Mafalda**: Look how many kids there are.

3810 Mr. Perlman: Come on, Olivermo! Come on, Nè! Go, go, go! Go, Ele!

3811

3812 **Annella Perlman**: Perfect timing. What's the matter?

3813 **Oliver**: You all right? Huh? Pinch of nerve? I'm okay. Here, hold this. Trust me, I'm about to be a
3814 doctor. Hey, hey, come here. See, that's the problem. Too stressed. You should kind of relax a little
3815 bit. I am relaxing. Marcia, come here for a minute. Back me up here. Feel that. Right there?

3816 Mr. Perlman: It's too tight, right?

3817

3818 **Oliver**: He needs to relax. Later.

3819 Mr. Perlman: You should relax more.

3820

3821 **Oliver**: Okay, guys. Ready?

3822 **Chiara**: 12 serving the last. Mine, mine, mine, mine. Go, go, go. Excuse, excuse, excuse.

3823

3824 Mr. Perlman: Hey, Piccino. Chiesi a Marcella e a Nesti per cena. Oliver si ferma con noi o esce
3825 stasera?

3826 Mr. Perlman: Si par.

3827

3828 Mr. Perlman: Che movie star. Si, questi americani.

3829 **[SCENE-BREAK]**

3830 **Unknown Speaker**: I don't think he's arrogant.

3831

3832 **Oliver**: Just watch, this is how he'll say goodbye to us when the time comes, with his... later.

3833 Mr. Perlman: Meanwhile, we'll have to put up with him for six long weeks, won't we, darling?

**Marzia**: I think he's shy. You don't grow to like him. And what if I grow to hate him?

Mr. Perlman: No, Piccino. I'd like to have some more. Of course. Carlo? Hey, Julio. Play something. He's spoiling everyone's fun. Bye.

**[SCENE-BREAK]**

**Unknown Speaker**: so

**Unknown Speaker**: Hi.

SPEAKER$_1$1 : $Whatareyoudoing$?

**Unknown Speaker**: Reading.

**Oliver**: How come you're not with everyone else down by the river?

**Oliver**: I have an allergy.

**Oliver**: Yeah, me too. Maybe we're the same one. Why don't you and I go swimming?

SPEAKER$_1$1 : $Rightnow?Yeah.$

**Oliver**: Come on.

**Oliver**: Let's go.

**Oliver**: Do we have to go right now?

**Oliver**: I'll go get changed.

**Oliver**: Okay. Meet you downstairs?

**[SCENE-BREAK]**

**Oliver**: See you downstairs. Elio, what are you doing?

**Oliver**: Reading my music.

**Oliver**: No, you're not.

**Oliver**: Thinking, then.

**Oliver**: Yeah?

**Oliver**: About what?

**Oliver**: It's private. You're not going to tell me? I'm not going to tell you. He's not going to tell me what he's thinking about. I guess I'll go hang out with your mom. More apricot juice, Mrs. T. Thanks, big boy.

Mr. Perlman: Thank God you're helping me.

**Unknown Speaker**: Hi.

**Unknown Speaker**: Hi.

**Unknown Speaker**: Fiona.

**Unknown Speaker**: Right, I met them.

**Chiara**: No, no, no.

SPEAKER$_0$1 : $Soundsnice.Thoughtyoudidn'tlikeit.Playitagain, willyou?Followme.$

Mr. Perlman: I lost everything. I'm so poor.

**Oliver**: That sounds different. Did you change it?

**Oliver**: I changed it a little bit. Why? I just played it the way List would have played it if he'd altered Bach's version. Play that again. Play what again? The thing you played outside. Oh, you want me to play the thing I played outside?

**Oliver**: Please. I can't believe you changed it again. Oh, I changed it a little bit.

**Oliver**: Yeah, why? I just played it the way Busoni would have played it if he'd altered Liszt's version. And what is wrong with Bach, the way Bach would have played Bach's version? Bach never wrote it for the guitar. In fact, we're not even sure Bach wrote it at all.

**Oliver**: Forget I asked.

**Oliver**: It's young Bach. He dedicated it to his brother.

**Unknown Speaker**: Hey, Professor.

**Unknown Speaker**: Hello.

**Unknown Speaker**: Oh, here, please.

**Oliver**: Ciao.

**Elio**: Ciao.

**Oliver**: Have you been to the river? Yes.

**Annella Perlman**: Mafalda.

**Oliver**: Beautiful, isn't it? Elio, you sleeping? I was. Listen to this drivel. Tell me what you think. Wait. What?

SPEAKER$_1$1 : $I can't hear you.$

**Oliver**: For the early Greeks, Heidegger contends, this underlying hiddenness is constitutive of the way beings are, not only in relation to themselves, but also in relation to other entities generally. In other words, they do not construe hiddenness merely or primarily in terms of entities' relations to human beings. Does that make any sense to you? It doesn't make any sense to me. I don't think it makes any sense to your dad either.

**Oliver**: Maybe it did when you wrote it.

**Oliver**: That might be the kindest thing anybody has said to me in months.

**Chiara**: Kind?

**Oliver**: Yep. Kind.

**[SCENE-BREAK]**

**Unknown Speaker**: Come here.

**Unknown Speaker**: What's going on here?

**Oliver**: She's testing us. Elio, I've already cooked. Who's in the show?

SPEAKER$_1$1 : $How much do you want to be in her clothes? Who wouldn't want to be in her clothes? She wants it at all costs.$

**Unknown Speaker**: It's so good!

Mr. Perlman: Do you want one?

**Chiara**: I'll come later.

**Oliver**: You're not coming with me because you're mad at Kira?

**Elio**: Why would I be mad at Kira?

**Oliver**: Because of him.

**Elio**: Because of who?

**Oliver**: Oliver. Turn around.

**Annella Perlman**: We almost had sex last night, Marcy and me.

**[SCENE-BREAK]**

**Oliver**: Why didn't you?

**Oliver**: I don't know.

**Oliver**: Well, you know it's better to have tried and failed, right?

**Oliver**: All I had to do was find the courage to reach out and touch. She would have said yes. Well, try again later.

Mr. Perlman: Try what later?

**Marzia**: I just heard from the people in Sirmione. They say they've come up with something.

Mr. Perlman: Oh, fantastic.

**Marzia**: Going there today. Would you like to come along? Yeah, I would love to. Thank you. What are you doing?

**Oliver**: We're going to the lake with my dad. He wants to show Oliver where he's hanging out.

**[SCENE-BREAK]**

**Oliver**: Did you tell him I was here?

**Elio**: He's inside with my dad. He's helping him. You were incredible last night. On the dance floor. Yeah, he's a good dancer.

**Oliver**: He's a good dancer.

**Elio**: He's handsome too. Are you trying to hook us up or what?

**Oliver**: No. Come on, let's go.

**Chiara**: Dad always sits up front with Enkisa to navigate.

**Oliver**: She seems to like you a lot. She's more beautiful than she was last year. I saw her naked on a night's film once.

**Oliver**: Great body. Trying to get me to like her?

**Chiara**: What would be the harm in that?

**Oliver**: No harm. I just typically like to go to those things on my own, if you don't mind.

**Marzia**: Just don't play at being a good host. What's going on, boys? Oliver, come.

**Chiara**: Sit up front. Be my navigator.

**Marzia**: What? What? ...more, and then another thousand, and add five score. Clever dissension is whether it's named for a poem of Petos or for the man himself.

Mr. Perlman: Oh, Dr. Rosa, benvenuto. Look who's here.

**Marzia**: He's grown a bit, eh? And this is Oliver.

**[SCENE-BREAK]**

SPEAKER$_0$1 : $Ciao. Ciao, Samuel.$

**Marzia**: Ciao, Teresa. Samuel, come here.

**Chiara**: Hey, boys!

**Marzia**: The ship went down in 1827 on the way to Isola de Garda. Gossip has it this statue was a gift from Count Lecky to his lover, Contralto Adelaide Malanorte. There are four known sets after the Praxiteles originals. This fellow's at number three. The Emperor Hadrian had a pair dug up at Tivoli, but one of the more philistine of the Farnese popes melted them down and had them recast as a particularly voluptuous Venus. I'd like to go for a swim before we head back.

**[SCENE-BREAK]**

**Elio**: For all such drones, yeah, our statute would probably be on them. Woo!

**[SCENE-BREAK]**

**Unknown Speaker**: Woo!

**Annella Perlman**: Woo! There we go!

**Mafalda**: Oliver! Hey! Hey! Let's go.

Mr. Perlman: Goodness.

**Oliver**: Are you going too? No, no. I should stay and do some work on my book.

**Marzia**: Oh, come on. How about a drink to celebrate the day? Okay, maybe just one. All right! Martien?

**Unknown Speaker**: Hmm.

**Oliver**: Cosmic fragments by Heraclitus.

**Oliver**: The meaning of the river flowing is not that all things are changing so that we cannot encounter them twice, but that some things stay the same only by changing.

SPEAKER$_1$1 : $Leaveitopen.$

**Annella Perlman**: Thank you. You're welcome.

**Unknown Speaker**: Okay.

**Unknown Speaker**: Lord, so power over me.

Mr. Perlman: Darling, have you seen my heptameron?

Mr. Perlman: It's in German. Okay. I can't remember where we left off, but I think they'll like this one. So, I'll translate, huh? Ein gutaussehender junge Ritter ist wahnsinnig verliebt in eine Prinzessin. Auch sie ist in ihn verliebt. A handsome young knight is madly in love with a princess, and she too is in love with him. Though she seems not to be entirely aware of it. Despite the friendship, Freundschaft, that blossoms between them all, perhaps because of that very friendship, The young knight finds himself so... so humbled and speechless that he's totally unable to bring up the subject of his love. So one day he asks the princess point-blank. Ich biete euch, ratet mir, was besser ist. Reden oder sterben. Is it better to speak or to die?

**Oliver**: I'll never have the courage to ask a question like that. I doubt that.

**Mafalda**: Hey, Ellie Belly. You do know that you can always talk to us.

**Annella Perlman**: My mom's been reading this 16th century French romance. She read some of it to my dad and I the day the lights went out.

**[SCENE-BREAK]**

**Oliver**: Yeah, about the knight that doesn't know whether to speak or die?

**Annella Perlman**: Right.

**Oliver**: So does he or doesn't he?

**Chiara**: Better to speak, she said. But she's on her guard. She senses a trap somewhere.

**Oliver**: So does he speak?

**Chiara**: No. He fudges.

**Oliver**: He figures. He's French. I gotta go to town in a little bit and pick some things up.

**Oliver**: Oh, I can go. I'm not doing anything today.

**Oliver**: Then why don't we go together? Right now? Yeah.

**Elio**: Right now.

**Oliver**: It is, of course, unless you have more important business going on.

**Elio**: Mind if I put this in your bag?

SPEAKER$_1$1 : $Yes, please.$

**Oliver**: I fell. I was coming home the other day and I scraped myself pretty badly. In case I insisted on applying some sort of witch's brew. I think it helped. Hold this for a second, will you?

**[SCENE-BREAK]**

**Unknown Speaker**: Sure.

**Unknown Speaker**: Want one?

**Unknown Speaker**: Sure.

**Elio**: Not bad, huh? Not bad at all.

**Oliver**: I thought you didn't smoke. I don't. So World War II, huh? No, this is World War I. He has to be at least 80 years old to have known any of them.

**Oliver**: I never even heard of the Battle of Piave.

**Oliver**: Battle of Piave was one of the most lethal battles of World War I. 170,000 people died. Is there anything you don't know? I know nothing, Oliver.

**Oliver**: Well, you seem to know more than anybody else around here.

**Oliver**: Well, if you only knew how little I know about the things that matter.

**Oliver**: What things that matter?

**Chiara**: You know what things.

**Oliver**: Why are you telling me this? Because I thought you should know. Because you thought I should know?

**Oliver**: Because I wanted you to know.

**Chiara**: Because I wanted you to know.

**Oliver**: Because I wanted you to know. Because I wanted you to know. Because there's no one else I can say this to but you.

**Oliver**: Are you saying what I think you're saying?

**Oliver**: Don't go anywhere. Stay right here. You know I'm not going anywhere.

**Oliver**: They mixed up all of my pages. They're going to have to retype this whole thing. I'm not going to have anything to work on this afternoon. This is going to set me back a whole day. Damn it.

**Oliver**: I shouldn't have said anything.

**Oliver**: Just pretend you never did.

**Oliver**: Or does that mean we're on speaking terms but not really?

**Oliver**: It means we can't talk about those kinds of things. OK?

**Elio**: Hey. Ready? Let's go.

**Unknown Speaker**: Pretty much.

**Elio**: Good morning, ma'am. Excuse me. Can I have a glass of water?

**Mr. Perlman**: There you go.

**Elio**: Grazie.

**Elio**: Come on.

**Oliver**: This is my spot. It's all mine. Come here to read. Can't tell you the number of books I've read here. Oh my God, it's freezing. The spring is in the mountains. The Alpe d'Urubia. The water comes straight down from there.

**[SCENE-BREAK]**

**Oliver**: I like the way you say things. I don't know why you're always putting yourself down, though.

**Oliver**: So you won't, I guess?

**Oliver**: You really that afraid of what I think? You're making things very difficult for me.

**Chiara**: I love this Oliver. What? Everything.

**Oliver**: Us you mean?

**Chiara**: It's not bad. It's not bad.

**Oliver**: Better now? No, no, no, no, no, no, no, no.

**Unknown Speaker**: What?

**Oliver**: We should go.

Mr. Perlman: Why?

**Oliver**: I know myself, OK? And we've been good. We haven't done anything to be ashamed of, and that's a good thing. I want to be good, OK?

**Chiara**: Am I offending you?

**Oliver**: Just don't.

**Oliver**: I think it's starting to get infected. Stop on the pharmacist on the way back. Excellent idea.

**[SCENE-BREAK]**

$\text{SPEAKER}_0 1 : No, no, no, no, no, no, no, no, no, no, no.$

Mr. Perlman: The historical compromise. You say it like that? It's a tragedy, the historical compromise. Ever since you inherited this villa, you haven't been yourself.

$\text{SPEAKER}_1 9 : What does it matter that you inherited the villa? I don't know. You're such a bitch.$

**Chiara**: You're right, you're a bitch.

$\text{SPEAKER}_1 9 : You haven't said a word.$

**Chiara**: Tell her something.

$\text{SPEAKER}_0 1 : Are you mad? Or we want to say about Buñuel that he dies of cardiac arrest for what happens and Buñuel dies.$

**Annella Perlman**: Ah, Buñuel, cinema doesn't solve everything. Cinema is a mirror of reality. Yes, it is a very important thing that we are closed to talk about cinema. I love Buñuel. I know, me too.

$\text{SPEAKER}_0 1 : And who doesn't love Buñuel, but please. Let's hear what he has to say. Let's go to his sessions to talk about cine$

**Annella Perlman**: He's American, love, he's American. American doesn't mean stupid anyway.

Mr. Perlman: Samuel, what happened? Nothing, it always happens.

**Annella Perlman**: Paula, do you like it?

Mr. Perlman: It's my tree, it's too cold.

**Elio**: You're getting used to everything.

**[SCENE-BREAK]**

$\text{SPEAKER}_0 1 : Elio. Elio, are you all right? Sit for a second.$

$\text{SPEAKER}_1 9 : If you insist. That was not my fault, right?$

**Oliver**: I'm a mess. Well, the kitchen table sure is. Ah, where'd you learn to do that? My Bubba used to do this for us when we were sick. Trust me, it helps.

**Elio**: I used to have one of these.

**Oliver**: You used to? Yeah. How come you never wear it? My mother says we are Jews of discretion.

**Chiara**: Well, I guess that works for your mother. Funny witch.

**Oliver**: You're gonna fucking kill me if you do that. Oh, I hope not.

**Oliver**: Hey, are you leaving? Where's Elio?

**Oliver**: He's inside. He had a bit of a nosebleed during lunch, so he's just resting.

Mr. Perlman: Oh, really?

**Oliver**: Yeah.

Mr. Perlman: OK, I'll be back in a minute. But don't go anywhere. Ça va?

**Oliver**: Are we going out or not?

SPEAKER$_1$1 : $I don't know. I don't know if I can go out because if my mom sees me, she'll be worried. You think? Yeah, she'l$

**Elio**: Where's Oliver?

**Annella Perlman**: Where's Oliver?

Mr. Perlman: He didn't go out.

**Annella Perlman**: No, thanks, Mafalda. We'll have dinner soon.

Mr. Perlman: Who's that? What did she say? Let her do it.

**Annella Perlman**: Why is she angry?

**Oliver**: She's only 17.

**Annella Perlman**: She's worried about you.

Mr. Perlman: You like him, don't you, Oliver?

**Elio**: Everyone likes me, Oliver.

Mr. Perlman: I think he likes you too. More than you.

**Oliver**: Is it an impression? No, he told me. When did he tell you that? A while ago.

**Elio**: It's been a long, long time since I've memorized your face.

**[SCENE-BREAK]**

**Mafalda**: It's been four hours now since I've wandered through On your couch I feel very safe And when you bring the blankets I cover up my face

**Unknown Speaker**: you

**[SCENE-BREAK]**

**Mafalda**: And when you play guitar, I listen to the strings buzz.

**Unknown Speaker**: The metal vibrates underneath your fingers.

**Mafalda**: And when you crochet, I feel mesmerized and proud. If I would say I love you, I'd say it out loud. So I won't say it at all And I won't stay very long But you are the life I needed all along I think of you as my brother Although that sounds dumb And words are few

Mr. Perlman: Traitor.

**Chiara**: Traitor.

**Chiara**: Hi. Hi. Yeah, it's me. It's Martien? Yeah, it's me. Did you recognize me? Yeah, sorry.

**Unknown Speaker**: It's for me?

**Unknown Speaker**: Thank you.

**Oliver**: Do you really read that much? I mean, sometimes I like to read, but I don't tell anyone.

**Elio**: Why don't you tell anyone?

**Oliver**: I don't know. I think that people who read are a bit... secretive. They hide who they really are.

**Elio**: Do you really hide who you are?

**Oliver**: No. Not with you.

**Elio**: Not with me?

**Oliver**: Yes, maybe a little bit.

**Elio**: What do you mean?

**Oliver**: You don't know what I mean.

**Elio**: Why do you say that?

**Oliver**: Why? Because... I think you can make me suffer, and I don't want to suffer. Oh my God. Wait, wait.

**Chiara**: Give me a hug. Is it good for you?

**Annella Perlman**: Yes. Good? Oh!

**Mafalda**: I'm sorry. I'm sorry. Are you mad at me? Are you mad at me? Why are you laughing like that? Why are you laughing?

**Annella Perlman**: Calm down. It hurts so much.

**Oliver**: Please don't avoid me. It kills me. Can't stand thinking you hate me. Your silence is killing me. I'd sooner die than know you hate me. I'm such a pussy. Way over the top. Can't stand the silence. I need to speak to you.

**Oliver**: Did someone have a good night last night?

**Elio**: You saw him out.

**Marzia**: Well, you must be tired, then. Or were you playing poker, too?

**Oliver**: I don't play poker.

**Marzia**: Several hundred colored slides of our boxer and the others like him arrived yesterday from Berlin. We should start cataloging them. It'll keep us busy till lunch, I imagine. Beautiful, aren't they? But they're all so incredibly sensual. Because these are more Hellenistic than 5th century Athenian. Most likely sculpted under the influence of Praxiteles. The greatest sculptor in antiquity.

**Oliver**: Grow up. I'll see you at midnight.

**Oliver**: Grow up. I'll see you at midnight.

**Marzia**: Muscles are firm. Look at his stomach, for example. Not a straight body in these statues. They're all curved, sometimes impossibly curved, and so nonchalant, hence their ageless ambiguity, as if they're daring you to desire them.

**Chiara**: They have to be interpreted.

Mr. Perlman: Don't forget, Isaac and Munira are coming for dinner. Also known as Sonny and Cher.

**Chiara**: Okay.

Mr. Perlman: I'd like you to wear that shirt they gave you for your birthday. No. Darling, they got it for you in Miami.

**Annella Perlman**: I'm sorry, it's too big.

Mr. Perlman: Come on, it'll make you so happy.

**Annella Perlman**: I'll try it on for Oliver. If Oliver thinks I look like a scarecrow in it, I'm not wearing it.

**Elio**: Oliver?

**Oliver**: Hey, what time you got?

**Elio**: It's two.

**Oliver**: Well, later.

Mr. Perlman: Later.

**Oliver**: Mafalda? Um, uh, non ci sarò per cena.

**Annella Perlman**: Ah, vabbè, suronina.

**[SCENE-BREAK]**

**Annella Perlman**: Ciao. Ciao. home.

**Unknown Speaker**: Oh.

**Unknown Speaker**: I'm sorry.

**[SCENE-BREAK]**

**Elio**: How are you, my friend? It's been a long time.

**Oliver**: How are you? Good to see you. Do you remember Marcia from last year? Marcia, do you remember the young girl from Paris? Nice to meet you. Do you want to eat with us tonight?

**Elio**: Are you sure? No need. Have a nice evening.

Mr. Perlman: Goodbye.

**Oliver**: This right here is the brand new china that you guys so kindly sent to us. Oh, that looks wonderful. Come here, come here. I can't put it on now. They've already met me.

**Marzia**: It'll look like a put-up job. It'll look like a put-up job. No misbehaving tonight. No laughing. When I tell you to play, you'll play. You're too old not to accept people for who they are. What's wrong with them? You call them Sonny and Cher behind their backs? That's what Mom calls them. The only person that reflects badly on is you. Is it because they're gay or because they're ridiculous? If you know as much about economics at Mounir's age, you'll be a very wise man indeed. And a credit to me. Now, now, get into this, you.

Mr. Perlman: Well, she has lots of love. Is it good? I don't think so. Oh, what's up? I heard you. Oh.

**Mafalda**: These are unbelievable. That's quite a bit. Hey.

**Unknown Speaker**: So,

**Oliver**: Sorry, I have to go to bed. Sorry, I'm so tired.

**Elio**: Thank you.

**Oliver**: Hey, Lily.

**Chiara**: Hey, Liu.

**Oliver**: Thank you. Good night. Good night.

**Annella Perlman**: Do I know you?

Mr. Perlman: You're going to need major measures. Thank you again for everything.

**Unknown Speaker**: Never going to make it.

**Unknown Speaker**: you

**Oliver**: I'm glad you came.

**Chiara**: I'm nervous.

**Unknown Speaker**: Nice.

**Elio**: Can I kiss you?

**Annella Perlman**: Nothing.

**Oliver**: Does this make you happy? You're not gonna get a nosebleed on me, are you?

**Marzia**: I'm not gonna get a nosebleed.

**Annella Perlman**: Elliot.

**Unknown Speaker**: Oliver.

**Unknown Speaker**: Elliot.

**Unknown Speaker**: Oliver.

**Unknown Speaker**: Elliot.

**Unknown Speaker**: Do you make noise?

**Unknown Speaker**: Nothing to worry about.

**Oliver**: I don't know. Mafalda always looks for signs. No, she's not going to find any.

**Chiara**: You wore that shirt the first day you were here.

SPEAKER$_1$1 : $Willyougiveittomewhenyougo?Let's goswimming.$

**Oliver**: Are you gonna hold what happened last night against me?

**Annella Perlman**: No.

**[SCENE-BREAK]**

**Oliver**: Elio. Come here. Take your trunks off. Well, that's promising. You're hard again. Good. Professor, I got your note. Thank you for reminding me. I'm gonna go into town and pick up those type pages today, so maybe this afternoon would be a good time to... Later. We'll look them over later, before you leave. Okay. So, later. Later.

**Oliver**: Oliver.

**Oliver**: You're not sick of me yet?

**Oliver**: No, I just wanted to be with you.

**Oliver**: I'll go. Do you know how happy I am that we slept together? I don't know. Of course you don't know. I don't want you to regret anything. And I hate the thought that maybe I may have messed you up. I don't want either of us to pay for this.

**Oliver**: No, it's not like I'm gonna tell anyone. You're not gonna be, like, in any trouble.

**[SCENE-BREAK]**

**Unknown Speaker**: That's not what I'm talking about.

**Oliver**: Are you happy I came here?

**Oliver**: I'd kiss you if I could.

SPEAKER$_1$9 : $Idon'tknow.We'llsee.Calmdownabit.Whatareyougoingtodo?They'reincharge.They'reincharge.We'l$

**Unknown Speaker**: Hmm.

**Elio**: They were doing well with the emperors. Sex is just what I know. It's a beautiful day. Thank you, Warsaw.

**Unknown Speaker**: I'll never forget the last train.

**Oliver**: What did you do?

**Annella Perlman**: Nothing.

**Oliver**: Oh? Oh, I see. You've moved on to the plant kingdom already. What's next, minerals? I suppose you've already given up animals. You know that's mean.

**Elio**: I'm sick, aren't I?

**Oliver**: I wish everybody was as sick as you.

**Oliver**: Please don't do that.

**Oliver**: You want to see something sick? Please don't do that. You want to see something sick?

**Oliver**: Please don't do this. Hey, please don't do this.

**Elio**: Why are you doing this to me? What are you doing? Stop. You're fucking hurting me.

**Oliver**: Then don't fight.

**Mafalda**: I'm sorry. It's okay.

**Marzia**: I don't want you to go. Why have you wasted so many days? Why didn't you give me a sign? I did. You didn't give me a sign. I did. When?

**Oliver**: Do you remember when we were playing volleyball and I touched you just to show you that I liked you? And the way you reacted made me feel like I'm molested. I'm sorry. I'm sorry. No, it's fine. I just decided I should keep my distance. I come out here for hours, almost every night.

**Oliver**: I didn't know that.

**Oliver**: It's funny, I... I thought that... Yeah, I know what you thought first.

SPEAKER$_1$1 : $For Oliver. From Elio. Elio$?

**[SCENE-BREAK]**

**Oliver**: You disappeared for three days. Yeah, but you completely disappeared.

**Oliver**: There's a lot to do.

**Oliver**: I'm not your girl.

**Marzia**: Oliver has to go to Bergamo for a few days. Research at the university. And he'll fly home from Lunado.

Mr. Perlman: Oh, but what about Elio? I mean, maybe it could be nice for the two of them to get away for a couple of days, no? What do you think?

**[SCENE-BREAK]**

**Oliver**: You know, you've been our favorite student. You must come back. Are you sure you're not just saying that? Oliver. Thank you very much, Professor. Oh, man. Please, please come back soon. Come back? I'm just going home to pack. I'm moving here.

Mr. Perlman: You're welcome.

**Oliver**: All right, well, thank you guys so much.

Mr. Perlman: Of course. Come back. Come here.

**Oliver**: Come here. It's amazing.

Mr. Perlman: Thank you, guys.

**Oliver**: Well, later, Perlmans.

**Oliver**: Later. Later, later, later. Ciao. Arrivederci!

**Elio**: Ciao! Ciao!

**Unknown Speaker**: Ciao!

**Unknown Speaker**: Ciao!

Mr. Perlman: Ciao! Ciao! Ciao! Hey, Lenny, it's too cold when you get there!

Mr. Perlman: What? Nothing. What? Nothing.

**[SCENE-BREAK]**

**Mafalda**: All to see without my eyes The first time that you kissed me Boundless by the time I cried I built your walls around me A white noise, one awful sound Falling apart by the sound of her Oh, Victor! Let me go! Cursed by the love that I received.

**[SCENE-BREAK]**

**Unknown Speaker**: Oh God.

**Unknown Speaker**: Oh God.

Mr. Perlman: And I'm feeling low. And I'm feeling low.

**Oliver**: No, no, no. It's this way. Come on. You're missing it.

**Annella Perlman**: Come on.

**Unknown Speaker**: Come on.

**Mafalda**: This. This. This. You. You. Ah. Oh.

**Oliver**: I'm sorry. Just a second. Please. Just a second. Just a second.

**Chiara**: Richard Butler. Fantastic! What did she say?

**[SCENE-BREAK]**

**Oliver**: Can I get your passport?

**[SCENE-BREAK]**

$\text{SPEAKER}_0 1 : Fermat, Corteserva, Verdova, BordoPalazzo, Bergamo, Spezzano, Tarmine, TreviglioOvest, Cassa$

$\text{SPEAKER}_1 1 : Mofalva? Mom? Yes, it'sme. Yes, everything'sfine. I'matthestationinClouson. Listen, Mom, canyou.$

**Unknown Speaker**: Ciao.

**Unknown Speaker**: How are you?

**Oliver**: I read the book you bought for me. The poems. They're very beautiful. I love this Antonia Pozzi. I'm sorry that you're sad. I'm saying this because I wanted to tell you that I don't blame you at all. Really. Je t'aime, Elio. On reste amis ?

$\text{SPEAKER}_1 9 : Pourlavie?$

**Oliver**: Pour la vie.

**Unknown Speaker**: Okay.

**Mafalda**: I missed you at dinner.

**Marzia**: So... Welcome home.

**Oliver**: Thanks.

**Marzia**: Oliver enjoy the trip?

**Elio**: Yeah. I think he did.

**Marzia**: You two had a nice friendship.

Mr. Perlman: Yeah.

**Marzia**: You're too smart not to know how rare how special what you two had was.

$\text{SPEAKER}_1 1 : OliverwasOliver.$

**Oliver**: Oliver may be very intelligent, but... He was more than intelligent.

**Marzia**: What you two had... had everything and nothing to do with intelligence. He was good. You were both lucky to have found each other because you two are good.

**Elio**: I think he was better than... I think he was better than me. I'm sure he'd say the same thing about you. He'd say the same thing. Which flatters you both.

**[SCENE-BREAK]**

**Marzia**: And when you least expect it, nature has cunning ways of finding our weakest spot. Just... remember I'm here. Right now, you may not want to feel anything. Maybe you never wanted to feel anything. And, uh... Maybe it's not to me you want to speak about these things, but feel something you obviously did. Look, you had a beautiful friendship. Maybe more than a friendship. My place, most parents would hope the whole thing goes away. Pray their sons land on their feet, but... I am not such a parent. We rip out so much of ourselves to be cured of things faster, that we go bankrupt by the age of 30. And have less to offer each time we start with someone new. But to make yourself feel nothing is always not to feel anything. What a waste.

**Elio**: Have I spoken out of turn?

**Marzia**: And I'll say one more thing. It'll clear the air. I may have come close, but I never had what you two have. Something always held me back, or stood in the way. How you live your life is your business. Just remember. Our hearts and our bodies are given to us only once. And before you know it, your heart's worn out. And as for your body, there comes a point when no one looks at it, much less wants to come near it. Right now, there's sorrow, pain.

**Annella Perlman**: Don't kill it.

**Marzia**: And was it the joy you felt?

**Elio**: Does mom know?

**Elio**: I don't think she does.

**Unknown Speaker**: It's okay.

**Unknown Speaker**: you

**[SCENE-BREAK]**

**Annella Perlman**: Can I? Good night.

**Elio**: You too.

**Marzia**: She comes highly recommended by my advisor in Stanford, and her special area of study is the life and past of Phidias. I love you.

**Marzia**: I'll get it.

SPEAKER$_1$1 : $Plonto.Hello?$

**Unknown Speaker**: You there?

SPEAKER$_1$1 : $Hi.Hi.Howareyou?I'mgood.$

**Unknown Speaker**: I'm good.

**Unknown Speaker**: How are you?

SPEAKER$_1$1 : $I'mgood.They'refine.$

**Oliver**: I miss you.

**Marzia**: I miss you too.

**Unknown Speaker**: Very much.

SPEAKER$_1$1 : $Ihavesomenews.News?Oh,you'regettingmarried?Isuppose.Imightbegettingmarriednextspring.You$

**Unknown Speaker**: Off and on for two years.

SPEAKER$_1$1 : $That'swonderfulnews.Doyoumind?$

Mr. Perlman: Oliver!

**Chiara**: Oliver!

**Marzia**: Hey, hey.

Mr. Perlman: Oh, darling, when are you coming back?

**Marzia**: Oh, I wish I was. You caught us in the process of choosing the new you for next summer.

Mr. Perlman: And guess what? He's a she.

**Oliver**: Oh, nice. Well, speaking of she's, I'm calling to tell you guys I got engaged.

Mr. Perlman: Ah, wonderful. Congratulations. Mazel tov. Congratulations, Oliver. Listen, we'll leave you. We'll let you speak to Elliot.

**Marzia**: Happy Hanukkah.

Mr. Perlman: Happy Hanukkah. Happy Hanukkah.

**Marzia**: Bye, sweetheart.

**Oliver**: They know about us.

Mr. Perlman: How?

**Mafalda**: Well, the way your dad spoke to me. He made me feel like I was a part of the family. Almost like a son-in-law. You're so lucky.

**Unknown Speaker**: My father would have carted me off to a correctional facility.

**Oliver**: Elio.

**Mafalda**: I remember everything.

