# OpenReview forum: "ScreenWriter: Automatic Screenplay Generation and Movie Summarisation"
_ICLR.cc/2025/Conference — Submitted to ICLR 2025_

### Official Review · Reviewer_47gi · 2024-10-28

**Soundness:** 2
**Presentation:** 2
**Contribution:** 2
**Rating:** 3
**Confidence:** 4

**Summary:**

This paper introduces ScreenWriter, a framework that generates screenplays and movie summaries directly from video and audio without relying on text-based transcripts. Designed to address the growing demand for video summarization, ScreenWriter overcomes limitations of existing methods that require screenplay or transcript data by operating solely on video content.  The framework includes several modules: 1. Scene Detection algorithm 2. Character Name Assignment algorithm 3. Screenplay and Summary Generation.

**Strengths:**

1. The paper proposes a scene detection algorithm to divide long video sequences into chunks. It segments a sequence of visual feature vectors into scenes by minimizing a cost function based on the minimum description length principle, encoding each scene’s vectors relative to their mean, and using dynamic programming to efficiently find the optimal partition.

2. The paper proposes a character face assignment algorithm to associate specific character names with the screenplay, incorporating face similarity between a character bank and scene information.

**Weaknesses:**

1. Most components of the proposed framework are adopted from existing works, lacking academic and technological contributions. There is almost no insight provided on summarizing long-context videos using a “screenplay.”

2. The paper lacks an ablation study on the effect of different model sizes/types on the summarization results.

3. The screenplay generation process seems somewhat unnecessary. If the transcript is summarized and then processed by the summarizer alongside visual information in parallel, why is a screenplay still needed?

4. The baseline selection in the experiment lacks justification. For face assignment, randomly assigning or commonly assigned methods are not ideal choices. For scene detection, there is no comparison with classical tools like SceneDetection.

5. There are existing video summarization frameworks, such as MM-VID[1], MM-Narrator[2] and MM-ScreenPlayer[3], that show high similarity in their pipelines. A discussion of these works should be included.

6. Some typo:
    - 'sctructure' in line 67 should be 'structure'
    - 'movide' in line 317 should be 'movie'


Reference:

[1]Lin, Kevin, et al. "Mm-vid: Advancing video understanding with gpt-4v (ision)." arXiv preprint arXiv:2310.19773 (2023).

[2]Zhang, Chaoyi, et al. "Mm-narrator: Narrating long-form videos with multimodal in-context learning." Proceedings of the IEEE/CVF Conference on Computer Vision and Pattern Recognition. 2024.

[3]Wu, Yongliang, et al. "Zero-Shot Long-Form Video Understanding through Screenplay." arXiv preprint arXiv:2406.17309 (2024).

**Questions:**

see weakness

---

> ### Author Response · Authors · 2024-11-23
>
> Thank you for your review. Here are our responses.
>
> **Most components of the proposed framework are adopted from existing works, lacking academic and technological contributions.**
>
> The scene-break detection algorithm and the algorithm for assigning character names to speaker IDs, which are essential components of our model, are both highly novel.
>
> **The screenplay generation process seems somewhat unnecessary. If the transcript is summarized and then processed by the summarizer alongside visual information in parallel, why is a screenplay still needed?**
>
> The transcript identifies speakers only by a randomly generated ID, e.g. “SPEAKER08: Now it places the lotion in the basket”, whereas the automatic screenplay we produce has character names, e.g. “Jame Gumb: Now it places the lotion in the basket” (from Fig 3). Additionally, the transcript does not include scene breaks, which are needed for hierarchical summarisation. The benefits both of scene breaks and name assignments are shown empirically in Table 4.
>
> **Additional baselines are needed, such as SceneDetection.**
> Thank you for this suggestion. We have added SceneDetect and another existing method as comparisons for the scene segmentation results.
>
> |                | acc     | ari     | nmi     |
> |----------------|---------|---------|---------|
> | unif-60        | 0.461   | 0.278   | 0.720   |
> | unif-75        | 0.441   | 0.244   | 0.712   |
> | unif-90        | 0.430   | 0.216   | 0.704   |
> | unif-oracle    | 0.436   | 0.228   | 0.710   |
> | yeung96        | 0.321   | 0.220   | 0.541   |
> | scenedetect            | 0.394   | 0.061   | 0.596   |
> | **ours**       | **0.564** | **0.375** | **0.746** |
>
> Table 1 in the paper has been updated with these results.
> We are working on adding additional baselines and will update the paper as soon as they are ready.
>
> **A discussion should be included of MM-VID, MM-Narrator and MM-ScreenPlayer.**
>
> We have added this to the related work section.
>
> **Some typo: 'sctructure' in line 67 should be 'structure', 'movide' in line 317 should be 'movie'.**
>
> Thank you, we have fixed these.

---

> > ### Comment · Reviewer_47gi · 2024-11-25
> > **Reply to Aurthor**
> >
> > For question 1, the ID assignment operation is common in a lot of work related to script generation/summarization.
> >
> > For question 2, my concern is that, after you generate the screenplay, the different part of the screenplay is separated and processed to get the summarization, then why don't you just directly send each part to the post-processing function? I can't tell the meaning of the middle state of the screenplay.
> >
> > For question 3, the result seems to lack convincing, since the directly uniformly sampled method can beat the previous SOTA scene detection method by a large margin, and the uni-oracle performer is worse than the uni-60 method, which lacks analysis.

---

> ### Comment · Reviewer_47gi · 2024-11-25
> **Leak of Aurthor information**
>
> It's confusing that In the updated version of the paper, the author directly includes the name in the paper. This violates the double-blind rule.

---

> > ### Author Response · Authors · 2024-11-26
> >
> > I very much apologize for this. It seems I accidentally uncommented the relevant line in the preamble while adding new packages to display the additional results. I fixed this as soon as I saw it, and the names are anonymized again.

---

> > > ### Author Response · Authors · 2024-11-26
> > >
> > > > the ID assignment operation is common in a lot of work related to script generation/summarization.
> > >
> > > If I understand correctly, you mean that the task of assigning names to speaker IDs has been considered by others. Let me clarify that we are not claiming the task is novel, but that our solution is novel.
> > >
> > > > I can't tell the meaning of the middle state of the screenplay.
> > >
> > > There could be uses for the screenplay such as video QA or plot analysis. In the present paper, we choose summarisation as a downstream task with which to evaluate the accuracy of the screenplay.
> > >
> > > > Confusing that the directly uniformly sampled method can beat the previous SOTA scene detection method by a large margin, and the uni-oracle performer is worse than the uni-60 method
> > >
> > > The explanation for the first point is that the two existing methods are quite inaccurate at predicting the number of scenes. SceneDetect in particular tends to predict a few hundred per movie, and this is after some tuning of the threshold parameter, for the default value it predicts close to 1000 per movie. Yeo96 is slightly better, but still significantly overestimates the number of scenes. The uniform baselines, on the other hand, are constrained to have reasonable numbers of scenes, in the same ball-park as the ground truth, which helps improve its scores. Determining the number of scenes is a difficult part of the scene detection task, and one of the advantages of our method is that it can do this more accurately than existing methods without even having any parameters to tune.
> > >
> > > For the second point, we believe the explanation is that, if partition predictions are generally of low accuracy, then predicting a smaller partition tends to minimize inaccuracies and produce a slightly higher score. The mean true number of scenes is around 90, so unif-90 predicts more than unif-60. Our method predicts 53 on average. Most of the difference with the ground truth average comes from the ten or fifteen movies with the most scenes, e.g. *The Dark Knight* (2008), which has 219 scenes. To show that our higher scores are not due to it predicting fewer scenes, we added a unif-50 baseline, which predicts fewer scenes than our method but scores lower.
> > >
> > > |                | acc     | ari     | nmi     |
> > > |----------------|---------|---------|---------|
> > > | unif-50        | 0.474   | 0.305   | 0.725|
> > > | unif-60        | 0.461   | 0.278   | 0.720   |
> > > | unif-75        | 0.441   | 0.244   | 0.712   |
> > > | unif-90        | 0.430   | 0.216   | 0.704   |
> > > | unif-oracle    | 0.436   | 0.228   | 0.710   |
> > > | yeung96        | 0.321   | 0.220   | 0.541   |
> > > | psd            | 0.394   | 0.061   | 0.596   |
> > > | **ours**       | **0.564** | **0.375** | **0.746** |

---

### Official Review · Reviewer_Q22q · 2024-10-28

**Soundness:** 3
**Presentation:** 3
**Contribution:** 3
**Rating:** 6
**Confidence:** 3

**Summary:**

This paper presents a system for generating automatic screenplays and summaries from movies. The system is a combination of several well-established techniques, including keyframe extraction, speaker diarization, face recognition, and scene segmentation.

After carefully reviewing the authors' responses, I believe this research would be highly valuable if the authors shared their code with the community. I would be willing to change my rating to 'above threshold' if the authors decide to make their code open source.

**Strengths:**

- Good Application: This system has many potential everyday applications.
- Scene Detection: Introduces a novel, parameter-free scene boundary detection method based on the minimum description length principle.
- Character Naming: Proposes a new algorithm for identifying and assigning character names based on a database of actors' faces.
- Hierarchical Summarization: Employs a hierarchical approach for summarizing movies, first condensing individual scenes and then fusing them into a final synopsis.
- Zero-Shot Learning: Utilizes zero-shot prompting for summarization, demonstrating the potential of leveraging large language models without task-specific fine-tuning.

**Weaknesses:**

Novelty: The system primarily combines existing techniques like keyframe extraction, speaker diarization, and face recognition into a single pipeline. While the integration of these techniques is non-trivial, it may be more convincing if the paper can clarify the contribution of each component and identify their novelty.

More Comparison: The paper lacks a comparative analysis with current powerful LLMs, both publicly available models like GPT-4 and Gemini. In addition, it is not clear how the commercial system (Otter AI) is used in comparison. It will be crucial to dig deep to assess the  potential advantages and disadvantages of the proposed system.

**Questions:**

- It seems this paper only uses text and audio as input, and does not consider image and visual input. Is my understanding correct? [Updated: from author responses, this submission uses visual input].

- From the paper it is not clear what are the details of Otter AI. I am not sure whether the comparison is done by using the product, or reimplementing the paper published in 2023.

- Consider expanding the evaluation by including additional metrics, especially end2end user evaluation.

---

> ### Author Response · Authors · 2024-11-23
>
> Thank you for your review. Here are our responses.
>
>
> **The system primarily combines existing techniques like keyframe extraction, speaker diarization, and face recognition into a single pipeline.. clarify the contribution of each component and identify their novelty.**
>
> The two main novel components are the algorithms for scene-break detection and for character name assignment. The design of these algorithms, particularly the former, differs substantially from any existing algorithms for the same tasks.
>
> **Provide a comparison to publicly available models like GPT-4 and Gemini.**
>
> We compare only to open-source models. We do not know the training setup or internals of closed-source proprietary models, so we would argue there is little to be learnt by comparison with them. We have added a line explaining this reasoning to the Summarisation paragraph in Section 5.
>
> **It is not clear how the commercial system (Otter AI) is used in comparison.**
> Thank you for this comment. What we compare to is called Otter, a publicly available model that simply produces a short textual description of each scene. We use the code from https://github.com/Luodian/Otter. We mistakenly referred to this as ‘ Otter AI’ at some points in the paper. Thank you for pointing this out. We have now corrected it.
>
>
> **It seems this paper only uses text and audio as input, and does not consider image and visual input. Is my understanding correct?**
>
> No, our paper uses video and audio input. There is no text input required. This is in fact one of the major contributions of our model.
>
> **Consider expanding the evaluation by including additional metrics, especially end2end user evaluation.**
>
> Thank you for this suggestion. Unfortunately, are very expensive to obtain. The automatic metrics we use in their place have been shown to correlate with human judgements. We will pursue human evaluation in future work.

---

> > ### Comment · Reviewer_Q22q · 2024-12-01
> >
> > Thanks for the responses to my and other reviewers. If the authors would like to open source their code I think it will be very beneficial for the community. If it is the case, I would like to upgrade my rating to 6 Above threshold.

---

> > > ### Author Response · Authors · 2024-12-01
> > >
> > > Thank you for the reply, and we are happy to hear about raising your score.
> > >
> > > Yes, we fully intend to open-source the code alongside the final paper.

---

### Official Review · Reviewer_Vigi · 2024-11-03

**Soundness:** 3
**Presentation:** 3
**Contribution:** 2
**Rating:** 5
**Confidence:** 5

**Summary:**

The paper presents a new problem of automatically generating the screenplay of a full-length movie (which is typically of at least 90 minutes length) utilizing only the video and audio information. The primary motivation is for movie summarization (as inferred from the title of the paper). This assumes that the actual original screenplay, based on which the movie had been originally shot, is unavailable. The method has four main components - text transcript generation from audio which essentially performs speaker diarization, video scene detection, identification of the names of the characters in the movie and associating them with the transcript using the face images, and finally screenplay generation using caption generation of three key-frames per scene. The contributions are (a) the new task (b) scene boundary detection algorithm (c) detection of character names. Evaluation results are provided for scene detection, character name assignment and movie summarization in text using the generated screenplay.

**Strengths:**

The strengths of this paper are:

(1) The authors propose a new task for movie content processing i.e. generating the screenplay of the movie using only the video and audio content of that movie.

(2) The authors have proposed a solution framework, as captured in Figure 1, which is both plausible and also provides scope for improvement for follow-up work.

(3) The quality of the summary generated, as shown in the example, is very good even when compared to the gold standard.

**Weaknesses:**

(1) The motivation of the paper is somewhat weak and can hope fully be strengthened. It is assumed that the screenplay of a movie is not available. Given that screenplays exist for all movie created (including the nascent attempts at creating movies using diffusion models), why not just purchase them? They are apparently available at www.scriptly.com (for example the screenplay for the movie Oppenheimer is available for $24.95 on that site), it does not seem worthwhile to work on this problem. Other than just taking it up as an intellectual challenge. It would be good to strengthen the motivation. A possible way to strengthen it is by thinking of situations where the screenplay might not be available. Or if there is a divergence between the original screenplay and the actual movie which could be captured via the generated screenplay.

(2) In general, it appears that the authors are not familiar with the literature in the multimedia content processing and computer vision areas. Scene boundary detection has been researched there for decades and while the proposed method appears to be novel, a proper literature survey needs to be done. Even NetFlix and Amazon Prime have published methods:

https://netflixtechblog.com/detecting-scene-changes-in-audiovisual-content-77a61d3eaad6
https://www.amazon.science/blog/automatically-identifying-scene-boundaries-in-movies-and-tv-shows

A comparison with the state of the art methods such as the above ones and also in the literature, will help establish the novelty as well as the superiority of the proposed approach.

(3) The name assignment method is simple and effective. However, it appears that there could be misses and also incorrect assignments. The frequency of its occurrence seems to be roughly 34% from Table 2. It is not clear what is the impact of these errors on the quality of the screenplay generation and also on the summarization?

(4) While the paper provides evaluation results for the summarization task, no evaluation is provided for the main work of the paper which is the screenplay generation. Why is there no quality evaluation of the screenplay generated by the proposed method? Only the indirect measure via the quality of summary generated is provided. It would strengthen the paper if an evaluation is done by comparing the generated screenplay with the original screenplay, perhaps using some of the text processing metrics used for the summary comparisons.

**Questions:**

(1) Please discuss the quality of the screenplay output of the method.
(2) What would be the quality of the summary generated by the name-only prompt of the current leaders of the ChatBot Arena (say ChatGPT-4o, Gemini 1.5 and Claude 3.5)? if it is substantially better than the proposed method, then the case for the proposed research will be weakened.
(3) In Table 3, the name-only prompt results are actually quite impressive. It is not clear why the quality degrades when more information is provided i.e. full script and whisperx script. That appears to be a bit counter-intuitive.

---

> ### Author Response · Authors · 2024-11-23
>
> Thank you for your review. Here are our responses.
>
> **Why not just purchase screenplays?**
>
> As you mention, the actual dialogue often deviates significantly from the official screenplay. We have observed this in our dataset, for example in V for Vendetta (2004) and Somethings’ Gotta Give (2003). There are also long videos for which screenplays are not available, such as documentaries. We are working on producing some generated summaries for documentary videos, which we will update the paper with as soon as they are ready. Additionally, $25 is not an insignificant amount to pay per video if one wants to purchase screenplays.
>
> **A proper literature survey on the multimedia content processing and computer vision areas, and comparison with SOTA models, needs to be done.**
>
> We have added further related work on video captioning and LLMs for video understanding. We have also added two more comparison models for the scene segmentation task, which are now in Table 1.
>
> **What is the impact of name assignment errors on the quality of the screenplay generation and on the summarization?**
> This is tested in the 'w/o names' ablation setting in Table 4, where we see a moderate degradation in summary quality. We observe that the LLM is sometimes able to put the correct name in the summary from the dialogue only, but that this is sometimes inaccurate.
>
> **There should be an evaluation comparing the generated screenplay with the original screenplay, perhaps using some of the text processing metrics used for the summary comparisons.**
> Thank you for the suggestion. Unfortunately, it is very difficult to automatically evaluate the quality of the generated screenplay. Text comparison metrics such as ROUGE or BLEU are designed for comparing much smaller texts of a couple of sentences, and struggle even with paragraph-length comparison. We have added several example generated screenplays to the supplementary material which can be evaluated qualitatively. Instead, we choose to indirectly evaluate the screenplay via the summarisation task.
>
> **What would be the quality of the summary generated by the name-only prompt of the current leaders of the ChatBot Arena (say ChatGPT-4o, Gemini 1.5 and Claude 3.5)? if it is substantially better than the proposed method, then the case for the proposed research will be weakened.**
> Thank you for the suggestion. We make the deliberate choice to compare only to open-source models. We do not know the training setup or internals of closed-source proprietary models, so we would argue there is little to be learnt by comparison with them. We have added line explaining this reasoning to the Summarisation paragraph in Section 5.
>
> **The name-only prompt results are actually quite impressive, why does the quality degrade when more information is provided i.e. full script and whisperx script.**
> When prompted with the name only, the model very likely effectively regurgitates an existing online summary. When given a very long prompt that includes dialogue and event descriptions, the model tries to actually summarise the information given, during which it can make mistakes. This is evidenced, for example, by the fact that, in the name-only setting, it simply repeats the same information over and over when forced to give a longer output. This suggests that it has a fixed-length answer it is retrieving, rather than dynamically generating the summary in response to the given information. We have added this analysis to the the Summarisation paragraph in Section 5.

---

> > ### Comment · Reviewer_Vigi · 2024-11-23
> >
> > Thank you for your response - it is helpful.

---

### Official Review · Reviewer_NG5j · 2024-11-03

**Soundness:** 3
**Presentation:** 3
**Contribution:** 2
**Rating:** 3
**Confidence:** 5

**Summary:**

This paper proposes a complete system to take an entire movie with associated audio track and automatically creates a textual summary of the movie. For the system a set of mostly existing methods are integrated into a full solution. The two elements that are (according to the authors) new are the scene segmentation and character identification method, for the text generation a standard LLM is employed.

**Strengths:**

- a complete system addressing all modalities
- well written, easy to read
- potential for interesting applications

**Weaknesses:**

- The tasks for the given "new" algorithms go back to the 90's of the previous century. Yet the methods are compared to non-algorithmic baselines like just doing random or uniform assignment. For example the Scene Transition Graph by IBM from 1997 (https://www.spiedigitallibrary.org/conference-proceedings-of-spie/3312/1/Classification-simplification-and-dynamic-visualization-of-scene-transition-graphs-for/10.1117/12.298470.short) and the NameIT system by Satoh (Name-it: Naming and detecting faces in news videos
S Satoh, Y Nakamura, T Kanade - Ieee Multimedia, 1999). Both have a multitude of citations so hence many follow-up systems. The authors should do a thorough review of the field and compare their methods to the best performing ones.
- Overall the system seems very much to be extensions of the work of Lapata which are the only ones mentioned in long-form summarization. Authors should do a far more in-depth analysis of the field. I googled "automatic summarization of complete movies" and already got a number of papers which are related. Authors should indicate how their method is different from those.
- This is not a typical ICLR paper where the main contribution of the paper is on the algorithmic side. As indicated above the contribution in terms of algorithms is limited so the main novelty is the integration of all of these into one system. Other venues like ACM Multimedia, SIGIR, or a similar conference are far more suited for that topic, so my suggestion would be to improve the work and submit the work there.

**Questions:**

Questions:
- What makes the two algorithms innovative if there is already so much out there?

- The objective: the movie application is exactly where these kind of techniques for automated objective textual summarization seem less interesting as you can find a movie summary for every movie at the IMDB website and if not, likely the actors can also not be found there. Personalized summaries of movies are in my opinion more interesting (although even more difficult to evaluate, you should probably work with real users for that). Another issue with the objective is that a typical summary should not contain any spoilers where the current system doesn't take that into account.

Remarks:

Scoping:  The paper says that summarization is usually done from the text perspective but that is maybe true for the NLP community but in Computer Vision there is (and has been from the 90's) a lot of work on visual video summarization. Many of those are referring to the datasets you are also mentioning (like MovieSum). So the authors should remove this claim (or substantiate it with evidence based on citations) and consider the computer vision driven video summarization literature and make more clear how the current work is different.

Ref [Abhimanyu Dubey] is not proper. The CLIP website actually gives you a proper way to refer to it.

---

> ### Author Response · Authors · 2024-11-23
>
> Thank you for your review. Here are our responses.
>
> **Compare scene-break alg to Scene Transition Graph by IBM from 1997.**
>
> We have added this comparison as you suggest.
>
> |                | acc     | ari     | nmi     |
> |----------------|---------|---------|---------|
> | unif-60        | 0.461   | 0.278   | 0.720   |
> | unif-75        | 0.441   | 0.244   | 0.712   |
> | unif-90        | 0.430   | 0.216   | 0.704   |
> | unif-oracle    | 0.436   | 0.228   | 0.710   |
> | yeung96        | 0.321   | 0.220   | 0.541   |
> | scenedetect            | 0.394   | 0.061   | 0.596   |
> | **ours**       | **0.564** | **0.375** | **0.746** |
>
> We have updated the paper with these results.
>
> **There should be a more in-depth discussion of existing methods for movie summarization.**
>
> We are not aware of any existing summarisation methods that (a) use only the video as input and (b) operate on very long videos, e.g. movie length. We have already included some discussion of long summarisation using text instead of video. In response to your comment, we have added references to the literature on summarising short videos: Sridevi & Kharde (2020); Seo et al. (2022); Zhou et al. (2018b) and Yang et al. (2023), which differ from our work in that the small video size allow them to use end-to-end models.
>
> **The contribution in terms of algorithms is limited.**
>
> We strongly disagree with the claim that the algorithms we propose are not novel. There are indeed other attempts to solve the same problems, such as segmenting a video into scenes, but the internal design of our scene segmentation algorithm is very different from existing methods. We are grateful for the suggestion to discuss some of these existing methods in our paper. The use of the minimum description length principle in particular is highly novel, allowing our algorithm, among other things, to automatically determine the number of scenes. While the problem of scene segmentation is not novel, our proposed solution is.
>
> **The movie application is exactly where these kind of techniques for automated objective textual summarization seem less interesting as you can find a movie summary for every movie at the IMDB website and if not, likely the actors can also not be found there.**
>
> There is an advantage to being able to generate the summary, as you can control the length and detail. The summaries you mention on the IMDB website are mostly restricted to 239 characters–about 1/10th the length of the summaries we generate. Additionally, long-form video-only summarisation is a useful testbed for developing video understanding systems in general.
>
> **Personalized summaries of movies are in my opinion more interesting.**
> Thank you for the suggestion–we have added this as a future work.
>
> **A typical summary should not contain any spoilers where the current system doesn't take that into account.**
>
> We are generating Wikipedia summaries, which include all plot information, including spoilers. These are detailed summaries of what happens in a movie, and test the models' understanding capabilities.  We are not generating synopses, which are intended to entice viewers to watch a show without revealing too much of the plot.
>
>
>
> **Many early works use the same datasets as us.**
>
> The MovieSumm dataset was only released in 2024.
>
> **Ref [Abhimanyu Dubey] is not proper.**
>
> Thank you, we have fixed this.

---

> > ### Comment · Reviewer_NG5j · 2024-11-25
> > **Thanks for the answers**
> >
> > The additional comparisons have made the paper stronger. Yet I still don't see this as a contribution that is fitting the scope and quality of ICLR. I do think that with the additional experiments and analysis it has a much better chance of being accepted at an appropriate venue.

---

> > > ### Comment · Reviewer_47gi · 2024-11-25
> > > **Information Leaking issue**
> > >
> > > Hi reviewer, Can you see the author name in current version? I am wondering if it is my problem or the author accidentally upload the wrong pdf.

---

> > > > ### Author Response · Authors · 2024-11-26
> > > >
> > > > I very much apologize for this. It seems I accidentally uncommented the relevant line in the preamble while adding new packages to display the additional results. I fixed this as soon as I saw it, and the names are anonymized again.

---

### Meta-Review · Area_Chair_mrGN · 2024-12-12

**Metareview:**

The paper proposes a movie understanding problem and solution. The reviewers appreciate the problem, writing, and aspects of the solution. However they raise concerns about motivation (given availability of screenplays), novelty (limited algorithmic contribution, limited discussion of broader prior work), and experiments (baseline selection, lack of comparison to commercial LLMs). There is some support for acceptance but stronger opposition (three negative and one positive score).

**Additional Comments On Reviewer Discussion:**

The reviewers engaged in the discussion in meaningful manner

---

### Decision · Program_Chairs · 2025-01-22

Reject